# A Water-Energy-Food Nexus Approach for Conducting Trade-off Analysis: Morocco's Phosphate Industry in the Khouribga Region

Sang-Hyun Lee[a], Amjad T. Assi[b], Bassel T. Daher[b], Fatima E. Mengoub[c,d], Rabi H. Mohtar[b,e,f]*

[a] Research Institute for Humanity and Nature, Kyoto, Japan [sanghyunsnu@gmail.com].
[b] Department of Biological and Agricultural Engineering, Texas A&M University, College Station, TX 77843-2117, USA [amjad.assi@tamu.edu; basseldaher@gmail.com]
[c] Policy Center for the New South, Suncity Complex, Building C, Angle of Addolb Boulevard and Albortokal Street, Rabat, Morocco [f.mengoub@policycenter.ma]
[d] Mohammed VI Polytechnic University, Lot 6620, Hay Moulay Rachid 43150, Benguerir, Morocco
[e] Zachry Department of Civil Engineering, Texas A&M University, College Station, TX 77843-3136, USA.
[f] Department of Agricultural and Food Sciences, American University of Beirut, Beirut, Lebanon.

* Correspondence: Rabi H. Mohtar, Professor and Dean, Faculty of Agricultural and Food Sciences, FAFS 101, American University of Beirut, P.O. Box 11-0236, Riad El Solh, Beirut 1107 2020, Lebanon.  Tel.: +961-1-350000 ext: 4400, Fax: +961-1-744460, [mohtar@aub.edu.lb; mohtar@tamu.edu].

## Abstract

The study's objective was to develop and use the Water-Energy-Food Nexus-Phosphate (WEF-P) Tool to evaluate the impact of Morocco's phosphate industry on water, energy, and food sectors of Khouribga, the representative phosphate mining region of Morocco. The developed WEF-P Tool enabled a trade-off analysis based on integrating the supply chain processes, transportation, and water-energy footprints of the region. Field data from the mining to transportation processes were collected and applied to possible supply chain scenarios in accordance with the type of product (phosphate rock and slurry). The potential impacts of the scenarios were considered in terms of the water supply in the agricultural areas. The analysis of the positive impacts of dynamic management suggests that seasonal management of phosphate production (less during the irrigated season, more during wetter/rainier seasons) is more effective. Additionally, while the transport of raw phosphate slurry through a pipeline increased the total water required to 34.6 million m³, an increase of 76% over the "business as usual scenario" (BAU), it also resulted in an energy savings of nearly 80% over BAU: slurry transport requires only 40.5 million litres of fossil fuel, instead of the 204 million litres required to transport rocks. During the dry or "water scarce" irrigated season (May to July), total ground water use decreased from 5.8 to 5.2 million m³. Dynamic management of the phosphate industry can also save 143 MWh of electricity annually and brings a reduction of 117 tons of $CO_2$ emissions. Making water available at the correct season and location requires analysis of complex scientific, technical, socio-economical, regulatory, and political issues. The WEF-P Tool can assist by assessing user-created scenarios, thus it is an effective management-decision aid for ensuring more sustainable use of limited resources and increased reliability of water resources for both agricultural and industrial use. This study on the application of WEF Nexus to the Phosphate industry offers a roadmap for other industrial application for which trade-offs between the primary resources must be considered.

*Keywords: Phosphate, Water-Energy-Food Nexus, Morocco, WEF-P Tool*

## 1 Introduction

Nexus thinking emerged from the understanding that natural resource availability can limit and is limited by, economic growth and other goals associated with human well-being (Hoff, 2011; Keulertz, 2016). The innovative aspect of nexus thinking is its more balanced view of the issues linking resources (Al-Saidi and Elagib, 2017). Thus, nexus frameworks identify key issues in food, water, and energy securities through a lens of sustainability and seek to predict and protect against future risks and resource insecurities (Biggs et al., 2015). The 2015 World Economic Forum identified water, food, and energy shocks as primary future risks, calling for increased efficiency in water use across all sectors and the implementation of integrated water resources management. Various conceptual frameworks relating to the nexus approach were developed: the FAO (2014) emphasized the role of the nexus in food security; the International Renewable Energy Association (IRENA, 2014) applied the nexus approach in transforming conventional energy systems to renewable systems.

The demand for water, energy, and food, is expected to increase due to drivers such as population growth, economic development, urbanisation, and changing consumer habits (Terrapon-Pfaff et al., 2018). The interlinkages across key natural resource sectors and improving their production efficiency offer a win-win strategy for environmental sustainability, whether for current or future generations (Ringler et al., 2013). Accordingly, application of the Water-Energy-Food (WEF) nexus concept or approach is expected to make implementation of the Sustainable Development Goals (SDGs) more efficient and robust (Brandi et al., 2014; Yumkella and Yillia, 2015). The SDGs are classic examples of the necessity to acknowledge multidimensional, nexus interlinkages and trade-offs, particularly as governments are challenged to maximize benefits and invest limited resources. Infrastructure and capital are needed to achieve national SDG targets and the nexus concept is now used to highlight interdependencies between resources and the need for integrated, sustainable governance and management of those resources (Pahl-Wostl, 2019).

The debate surrounding effectively addressing water and food security challenges stems from questions about whether the water-food crisis is due to a poor understanding of the resources or to their improper management (Mohtar et al., 2015). One long-standing challenge to water management lies in the lack of integration among the multiple sectors that interact with the water sectors across geographical areas or within large, transboundary, basins (Mohtar and Lawford, 2016). Projections about availability and quality of water, food, energy, or soil resources are often alarming. A fundamental shift is needed away from traditional 'silo' approaches and toward more integrative, systems approaches (Daher and Mohtar, 2015). Energy and water are crucial for economic growth, especially in industrialized areas (Flörke et al., 2013; Cai et al., 2016), making the rapid increase in demand for these resources a serious issue for both economics and the environment. While technology to reduce industrial demand for water and energy is important, we must also understand the relationship between economic growth, water–energy consumption, the impact of industrial activity on agriculture at the local level. Increase of industrial products can cause steep increases in demand for water and energy, which in turn, leads to issues of downscaling water or energy securities.

The nexus framework is dependent on the stakeholders, system boundary, and analytical tools. In considering the application of the nexus as a platform, an integrated modelling approach is essential. These issues manifest in very different ways across each sector, but their impacts are often closely related in terms of trade-offs. In particular, the sub-nexus needs to be effectively conceptualized and a theoretical sub-nexus developed. Private-sector water, energy, and food supply chain players are the key stakeholders to address current contradictions arising as a consequence of attempts to develop a grand nexus approach (Allan et al., 2015). Accordingly, we must consider the "specialized" nexus of multi-stakeholders, such as agriculture, industry and urban areas, for which water, energy and food are treated as subsystems. Current nexus frameworks often focus on macro-level drivers of resource consumption patterns (Biggs et al., 2015), but major nexus challenges are faced at local levels (Terrapon-Pfaff et al., 2018). Thus, 'larger scale' extraction and consumption of natural resources may lead to depletion of natural capital stocks and increased climate risk with no equitable share of the benefits (Hoff, 2011; Rockström et al., 2009). Al-Saidi and Elagib (2017) showed the importance of exploring driving forces and interactions at different scales in the

conceptual development of the nexus, emphasizing more case-study based recommendations in the reality of institutions, bureaucracies, and environmental stakeholders.

Morocco's phosphate and agriculture industries offer an example of increasing resource pressures attributable to near- and medium-term growth across these sectors (Taleb, 2006). A holistic approach that considers the needs of all stakeholders is necessary to resolve resource allocation pressures. Between 1990 and 2016, Morocco's population grew from 25 million to 35 million people (World Bank, 2019a). Both crop production and total cultivated land significantly increased since 1971, and half of Morocco's arable land receives less than 350 mm of rainfall annually and nearly 87.3% of Morocco's total water withdrawals are used for agriculture (FAO, 2015). Per capita consumption of electric power increased from 358 kWh (1990) to 901 kWh (2014); energy use by oil equivalent per capita increased from 306 to 553 kg during the same period (World Bank, 2019b). Proper management of water, energy, and food resources is critical to economic, social, and environmental well-being. Globally, phosphates lie at the heart of agriculture and soil enhancement. More than 75% of global phosphate reserves, representing 30% of the global market share, are found in Morocco, positioning that country for a leading role in global food security (OCP, 2013). Phosphate mining and its chemical processing require considerable water, energy, land, and other resource inputs. Morocco uses recycling and reverse osmosis desalination to relieve some of the pressure on its fresh water resources and help secure the water necessary for phosphate production processes (OCP, 2016b). Each water source carries a distinct energy tag that must be accounted for, especially in a country that imports nearly 90% of its consumed energy (World Bank, 2019c). Water, energy, land, and financial resources are frequently shared between multiple sectors, especially agriculture (food production) and municipal (growing urbanization): Morocco is no exception. It is critical that potential sectoral competition be understood, quantified, and accounted for when planning for the sustainable progress of all sectors. An integrated approach to resource allocation is needed to minimize inevitable competition for resources: one that quantifies the trade-offs associated with the possible pathways. As Morocco heads toward achieving its phosphate production goals, the ability to account for the resources associated with that achievement should be balanced with the associated (and increasing) agriculture and municipal demand projections: this is key to sustainable resource allocation (OCP, 2013).

This study adapted the WEF Nexus Tool linking industry and agriculture to integrate the supply chain for industrial products. Using the Tool, the authors evaluated the impact of Morocco's phosphate production on the water, energy, and food resource systems of its mining region and then addressed the resource elements in the supply chain management of phosphate production. Specifically, they assessed the impact of phosphate mining and transportation by slurry pipeline on potential water and energy savings in the mining area. The results suggest the need for dynamic management of phosphate production, one that adjusts monthly phosphate production in consideration of its potential impacts on water and energy management in agricultural areas. The specific objectives of the study are to quantify the water, food, energy used by the phosphate industry in the Khouribga region of Morocco and to assess the trade-offs of resource allocations between agriculture and industry.

## 2 Materials and methods

### 2.1 Site description

We contacted the managers and engineers working in the Office Cherifien des Phosphates (OCP) group which is that country's leading phosphate producer in Morocco, and had a lot of discussion about the site, data, policy, and goals. OCP group accounts for 3% of the country's gross domestic product and about 20% of national exports in value over the course of the 20th century (Croset, 2012). The OCP group ran three mining fields: in central Morocco, near the city of Khouribga, and on the Gantour site. Khouribga, the largest mining area, includes three main sites from which raw phosphate is excavated and transported for chemical processing and fertilizer production: Sidi Chennane (SC), Merah Lahrach (MEA), and Bani Amir (BA) (Figure 1). The output in Khouribga is raw phosphate produced as either rock or slurry, the main component of manufactured phosphorous fertilizers. The transport of the phosphate (rocks and slurries) from Khouribga (mining area) to Jorf Lasfar (industrial

production area) is a primary project in Morocco (OCP, 2016a). The demand for raw phosphate and the production and export
of fertilizer and its products from Jorf Lasfar drive the upstream mining activity of Khouribga. In 2015, approximately 20.1
million tons of raw phosphate were excavated, which was 58 % of total raw phosphate excavated in Morocco in 2018 (OCP,
2020), and transported to Jorf Lasfar; about 40% of this product was transported via pipeline as slurry and the balance via train
as rock.
The pipeline from Khouribga to Jorf Lasfar is 187 km and ensures the continuous transport of phosphate from the Khouribga
to Jorf Lasfar (Figure 1). As the plan was to increase phosphate production and phase out transport by train, tracks were
replaced by pipeline that ensures the continuous flow of raw phosphate from the mining to the industrial area (OCP, 2016a).
The plans impact regional water, energy, and food management: in particular, shifting from train to pipeline requires additional
water to convert dry rock into liquid slurry. Shifting from train to pipeline changes the demand for water and energy resources
at both the mining and the production locations.
In accordance with the "Green Morocco (Plan Maroc Vert)" (Stührenberg, 2016) and the National Water Plan for Morocco,
the use of surface water as a substitute for groundwater is encouraged: water withdrawals from regional aquifers are being
phased out since 2010, to be replaced entirely by surface water from the nearby Ait Messaoud dam, which has a capacity of
13.20 million m³. The plan is to allocate 4.5 million m³ yr$^{-1}$ of water from the dam to the mining site. Additionally, OCP
launched a plan to complete treatment plants for urban wastewater (capacity 5 million m³ yr$^{-1}$) to be used for washing phosphate
and industrial reuse in the mining area (OCP, 2016b). The phosphate mining area is encircled by cropland, whose water is also
supplied from the dam. In this study, the authors consider the allocation of treated water to both the phosphate industry and
agricultural irrigation (Tian et al, 2018). Both the mining and the agricultural activities of the region represent growing
enterprises that place added pressure on available water resources, making the sustainable management of the water supply a
hotspot to be considered in trade-off analyses.

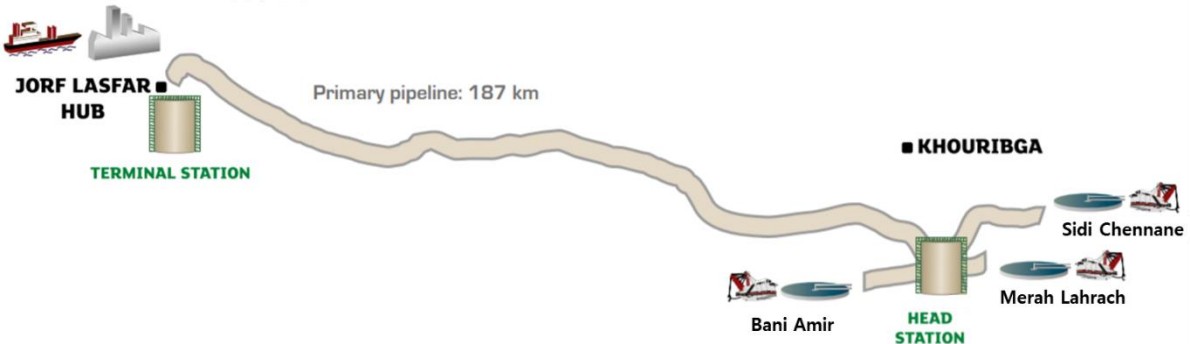


**Figure 1** Study area: phosphate mining area (Khouribga), fertilizer manufacturing area (Jorf), and transportation system
(slurry pipeline) (http://www.ocpgroup.ma/ocpslurrypipeline/slurry-pipeline)
**2.2 Development of Water-Energy-Food-Nexus - Phosphate (WEF-P) Tool**
*2.2.1 Overall Framework of WEF-P Tool*
The developed WEF-P Tool, adapted from the WEF Nexus Tool 2.0 (Daher and Mohtar, 2015), considers the supply chain of
final product in terms of its resource consumption, including the set of processes that pass materials forward (La Londe and
Masters, 1994; Mentzer et al., 2001), and various organizations or individuals directly involved in the flow of products
(Mentzer et al., 2001). It assesses the impact of various scenarios and possible responses to regional resource management
needs. Table 1 shows the differences between WEF Nexus Tool 2.0 and WEF-P Tool in the context of variables, scenarios,
analytical tools, and quantitative assessments.
Both the Tools offer a platform for development of the analytics necessary to understand the trade-offs and catalyse a
stakeholder dialogue (Mohtar and Daher, 2016; Mohtar, R. H. and Daher, 2014). The core of the WEF Nexus is that production,
consumption, and distribution of water, energy, and food are inextricably inter-linked: decisions made in one sector impact the

other sectors (Hoff 2011, Mohtar and Daher, 2012). The WEF Nexus Tool 2.0 allows holistic quantification of the impact of resource allocation strategies to support informed, and inclusive stakeholder dialogue between policy makers, private sector firms, and civil society (Daher and Mohtar, 2015). Each stakeholder becomes involved at different stages and scales in the decision-making process. In the WEF-P Tool (Figure 2), water resources are shared between the phosphate industry and agricultural interests in the region of study. Sustainable water management must holistically consider the allocation of water resources for both industrial production and agricultural irrigation. New water (treated urban wastewater) has the potential to contribute significantly to bridging water and food gaps (Mohtar et al., 2015). However, it carries an energy footprint that must be considered when increasing local food production. Potentially, agriculture's demand for water competes with those of a growing industry. The Tool quantifies the use of water and energy and the amount of $CO_2$ emitted for each scenario. It also quantifies the water and energy savings resulting from choices made regarding transportation scenarios. The Tool assesses the effects of decisions of dynamic management of phosphate production as these impact water and energy securities. The WEF-P tool can assess various scenarios and help account for interdependencies between food and industrial production, and between water and energy consumption, thus allowing the trade-offs associated with potential resource allocation pathways to be quantified.

Throughout the tool development process, the supply chain was verified with OCP and the OCP Policy Center in various ways: (i) during the data collection phase, through meetings with the OCP steering committee, financial managers, technical managers and engineers; and (ii) through follow ups with OCP Policy Center team (conference calls and email). The OCP Policy Center team shared with WEF Nexus Team their main concerns regarding the tool structure, based on input from the OCP technical team. The WEF Nexus Team used these shared concerns in their considerations of revisions to the tool structure and associated excel spreadsheets of the model. Specifically, the major aggregated processes and lines of productions were revised and identified in a functional supply chain to maximize the abilities and flexibilities of the model and ensure efficacy of the available data base for processes and production lines.

However, the WEF-P Tool has limitations in assessing economic impacts such as cost and benefit analysis. This is because cost must include the price of water, which is still under discussion, and the price of products when analysing their benefits. Raw phosphate is transported to the manufacturing area and used in the production of various fertilizers that have different prices: this makes it difficult to set the price of excavating raw phosphate in the mining area. Sustainability assessment also has qualitative aspects in terms of environmental impact. The WEF Nexus Tool 2.0 applied the sustainability index based on resource capacity and availability, however, it is still a quantitative aspect. We should consider the meaning and definition of sustainability, both quantitatively and qualitatively, and then assess the index using the stakeholders' weights for the variables related to sustainability. Additionally, spatial and temporal scales should be included in a sustainability index. For example, the pipeline transportation system requires water, which is transported with products: the pipeline causes greater water use at the origin, but also provides additional water to the destination area. Also, the water requirement differs with temporal season, such as the water intensive agricultural production season. Thus, more research is needed for a sustainability assessment based on economic and environmental impact. However, the quantitative analysis is an essential factor for assessing sustainability, therefore, the WEF-P Tool focuses on quantification of 1) water and requirements for phosphate production and transportation, 2) carbon emissions by energy used in product processes, 3) water supply system and transportation, and 4) dynamic production impacts on water and energy savings.

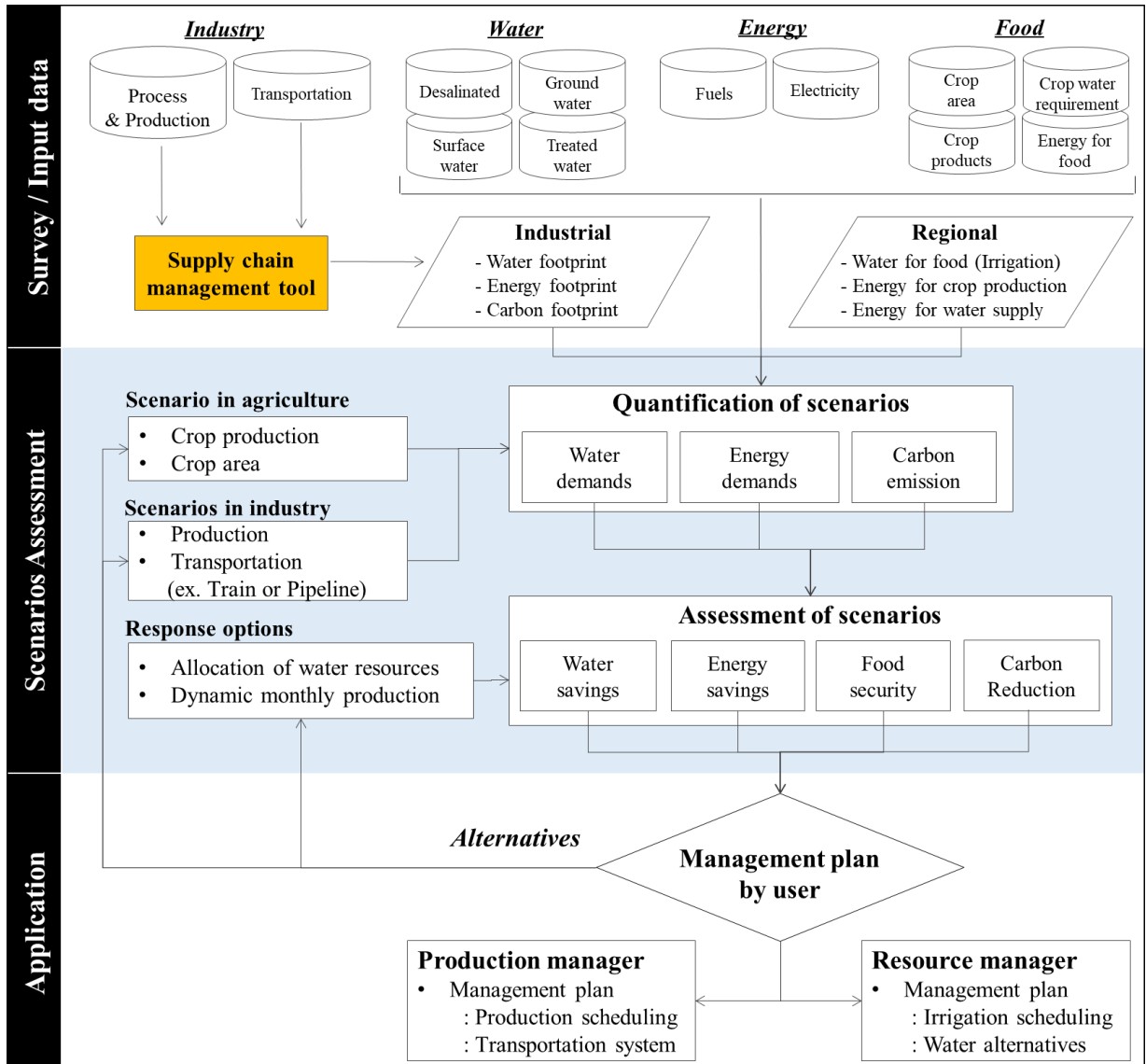

**Figure 2** Assessment of holistic impacts of various scenarios relating to phosphate industry, agriculture, and resource
management using WEF-P Tool

**Table 1** Comparison between WEF Nexus Tool 2.0 abd WEF-P Tool

| | WEF Nexus Tool 2.0 | WEF-P Tool |
|---|---|---|
| Variables and scenarios | • Self-sufficiency of produced crops<br>• Type of agricultural production<br>• Sources of water (groundwater, surface water, treated water and so on)<br>• Sources of energy (natural gas, diesel, solar, wind and so on)<br>• Trade portfolio (countries of import and amounts per country) | • Static and dynamic phosphate production<br>• Transportation modes (train and pipeline)<br>• Sources of water (groundwater, surface water, treated water and so on)<br>• Water allocation between industry and agriculture |
| Analytical tool | • Food product base analysis<br>• Food-centric interlinkages among water, energy, and food<br>• Water and energy footprint based on product (ex. water footprint of crops) | • Process base analysis<br>• Phosphate-centric interlinkages among production, transportation, and resource allocation<br>• Water and energy footprint based on processes (ex. water footprint in washing process) |
| Quantitative assessment | • Water requirement for energy and agricultural production<br>• Energy requirement for agricultural and water production<br>• Land footprint for agricultural and energy production<br>• Carbon emissions from energy used for water and food production<br>• Financial cost | • Water and requirement for phosphate production and transportation<br>• Carbon emission by energy used in product processes, water supply system and transportation<br>• Dynamic production impacts on water and energy savings |

*2.2.2 Analysis of integrated supply chains linking sub-processes and transportation modes*

The WEF-P Tool used the WEF Nexus approach to assess the life cycle of the final products supply chain. The water and energy used to produce sub- and final products were calculated by adding the water and energy requirements from the sub-processes through the production supply chain. In Khouribga, raw phosphate products pass through a sequence of functional processes: Mining and Screening (S), Washing (WW), Grinding (WG), Flotation (WF), Adaptation including powdering (WA), and Drying (COZ - Complex Ouad Zem) for SC and (UB) for MEA (Figure 3). The mining and screening processes include extraction from the ground, tone removal, and screening to produce pieces of phosphate rock. Here, the supply chain is determined by the quality and size of the phosphate rock, which in turn, depends on the phosphate content at extraction, ranges from very low to high. High quality phosphate rock is transported to a drying process from which it will either be marketed or chemically transformed into fertilizer at the manufacturing site. Low to medium quality phosphate rock goes is washed, dried, ground, and subjected to flotation, intended to increase the phosphate content.

The change in transportation system can affect the supply lines (Figure 3). In the mining area, the products are phosphate rocks and slurry, both of which are transported to the manufacturing area, each with its own resource requirements. Slurry requires flotation and adaptation, thus is more water intensive; phosphate rock is dried in an energy intensive process that consumes most of the energy produced in the mining area. Slurries are transported via pipeline and rock by train; each mode has distinct resource needs at different stages. The two transportation systems are also distinct: the pipeline supply chain includes washing (water) and adaptation to produce slurry; the train supply chain includes a fuel intensive drying process. It is possible to quantify the flow of products according to the transportation system used. When transport changes from train to pipeline, supply lines also change: the drying process is replaced by the adaptation process. If the phosphate is transported by pipeline, it must first be transformed to slurry, adding the adaptation process to the supply chain. Changes in the supply chain impact

the water and energy consumed and, consequently, the $CO_2$ emitted. The mining and screening processes include extraction
from the ground, tone removal, and screening to produce pieces of phosphate rock.

**Figure 3** The functional processes and flows of products in Khouribga (mining) by transportation method.

*2.2.3 Adaptation of process-base water, energy, and $CO_2$ footprints*

The main function of the WEF-P Tool is identification of the relationship between resources and production, and the
quantification of the resources consumed in phosphate production. The methodology is based on life cycle assessment. The
water and energy footprints were analysed, indicating the quantity of water or energy consumed in various sub-processes in
the supply chain's integration of production and transportation. The technical details of each process are specific and
aggregated into functional processes. The main component is the footprint, which indicates the water and energy requirements
for phosphate products, and the $CO_2$ emitted through energy consumption. Each process has a specific footprint based on field
data and fed into the tool monthly, or when a significant change in capacity of the functional processes has occurred. For all
footprint processes in Khouribga, the amount of raw phosphate is measured in commercial metric tons embedded in slurries
and rock. Even if the phosphate rock changes to slurry through several processes, the amount of raw phosphate embedded in
products is not changed. Thus, the tons of phosphate in water and energy footprints indicate the raw phosphate embedded in
the products in each process and is constant through entire supply chains.
From the technical (engineering) perspective, footprints are calculated using a regression function, or average value based on
survey data; technical experts in each process can modify this relation function as needed. The WEF-P Tool uses historical
data (from 2015) to estimate the average value of the footprint and the relationship between water/energy consumption and
phosphate production. First, the relationship between outputs of each process and water (or energy) consumption was analysed.
Second, the WEF-P Tool considered transportation of water and consumption of energy by train and pipeline. Transportation
by train was only related to fuel, i.e., diesel, consumption. However, the pipeline station consumes electricity for operating the
pipeline and freshwater is transported with slurry. The pipeline should be full, but as it is impossible to fill the pipeline with
slurry, it alternately carries slurry and freshwater. Therefore, total water (or energy) consumption in the mining area includes
not only water (or energy) used in processes but also that used in transportation systems and the water consumed at the pipeline
station in mining area, which basically indicates the transported water used in the manufacturing area. WEF-Tool could
quantify water and energy consumption of the various processes and at the pipeline station, as shown in equations (Eqs. 1-5).

$$WC_{mining\ area} = \sum_i^n (P_i \times WFP_i) + WC_{pipleline\ station} \quad\quad (1)$$

$$WC_{pipleline\ station} = P_{slurry} \times WC_{pipleline\ station} \quad\quad (2)$$

$$EC_{mining\ area} = \sum_i^n (P_i \times EFP_i) + EC_{pipleline\ station} + EC_{train} \quad\quad (3)$$

$$EC_{pipleline\ station} = P_{slurry} \times EFP_{pipleline\ station} \quad\quad (4)$$

$$EC_{train} = P_{phosphate\ rock} \times EFP_{train} \quad\quad (5)$$

where $WC_{mining\ area}$ (m³) is total water consumption in mining area, $EC_{mining\ area}$ (MWh or L) is total energy consumption in mining area, $P_i$ (ton) is production from each process (i) in mining area such as mining, screening, washing, flotation, and drying. $WFP_i$ (m³ ton$^{-1}$) and $EFP_i$ (MWh ton$^{-1}$ or L ton$^{-1}$) are water and energy footprints in each process (i). $WC_{pipeline\ station}$ (m³) is water consumption in pipeline station, $EC_{pipeline\ station}$ (MWh or L) is energy consumption in pipeline station, and $EC_{train}$ (MWh or L) is energy consumption by train to transport phosphate rock to manufacturing area. $P_{slurry}$ and $P_{phosphate\ rock}$ (ton) are production of slurry and phosphate rock. $WFP_{pipeline\ station}$ (m³ ton$^{-1}$) is water footprint at pipeline station in mining area. $EFP_{pipeline\ station}$ and $EFP_{train}$ (MWh ton$^{-1}$ or L ton$^{-1}$) are energy footprints in pipeline station and of transportation by train. It is worth mentioning that the tool distinguishes between two types of water: water transported from mining to manufacturing area by pipeline, and the embedded water in slurry.

$CO_2$ emissions are relevant when assessing the environmental impact of phosphate production. Although real emission in each process in supply chain should be measured, this study is limited measuring $CO_2$ emission in mining area. In addition, $CO_2$ emission in crop area is related to soil and crops, and it is another level of research. Thus, we limited $CO_2$ emission to that emitted by fuel energy use by machinery (direct emission) and electricity generation in power plants (indirect emission), and the reference $CO_2$ footprints were applied (Table 2). Fossil fuels (gasoline, diesel, coal, etc.), when burned, produce direct $CO_2$ emissions. Indirect $CO_2$ emission is also related to the source fuel used in generating electricity: indirect emission occurs in the generation of electricity from other (non-fossil) sources, such as hydroelectric, wind power, or solar. According to USEIA (2019), one litre of gasoline used by machinery or a facility produces 2.6 kg of direct $CO_2$ emission; a power plant burning only coal emits 1,026 tons of $CO_2$ kWh$^{-1}$. Renewable (non-fossil) electricity emits only 15.8 tons of $CO_2$ kWh$^1$. A survey of sources of electricity generation in Morocco indicates that coal is the main fuel for power generation (43.4% of the national production). Oil and natural gas account for 25.3% and 22.7% respectively: fossil fuels account for 90 % of the electricity produced in Morocco (IEA, 2014). Based on reference data, direct and indirect $CO_2$ emission is calculated as shown in equations (6-7).

$$DCO_2 = \sum_i^n CFF\_F_i \times FC_i \quad\quad (6)$$

$$InDCO_2 = \sum_j^n CFP\_E_j \times ELC_j \quad\quad (7)$$

where $DCO_2$ (ton) is direct $CO_2$ emissions and $InDCO_2$ (ton) is indirect $CO_2$ emissions. $CFF\_F_i$ (ton L$^{-1}$) is $CO_2$ footprint by burning fuel, $FC_i$ (ton L$^{-1}$) is fuel consumption by machine excluding fuel use for electricity generation, and i is the types of fuels such as diesel or gasoline. $CFP\_E_j$ (ton MWh$^{-1}$) is $CO_2$ footprint by generating electricity, $ELC_j$ (MWh) is electricity consumption, and j is the source of electricity generation such as coal, petroleum, natural gas, solar, wind, and hydropower.

**Table 2** $CO_2$ emission by burning fuels and generating electricity

| CO₂ emission by burning fuel | | CO₂ emission by generating electricity | | | |
|---|---|---|---|---|---|
| Sources | CO₂ emission[1] (kg of CO₂ L$^{-1}$) | Sources | CO₂ emission by sources[1] (ton of CO₂ $10^{-6}$ kWh) | Proportion of sources in Morocco[2] (%) | CO₂ emission (ton of CO₂ $10^{-6}$ kWh) |
| Gasoline | 2.59 | Coal | 1,026 | 43.4% | |
| Diesel | 2.96 | Petroleum | 1,026 | 25.3% | |
| | | Natural gas | 504 | 22.7% | 820.9 |
| | | Hydroelectricity | 19.7 | 6.9% | |
| | | Renewables | 15.8 | 1.7% | |

[1] U.S. Energy Information Administration (USEIA) [2] International Energy Agency (IEA)

**2.3 Agricultural water requirement for food production**

In this study, "water for food" indicates water withdrawn for crop production, generally irrigation. CROPWAT 8.0 is a decision support tool developed by the Land and Water Development Division of FAO (Smith, 1992), and used to calculate the evapotranspiration, crop water requirements, and irrigation requirements of four crops grown in the region. The climate data (temperature, precipitation, humidity, wind speed, and hours of sunshine) were taken from the climatic database CLIMWAT 2.0, which offers observed agro-climatic data from 5000 stations worldwide and provides long-term, monthly mean values of climatic parameters. The compiled data of CLIMWAT 2.0 generally includes the period 1971-2000 (when this data was not available, series ending after 1975 that include at least 15 years of data were used). Table 3 showed the average climate data in Khouribga area provided from CLIMWAT 2.0.

CROPWAT 8.0 was used to calculate crop water and irrigation requirements based on soil, climate, and crop data. The calculation procedures used in CROPWAT 8.0 are based on the FAO publication, Irrigation and Drainage Series: No. 44 and 56, Crop Evapotranspiration-Guidelines for computing crop water requirements (Allen et al, 1998; Smith, 1992). Irrigation water requirements were calculated by estimating crop evapotranspiration (ETc), determined by multiplying the crop coefficient (Kc) by the reference crop evapotranspiration (ETo), (see Eq. (8)). ETo is calculated using the FAO Penman–Monteith method, as recommended by FAO and described in Eq. (9) (Allen et al., 1998).

$$ET_c = ET_0 \times K_c \tag{8}$$

$$ET_0 = \{0.408\Delta(R_n - G) + \gamma(900/T + 273)u_2(e_s - e_a)\}/\{\Delta + \gamma(1 + 0.34u_2)\} \tag{9}$$

where $ET_0$ is the reference crop evapotranspiration (mm day$^{-1}$); $ET_c$ is the crop evapotranspiration (mm d$^{-1}$); $K_c$ is the crop coefficient; $\Delta$ is the slope of the saturated vapor pressure/temperature curve (kPa °C$^{-1}$); $\gamma$ is the psychrometric constant (kPa °C$^{-1}$); $u_2$ the wind speed at 2 m height (m s$^{-1}$); $R_n$ is the total net radiation at crop surface (MJ m$^{-2}$ day$^{-1}$); G is the soil heat flux density (MJ m$^{-2}$ day$^{-1}$); T is the mean daily air temperature at 2 m height (°C); $e_s$ is the saturation vapor pressure (kPa); and $e_a$ is the actual vapor pressure (kPa). Crop coefficients are influenced by cultivation, local climatic conditions, and seasonal differences in crop growth patterns (Kuo et al., 2006). FAO provides crop coefficients for each stage. The values for Mediterranean countries were applied, as shown in Table 4 (Allen et al., 1998). Irrigation water requirement was calculated by ETc and effective precipitation, as shown in Eq. (10). The effective precipitation indicated the precipitation except for runoff, and was calculated using the USDA Soil Conservation Service method (Eq. 11) (Smith, 1992).

$$IRReq = ET_c - P_{eff} \tag{10}$$

$$P_{eff} = P_{tot}(125 - 0.2\,P_{tot})/125 \qquad \text{for } P_{tot} < 250\ mm \tag{11}$$

$$P_{eff} = 125 + 0.1\,P_{tot} \qquad \text{for } P_{tot} > 250\ mm$$

where $IRReq$ is irrigation water requirement, ETc is the crop evapotranspiration, $P_{eff}$ is effective precipitation, and $P_{tot}$ is total precipitation.

**Table 3** Climate information in Khouribga

| Month | Precipitation (mm m$^{-1}$) | Temperature min. (°C). | Temperature max. (°C) | Relative humidity (%) | Sunshine hours (h d$^{-1}$) |
|---|---|---|---|---|---|
| Jan | 56 | 3.8 | 17.3 | 72 | 5.6 |
| Feb | 65 | 5 | 19 | 76 | 5.7 |
| Mar | 94 | 7.2 | 21.8 | 69 | 6.4 |
| Apr | 70 | 9.5 | 25.3 | 67 | 7.4 |
| May | 32 | 12.5 | 29.3 | 55 | 8.8 |
| Jun | 9 | 16.6 | 34.5 | 48 | 9.8 |
| Jul | 2 | 19.8 | 39.7 | 39 | 10.9 |
| Aug | 7 | 20 | 39.6 | 37 | 10.3 |
| Sep | 12 | 17.5 | 34.5 | 47 | 9.1 |
| Oct | 27 | 13.5 | 29 | 58 | 7.6 |
| Nov | 71 | 8.8 | 22 | 70 | 5.2 |
| Dec | 81 | 5.1 | 18.6 | 71 | 5.5 |

**Table 4** Crop planting and harvesting seasons, stage length and crop coefficients

| Crop | Planting season | Harvesting season | Stage length (Days) | | | | | Crop coefficients | | |
|---|---|---|---|---|---|---|---|---|---|---|
| | | | Init. | Dev. | Mid | Late | Total | Kc init | Kc mid | Kc end |
| Olives | March | November* | 30 | 90 | 60 | 90 | 270 | 0.65 | 0.7 | 0.7 |
| Wheat | November | June* | 30 | 140 | 40 | 30 | 240 | 0.7 | 1.15 | 0.25 |
| Barley | March | July | 20 | 25 | 60 | 30 | 135 | 0.3 | 1.15 | 0.25 |
| Potato | Jan | April | 25 | 30 | 30 | 30 | 115 | 0.5 | 1.15 | 0.75 |

\* Next year

**3 Results and discussion**
**3.1 Application of scenarios**
Increasing the exportable phosphate products and changing the transportation system from train to pipeline are considered top
priorities for OCP group. Therefore, we assessed the impact of increased production by applying the scenarios (Table 5). Until
recently, dried phosphate was transported by train from mining to manufacturing site, but, in the near future OCP group will
use only pipeline transport. The change of from train to pipeline can affect not only direct energy or water consumption by
transportation system but also that of the total supply chain in the mining site. Consequently, the production processes for
slurry and for rock consume different quantities of water and energy, so that the mode of transport also becomes a scenario to
allow quantification of their respective water and energy requirements.
Therefore, we applied the scenario about transportation system which indicates the only usage of pipeline. Table 4 showed the
scenarios combining production and transportation. The first two scenarios are related to the 'business as usual (BAU)'
scenario for production in 2015 but changing the transportation system from Khouribga to the terminal station at Jorf Lasfar.
The other scenarios are related to the increase in the production.

**Table 5** Scenarios through combination of production and transportation system

| Scenario | Phosphate production | Transportation of phosphate products | |
|---|---|---|---|
| | | by pipeline | by train |
| BAU | Production in 2015 | 40 % of total phosphate | 60 % of total phosphate |
| Scenario 01 | | 100% of total phosphate | None |
| Scenario 02 | 50% increase of phosphate export | 40 % of total phosphate | 60 % of total phosphate |
| Scenario 03 | | 100% of total phosphate | None |


**3.2 Quantification of water and energy consumption and $CO_2$ emission by production and transport of phosphate**
To quantify the water, energy, and $CO_2$ emissions, water and energy footprints of each process in each mining site were
analysed based on survey data. For example, the adaptation process is essential for pipeline transportation and large amounts
water are needed in comparison to other processes, thus the relationship between the amounts of phosphate and water used in
adaptation process were analysed (Figure 4 (a)). In addition, energy footprint includes electricity and fuel consumption;
analysed through the linear relationship (Figure 4 (b)).
Production and transport scenarios were applied and quantified for water, energy, and $CO_2$ emissions in each scenario (Table
6). In the mining area, 20.1 million tons of raw phosphate were produced in the 2015, the "business as usual" (BAU) scenario.
40% of the production was in the form of slurry and transported by pipeline; 60% was in the form of rock and transported by
train. Scenario BAU indicates that 15.84 million m³ of water were used in all processing (both rock and slurry). Additional
fresh water was transported through the pipeline to maintain slurry consistency in the system. For the BAU scenario, 3.85
million m³ of fresh water were transported by pipeline to the industrial area. Scenario 1 (all raw phosphate transported by
pipeline) increases the total water used to 32.14 million m³ (103% increase over BAU). Fresh water is also used to maintain
the good operation of the pipeline, but with the increase in slurry transported by pipeline, the quantity of 'maintenance' fresh
water decreased from 3.85 to 2.47 million m³, leading to a smaller total consumption of fresh water: a 76% increase was shown
for total water consumption (for both processing and transport by pipeline).
Using only the pipeline for transport requires an additional 131,832 MWh in electricity for the flotation and adaptation
processes used to produce slurry (31% increase in comparison to BAU). However, the consumption of fuel significantly
decreases as there is no need to dry phosphate rock. This results in a nearly 86% decrease in fuel consumption over the BAU
scenario and a fuels savings of 176.4 million litre, which translates into a 40% decrease in $CO_2$ emission in Scenario 1. In
Scenario 2, there was a 50% increase of raw phosphate export over the BAU scenario, with transport the same as in BAU.
Total water consumed, including fresh water transported through the pipeline, increased by 16% over BAU, energy
consumption increased by 46 %, and $CO_2$ emission increased by 39%. Scenarios 3 and 4 represent a 50% increase in phosphate
exports, thus target production was set at 2.45 million tons month$^{-1}$ for raw phosphate, in total, 29.3 million tons yr$^{-1}$. Scenario
3 indicates a total water consumption increase to 46.6 million m³ (137% over BAU), and electricity consumption increase of
75%. However, transport by pipeline also led to an 80% decrease in fuel consumption (compared to BAU), and consequent
18% decrease in $CO_2$ emissions.
In summary, the comparison between BAU and scenario 1 showed the trade-off between water and energy by the change in
transportation method. Pipeline transportation can save energy use and reduce $CO_2$ emission, but more water is required due
to additional processes, such as adaptation and water used to operate the pipeline. However, since the water used to operate
the pipeline is actually transported to Jorf Lasfar and re-used in fertilizer factories, it could be considered non-consumptive
water in terms of the supply chain integrating Khouribga and Jorf Lasfar, even though it is still real water withdrawn form
Khouribga.

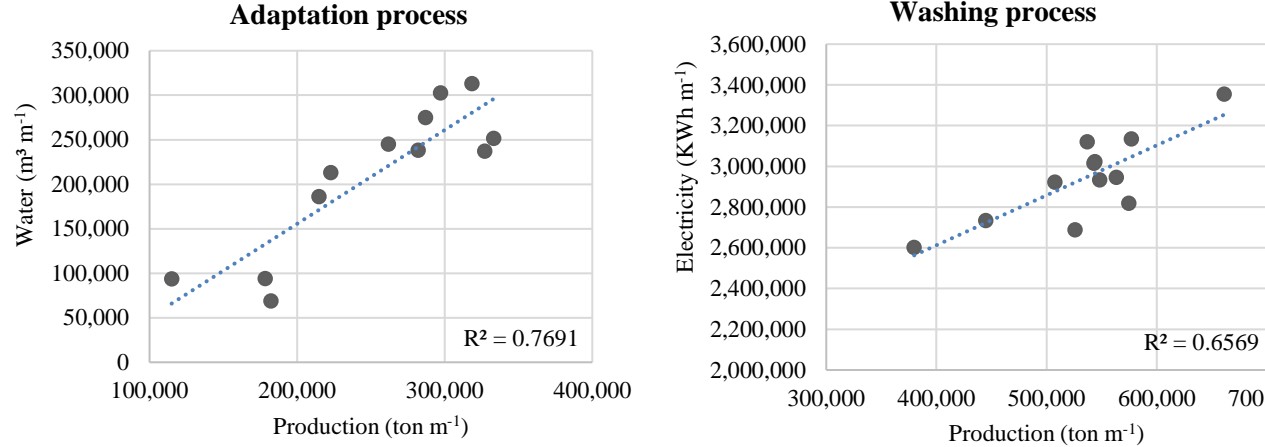

(a) Water footprint in an adaptation process in MEA       (b) Energy footprint in a washing process in MEA

**Figure 4.** Water and energy footprints in MEA based on the BAU data-base
**Table 6** Water and energy use, and $CO_2$ emission by scenario of phosphate production and transport

| Scenario (Transportation) | Water ($10^6$ m³ yr$^{-1}$) | | Energy | | $CO_2$ emission ($10^6$ ton yr$^{-1}$) | |
|---|---|---|---|---|---|---|
| | Water used in processes | Transported water | Electricity (MWh yr$^{-1}$) | Fuels ($10^6$ L yr$^{-1}$) | Direct | Indirect |
| BAU | 15.84 | 3.85 | 424,512 | 204.0 | 0.53 | 0.35 |
| Scenario 1 | 32.14 | 2.47 | 556,344 | 27.6 | 0.07 | 0.46 |
| Scenario 2 | 19.35 | 3.45 | 551,495 | 297.9 | 0.77 | 0.45 |
| Scenario 3 | 45.15 | 1.45 | 743,928 | 40.5 | 0.11 | 0.61 |

**3.2 Assessment of the impacts of water allocation and treated water use in the industrial and the agricultural areas**

The main challenge of the mining area is sustainable water allocation for both the phosphate industry and irrigated agricultural areas. Thus, production targets were established for both phosphates and crops, and scenarios evaluated using the WEF-P Tool. Target crop production rates for Morocco's primary food crops (wheat, olive, barley, and potato), were set as 0.1 % of national production. Table 7 shows that 25.07 million m³ yr$^{-1}$ of irrigation water is required to produce 5,722 ha of crops. In the case of wheat, irrigation requirements were calculated at 313.7 mm yr$^{-1}$, equivalent to 9.08 million m³ to produce 0.1 % of national production annually.

The main water resource for the mining area is the Ait Messaoud dam. Water allocations from this source affect both phosphate and agricultural areas. Water used for phosphate production increases when the pipeline is used to transport slurry (verses dry rocks transported by train). The impact of water allocation under only the pipeline is calculated using various scenarios for water allocation (Table 8) and the treated wastewater from urban area was considered as a water resource for both the phosphate industry and agriculture.

In the "Alloc 1" scenario, supply capacity from the dam was set at 80% for the phosphate industry and 20% for the agricultural area. The waste-water treatment plant operates in the mining area. For scenario Alloc 1, all treated water was assigned to the phosphate industry. The "Alloc 2" scenario focuses on the importance of water for agriculture and assigns the water equally between the phosphate and agricultural areas. Water supplied from the dam, plus treated water from the plant may be insufficient for both industries. To address this issue, treated water and ground water were considered as supplementary water sources and a treated water quantity of 2.5 million m³ yr$^{-1}$ (50 % of current operation) was assigned to the two industries.

When water resources were allocated according to the Alloc 1 scenario (80% of surface water and 100% of treated water allocated to phosphate mining area), 9.68 million m³ additional water are required for agriculture (Table 9). When water is allocated equally between the two industries (Alloc 2 scenario), there is a shortfall of 9.61 million m³ in the phosphate industry, but of only 70,000 m³ for agricultural irrigation. In the case of a 50% increase in phosphate production over BAU and using the pipeline as the only mode of transport, the Alloc 1 scenario indicates intensive water supply to phosphate mining area rather than to agricultural area and causes an annual shortage of 5.59 and 16.07 million m³ water in the phosphate mining and the agricultural area, respectively. To address this shortage, 2.5 million m³ of treated water could be supplied in addition to 19.16 million m³ of ground water.

Additionally, electricity is required to pump ground water and treat wastewater. Thus, the source of water may also affect electricity consumption. Goldstein and Smith (2002) noted that 0.198 kWh is required to supply 1 cubic meter of ground water, and the least electricity required to supplying surface water is 0.079 kWh m$^{-3}$. Therefore, a 50% increase over BAU is accompanied by 3,794 MWh yr$^{-1}$ electrical consumption for pumping ground water (Table 9). Increasing the use of treated water releases the demand for ground water use, but the costs of building and operating the infrastructure and treatment facility must be considered. In this study, the capacity of treated water was set at 2.5 million m³ yr$^{-1}$, and ground water requirements were changeable only as scenarios of water allocation.

**Table 7** Water use for crop production under Moroccan condition

| Crops | Production* | Productivity | Area | Irrigation water requirement | |
|---|---|---|---|---|---|
| | ton | ton ha$^{-1}$ | ha | mm yr$^{-1}$ | 10⁶ m³ yr$^{-1}$ |
| Olive | 834 | 1.28 | 652 | 622.4 | 4.06 |
| Wheat | 4054 | 1.40 | 2895 | 313.7 | 9.08 |
| Barely | 1840 | 0.87 | 2115 | 562.7 | 11.90 |
| Potato | 1417 | 23.43 | 60 | 48.9 | 0.03 |
| Total | 8146 | | 5722 | 1547.7 | 25.07 |

*Crop production is the amount of 0.1% of national production in Morocco

405

**Table 8** Water allocation and treated water use scenarios

| Scenario | Sources | Capacity | Assignment of capacity | |
| --- | --- | --- | --- | --- |
| | | $10^6$ m³ yr$^{-1}$ | Phosphate | Agriculture |
| Alloc 1 | Dam | 45.0 | 80% | 20% |
| | Treated water | 5.0 | 100% | 0% |
| Alloc 2 | Dam | 45.0 | 50% | 50% |
| | Treated water | 5.0 | 50% | 50% |
| Treated water supply | | 2.5 | 1st priority | 2nd priority |

407

**Table 9** Additional water and energy for solving water shortage by sceansrios of phosphate production

| Production (Only pipeline) | Water allocation | Water shortage | | Additional water supply | | Energy use for water supply | |
| --- | --- | --- | --- | --- | --- | --- | --- |
| | | Phosphate | Agriculture | Treated water | Ground water | Treated water | Ground water |
| | | $10^6$ m³ yr$^{-1}$ | $10^6$ m³ yr$^{-1}$ | $10^6$ m³ yr$^{-1}$ | $10^6$ m³ yr$^{-1}$ | MWh yr$^{-1}$ | MWh yr$^{-1}$ |
| Production as BAU | Alloc 1 | 0.00 | 9.68 | 2.50 | 7.18 | 1653 | 1421 |
| | Alloc 2 | 9.61 | 0.07 | | | | |
| 50% Increase over BAU | Alloc 1 | 5.59 | 16.07 | 2.50 | 19.16 | 1653 | 3794 |
| | Alloc 2 | 21.59 | 0.07 | | | | |

## 3.3 Assessment of the impact of dynamic management of phosphate production on ground water and energy savings

Water resource availability and water requirements for crop production are seasonal. Rainfall in June and July is less than 10 mm month$^{-1}$ and irrigation water requirements exceed 80 mm month$^{-1}$ (Figure 5). Thus, there is water scarcity in the agriculture area during June and July. Given that water resources are shared between the phosphate industry and the agriculture industry, static production of phosphate could accelerate water shortage for agriculture. Dynamic production of phosphate is a scenario with greater agricultural production during non-irrigation seasons and less production during irrigation seasons. Using the dynamic phosphate production scenario, the monthly production of phosphate decreases from 1.68 to 0.91 million tons month$^{-1}$ between May and October, representing a 50% decrease in raw phosphate export compared to BAU scenario. Between November and April, phosphate production increases to 2.45 million tons month$^{-1}$, representing a 50% increase in raw phosphate export in compared to BAU scenario.

419

420

421

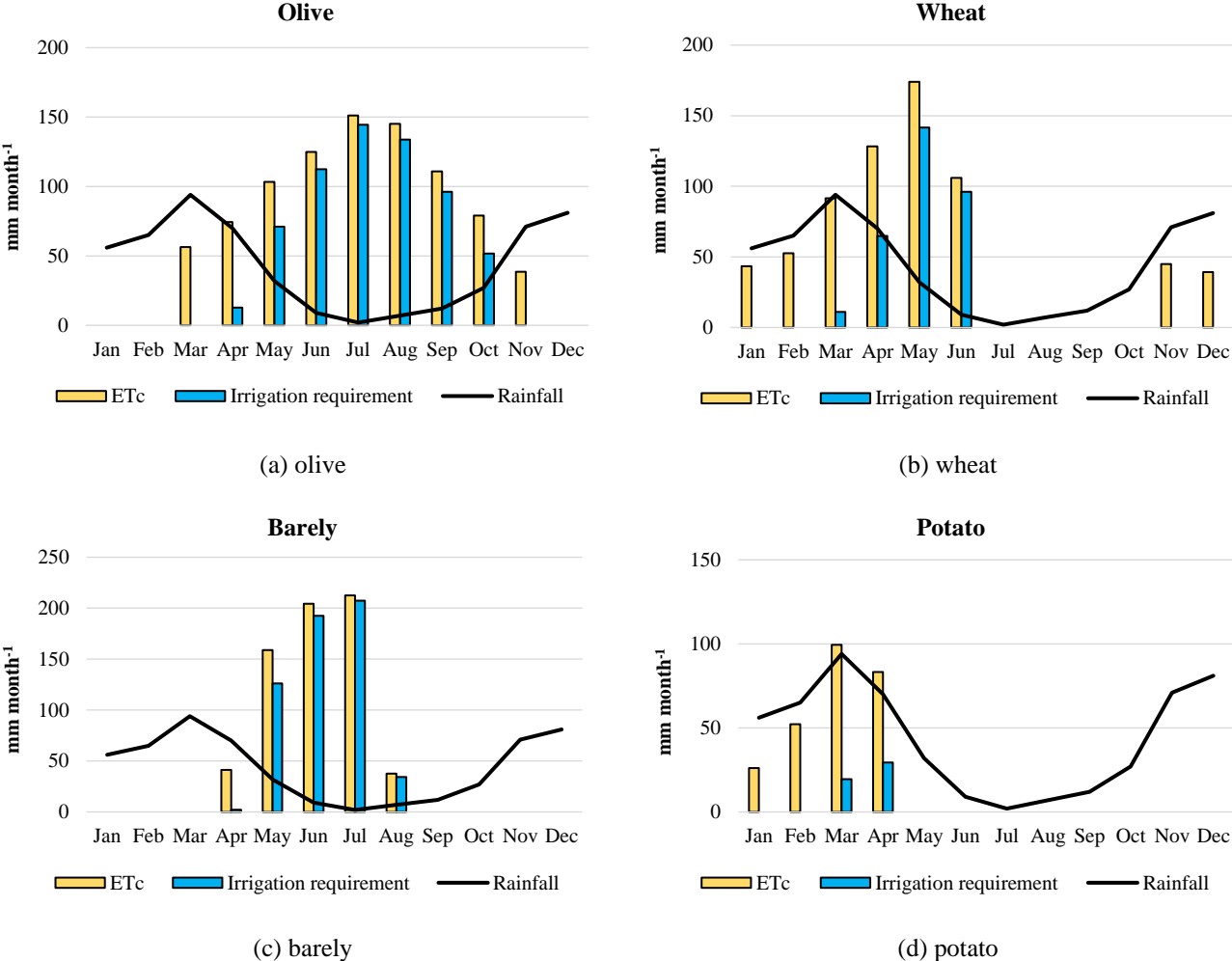

(a) olive

(b) wheat

(c) barely

(d) potato

**Figure 5** Monthly irrigation water requirement and rainfall in Khouribga

Water availability and irrigation water requirements differ seasonally: dynamic monthly production of phosphate can contribute to sustainable water management. The effect of dynamic phosphate production on water supply becomes obvious when the pipeline is the only mode of transport: slurries are more water intensive than rock. Under static phosphate production, the monthly demand for water from the dam in January and February was about 2.5 million m³, increasing to 7.0 million m³ month[-1] in June (Figure 6). Nevertheless, dynamic phosphate production decreases the water demanded during the water scarce season. Moreover, the lack of water supply is covered by ground water: dynamic production uses less ground water than static production (Figure 7). During the water scarce season (May to July), total ground water used is 5.77 million m³ in static phosphate production. This decreases by 10% in dynamic production, potentially saving 0.58 million m³ of ground water during the water scarce season. Groundwater resources constitute an important aspect of the national hydraulic heritage and represent the only water resource in this hyper arid climate (Tale, 2006). Thus, dynamic phosphate production carries positive impact on sustainable water management and water conservation.

Dynamic phosphate production also contributes to electricity savings: supplying water from the dam, ground, or treatment require electricity for pumping, transporting, and treating (Figure 8). Total electricity consumed in supplying water to the phosphate and agriculture industries was 9,971 MWh yr[-1] under the static production scenario (phosphate slurries, no rocks). This number decreased to 9,828 MWh yr[-1] when phosphate slurries were produced dynamically. About 143 MWh electricity can be saved annually, accompanied by a reduction of 117 tons of $CO_2$ emission.

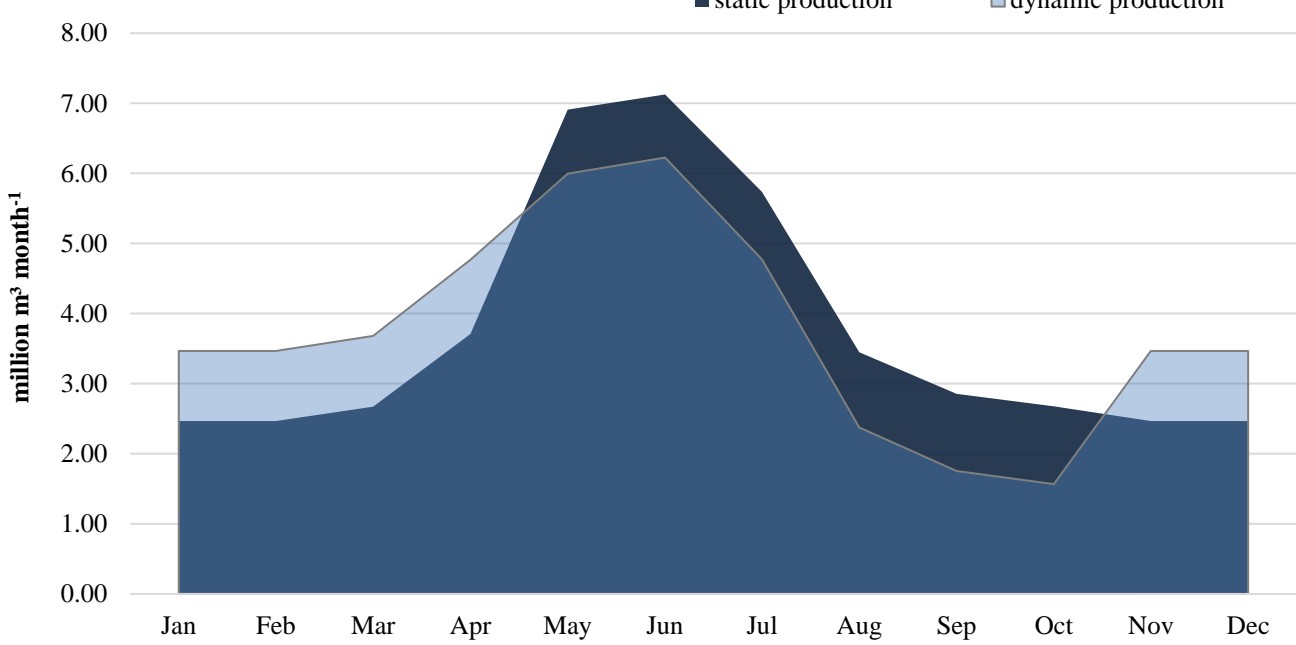

**Figure 6** Monthly water supply from Aït Messaoud Dam

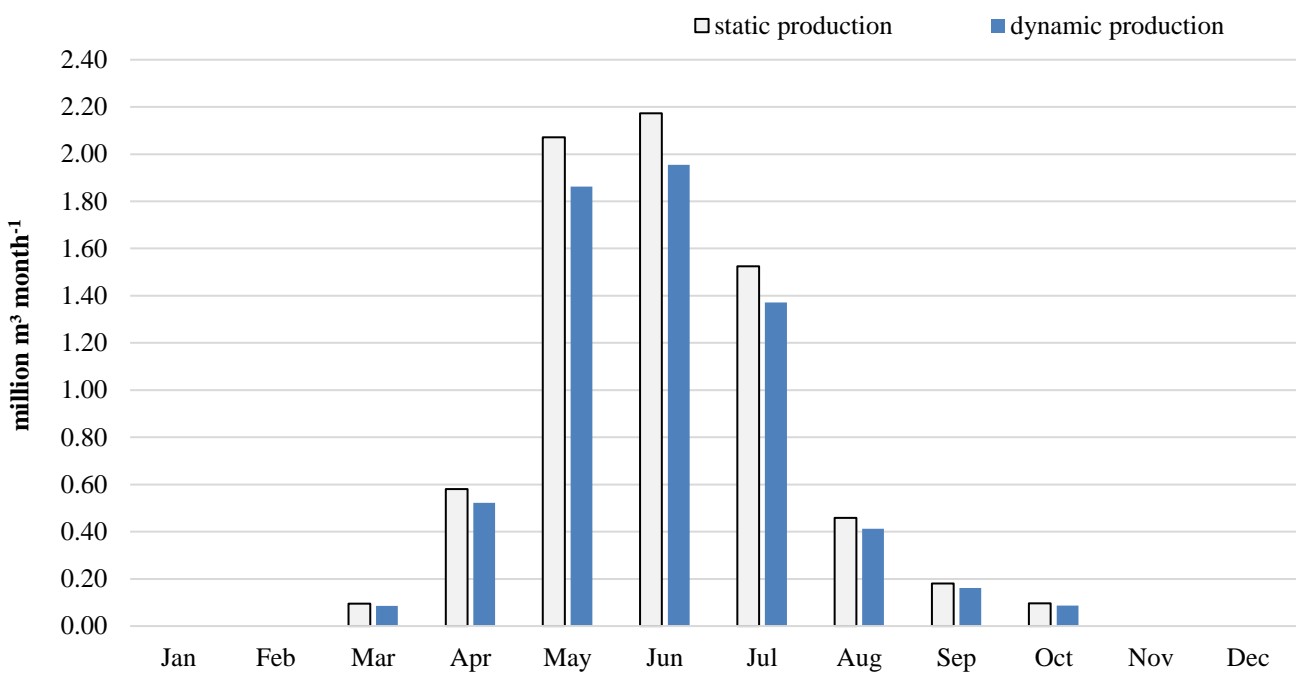

**Figure 7** Monthly ground water use by static and dynamic production of phosphate slurries transported by pipeline


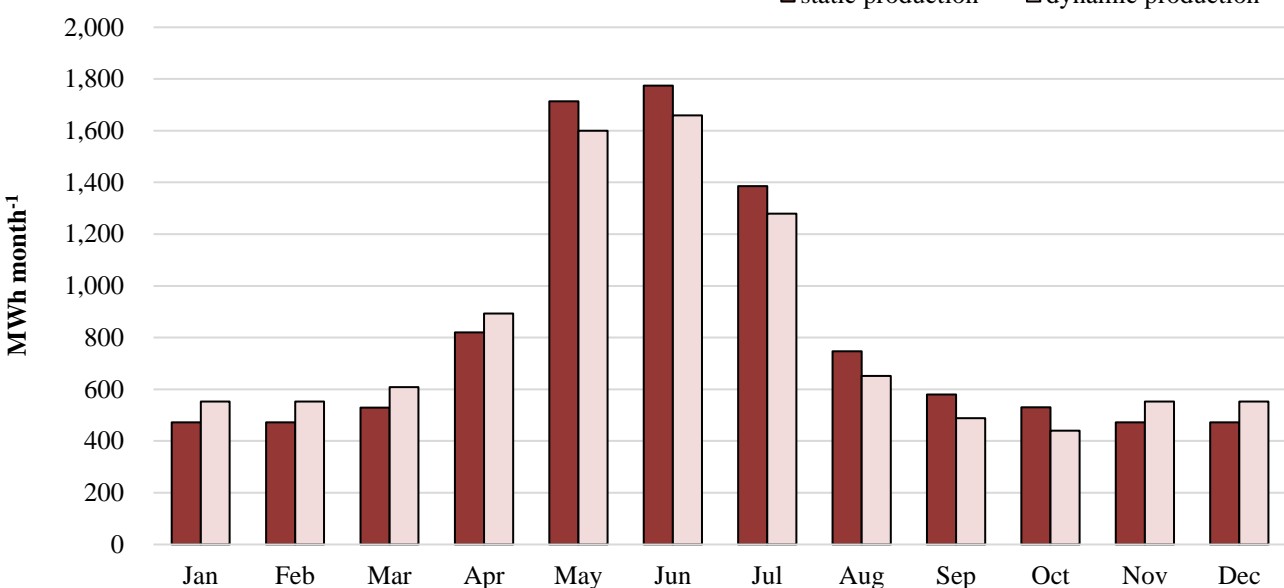

**Figure 8** Monthly electricity consumption for supplying water by static and dynamic production of phosphate slurries
transported by pipeline
**4 Conclusions**
As Morocco continues to work toward meeting its projected phosphate production goals, it is important to assess and quantify
the potential resource competition between the growing municipal and agricultural sectors. Sustainable resource management
strives for symbiosis between the phosphate industry and other sectors, and endeavours to create synergy through multiple
strategies. The WEF-P Tool integrates water-energy-food management and supply chain management for phosphate
production, considering the trade-offs between water, energy, and food, as well as a systematic analysis based on the total
supply chain management of phosphate production. In other words, the WEF-P Tool offers a decision support system to
provide quantifiable trade-off analyses for management decisions such as increasing production, transportation systems, and
water allocation. The developed WEF-P Tool enables users to:
• understand and identify the associated footprints of the primary functional production processes and existing flows
in production lines;
• identify the main sources of data to be gathered and fed into the model on a specific temporal basis;
• identify the techniques employed to conserve or produce water and energy and minimize the impacts of phosphate
production;
• form a translational platform between sectors and stakeholders to evaluate proposed scenarios and their associated
resource requirements
As phosphate mining increases, options that contribute to reducing water and energy stress include increased reliance on
transport by pipeline and dynamic management of phosphate production. This tool assesses the impacts of various production
pathways, including specific process decisions throughout the phosphate supply chain, such as the choices for transport by
pipeline or train and the impacts on regional water and energy use. For example, transport by pipeline instead of train can
contribute to energy savings due to the elimination of the phosphate drying process (a main consumer of fuels). At the same
time, the slurry adaptation processes are main consumers of water, though because the pipeline also transfers fresh water to
Jorf Lasfar where the fertilizers are produced, the water embedded in slurry is a main water resource for Jorf Lasfar. Previously,
the main water resource in Jorf Lasfar was desalinated water, which consumes energy in desalination. Transport by pipeline

contributes to a savings of desalinated water and energy for desalinating. The dynamic management scenario is assessed for its impacts on regional water and energy savings: dynamic management of phosphate production indicates different production quantities during irrigated and non-irrigated seasons. Less phosphate production during irrigation season can contribute more surface water for agricultural use and is accompanied by a savings of ground water and the energy required to pump ground water.

Further consideration of the economics of the phosphate operation is needed: static production may bring stability to operations (meeting local and export demand), but there are benefits from dynamic production that can be attributed to reduced competition with other water consuming sectors. Additional variables, such as facility operation, labour, economic cost/benefit of static and dynamic production, etc., should be quantified and included for additional trade-off assessments. Quantification of water and energy for phosphate production is strongly dependent on the relationship between production and resource consumption: this can change in future scenarios. Proper water availability for the right place and time in a changing climate requires analysis of complex scientific, technical, socio-economical, regulatory, and political issues.

Beyond the limitations, the deliverables from this study include a conceptual and analytical model of the phosphate supply chain in Morocco, the WEF-P Tool. The Tool can assess the various scenarios to offer an effective means of ensuring the sustainable management of limited resources to both agricultural area and phosphate industry. It quantifies the products (phosphate) and resource footprints (water, energy) across the supply chain; identifies the interlinkages between water and energy in phosphate production and transport, and establishes reference values for comparison of outcomes and performance. The WEF-P Tool enables the user to evaluate trade-offs between water resource allocations and the impact of the Moroccan phosphate industry with agricultural water use.

## Author contribution

Sang-Hyun Lee, Amjad T. Assi, and Rabi H. Mohtar conceived and designed the research; Sang-Hyun Lee and Amjad T. Assi analysed the data; Sang-Hyun Lee, Bassel T. Daher, and Fatima E. Mengoub contributed analysis tools; Sang-Hyun Lee and Amjad T. Assi wrote the paper.

## Competing Interests

The authors declare that there are no conflicts of interest regarding this publication.

## Acknowledgments

This research was funded by the OCP Policy Center and the OCP foundation and also supported by the Japan Science and Technology Agency as part of the Belmont Forum. The authors thank the OCP Policy Center for arranging meetings with their engineers and managers in the mining and production areas for data collection. The authors also express their appreciation to Mary Schweitzer of the WEF Nexus Research Group at Texas A&M University for her editorial assistance and contributions.

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

# A Water-Energy-Food Nexus Approach for Conducting Trade-off Analysis: Morocco's Phosphate Industry in the Khouribga Region

Sang-Hyun Lee[a], Amjad T. Assi[b], Bassel T. Daher[b], Fatima E. Mengoub[c,d], Rabi H. Mohtar[b,e,f]*

[a] Research Institute for Humanity and Nature, Kyoto, Japan [sanghyunsnu@gmail.com].
[b] Department of Biological and Agricultural Engineering, Texas A&M University, College Station, TX 77843-2117, USA [amjad.assi@tamu.edu; basseldaher@gmail.com]
[c] Policy Center for the New South, Suncity Complex, Building C, Angle of Addolb Boulevard and Albortokal Street, Rabat, Morocco [f.mengoub@policycenter.ma]
[d] Mohammed VI Polytechnic University, Lot 6620, Hay Moulay Rachid 43150, Benguerir, Morocco
[e] Zachry Department of Civil Engineering, Texas A&M University, College Station, TX 77843-3136, USA.
[f] Department of Agricultural and Food Sciences, American University of Beirut, Beirut, Lebanon.

* Correspondence: Rabi H. Mohtar, Professor and Dean, Faculty of Agricultural and Food Sciences, FAFS 101, American University of Beirut, P.O. Box 11-0236, Riad El Solh, Beirut 1107 2020, Lebanon.  Tel.: +961-1-350000 ext: 4400, Fax: +961-1-744460, [mohtar@aub.edu.lb; mohtar@tamu.edu].

## Abstract

The study's objective was to develop and use the Water-Energy-Food Nexus-Phosphate (WEF-P) Tool to evaluate~~d~~ the impact of Morocco's ~~impact of the~~ phosphate industry on ~~regional~~ water, energy, and food sectors of ~~in~~ Khouribga, the representative phosphate mining ~~area~~region of Morocco~~, using the developed WEF-P Tool~~. The developed ~~The aim of the study is to apply the Nexus approach to conduct trade-off analysis between industrial and agricultural areas. A Water Energy Food Nexus-Phosphate (~~ (WEF-P~~)~~ Tool ~~was developed~~enabled a trade-off analysis~~,~~ based on integrating the supply chain processes, transportation, and water-energy footprints of the region. Field data from the mining to transportation processes were collected and applied to possible supply chain scenarios in accordance with the type of product (phosphate rock and slurry). The ~~The study evaluated the impact of the phosphate industry on regional water, energy, and food in Khouribga, the representative phosphate mining area of Morocco, using the developed WEF-P Tool. To address the~~ potential impacts of the scenarios were considered in terms of ~~on~~ the water supply in the agricultural areas~~. , the field data of processes (from mining to transportation) were collected and applied to possible supply chain scenarios according to type of product (phosphate rock and slurry). A~~The analysis of the positive impacts of dynamic management suggests that seasonal management of phosphate production (~~allows~~ less ~~phosphate production~~ during the ~~irrigation~~ irrigated season~~. (thereby increasing water available for agriculture) and greater~~more ~~phosphate production~~ during wetter/rainier seasons) is more effective. ~~(when less water is demanded for agricultural production).~~Additionally, while the transport of raw phosphate slurry through a pipeline increase~~d~~s the total water required ~~ to 34.6 million m³, an increase of 76% over the "business as usual scenario" (BAU), ~~but~~it also result~~ed s~~in an energy savings of nearly 80% over BAU~~: , as~~slurry transport requires only 40.5 million litres  of fossil fuel, ~~compared~~instead of the ~~with~~ 204 million litres required to transport ~~for~~rock~~s transport~~. During the dry or "water scarce" irrigated season (May to July)~~, when irrigation is needed,~~ total ground water use decreased from 5.8 to 5.2 million m³. Dynamic management of the phosphate industry can also save 143 MWh of electricity annually and ~~is~~ brings ~~accompanied by~~ a reduction of 117 tons of $CO_2$ emissions. ~~In a changing climate, m~~Making water available at the correct season and location requires analysis of complex scientific, technical, socio-economical, regulatory, and political issues. The WEF-P Tool can assist ~~in~~ by assessing user-created scenarios, thus ~~becomes~~ it is an effective management-decision aid for ~~effectively~~ ensuring more sustainable ~~management~~ use of limited resources and increased reliability of water resources for both agricultural and industrial use. This study on the application of WEF Nexus to the Phosphate industry ~~can~~ offers~~be~~ a roadmap for other industrial application ~~where~~ for which trade-offs between the primary resources must be considered~~exist~~.

*Keywords*: Phosphate, Water-Energy-Food Nexus, Morocco, WEF-P Tool

# 1 Introduction

Nexus thinking emerged from the understanding that natural resource availability can limit and is limited by economic growth and other goals associated with human well-being (Hoff, 2011; Keulertz, 2016). The innovative aspect of nexus thinking is a more balanced view of the issues linking resources (Al-Saidi and Elagib, 2017). Thus, nexus frameworks identify key issues in food, water, and energy securities through a lens of sustainability, to predict and protect against future risks and resource insecurities (Biggs et al., 2015). The 2015 World Economic Forum identified water, food, and energy shocks as primary future risks, calling for increased efficiency in water use across all sectors and the implementation of integrated water resources management. Various conceptual frameworks relating to the nexus approach were developed: the FAO (2014) emphasized the role of the nexus in food security; the International Renewable Energy Association (IRENA, 2014) applied the nexus approach in transforming conventional energy systems to renewable systems.

The demand for water, energy, and food, is expected to increase due to drivers such as population growth, economic development, urbanisation, and changing consumer habits (Terrapon-Pfaff et al., 2018). The interlinkages across key natural resource sectors and improving their production efficiency offer a win-win strategy for environmental sustainability, whether for current or future generations (Ringler et al., 2013). Accordingly, application of the Water-Energy-Food (WEF) nexus concept or approach is expected to make implementation of the Sustainable Development Goals (SDGs) more efficient and robust (Brandi et al., 2014; Yumkella and Yillia, 2015). The SDGs are classic examples of the necessity to acknowledge multidimensional, nexus interlinkages and trade-offs, particularly a. As governments are challenged to maximize benefits and invest limited resources. Infrastructure and capital are needed to achieve national SDG targets and the concept is now used to highlight interdependencies between resources and the need for integrated, sustainable governance and management of those resources (Pahl-Wostl, 2019).

The debate surrounding effectively addressing water and food security challenges stems from questions about whether the water-food crisis is due to a poor understanding of these resources or to their improper management (Mohtar et al., 2015). One long-standing challenge to water management lies in the lack of integration among the multiple sectors that interact with the water sectors across geographical areas or within large, transboundary, basins (Mohtar and Lawford, 2016). Projections about availability and quality of water, food, energy, or soil resources are often alarming. A fundamental shift is needed away from traditional 'silo' approaches and toward more integrative, systems approaches (Daher and Mohtar, 2015). Energy and water are crucial for economic growth, especially in industrialized areas (Flörke et al., 2013; Cai et al., 2016), making the rapid increase in demand for these resources a serious issue for both economics and the environment. While technology to reduce industrial demand for water and energy is important, we must also understand the relationship between economic growth, water–energy consumption, the impact of industrial activity on agriculture at the local level. Increase of industrial products can cause steep increases in demand for water and energy, which in turn, leads to issues of downscaling water or energy securities.

The nexus framework is dependent on the stakeholders, system boundary, and analytical tools. In considering the application of the nexus as a platform, an integrated modelling approach is essential. These issues manifest in very different ways across each sector, but their impacts are often closely related in terms of trade-offs. In particular, the sub-nexus needs to be effectively conceptualized and a theoretical sub-nexus developed. Private-sector water, energy, and food supply chain players are the key stakeholders to address current contradictions arising as a consequence of attempts to develop a grand nexus approach (Allan et al., 2015). Accordingly, we must consider the "specialized" nexus of multi-stakeholders, such as agriculture, industry and urban areas, for which water, energy and food are treated as subsystems. Current nexus frameworks often focus on macro-level drivers of resource consumption patterns (Biggs et al., 2015), but major nexus challenges are faced at local levels

(Terrapon-Pfaff et al., 2018). Thus, 'larger scale' extraction and consumption of natural resources may lead to depletion of
natural capital stocks and increased climate risk with no equitable share of the benefits (Hoff, 2011; Rockström et al., 2009).
Al-Saidi and Elagib (2017) showed the importance of exploring driving forces and interactions at different scales in the
conceptual development of the nexus, emphasizing more case-study based recommendations in the reality of institutions,
bureaucracies, and environmental stakeholders.
Morocco's phosphate and agriculture industries offer an example of increasing resource pressures attributable to near- and
medium-term growth across these sectors (Taleb, 2006). A holistic approach that considers the needs of all stakeholders is
necessary eded to resolve the resource allocation pressures. Between 1990 and 2016, Morocco's population grew from 25
million to 35 million people (World Bank, 2019a)(https://data.worldbank.org). Both crop production and total cultivated land
significantly increased since 1971, and half of Morocco's arable land receives less than 350 mm of rainfall annually and nearly
87.3% of Morocco's total water withdrawals are used for agriculture (FAO, 2015) (http://www.fao.rg/ag/aquastat). Per capita
consumption of electric power increased from 358 kWh (1990) to 901 kWh (2014); energy use by oil equivalent per capita
increased from 306 to 553 kg during the same period (World Bank, 2019b)(https://data.worldbank.org). Proper management
of water, energy, and food resources is critical to economic, social, and environmental well-being.
Globally, phosphates lie at the heart of agriculture and soil enhancement. More than 75% of global phosphate reserves,
representing 30% of the global market share, are found in Morocco, positioning that country for a leading role in global food
security (OCP, 2013). Phosphate mining and its chemical processing require considerable water, energy, land, and other
resource inputs. Morocco uses recycling and reverse osmosis desalination to relieve some of the pressure on its fresh water
resources and help secure the water necessary eded for phosphate production processes (OCP, 2016b) and to relieve some of
the pressure on its fresh water resources. Each water source carries a distinct energy tag that must be accounted for, especially
in a country that imports nearly 90% of the its consumed energy it consumes (World Bank, 2019c). Water, energy, land, and
financial resources are frequently shared between multiple, sectors, especially agriculture (food production) and municipal
(growing urbanization): Morocco is no exception. It is critical that this potential sectoral competition be understood, quantified,
and accounted for when planning for the sustainable progress of all sectors. An integrated approach to resource allocation is
needed tTo minimize inevitable competition for resources:, an integrated approach to resource allocation is needed, one an
approach that quantifies the trade-offs associated with the possible pathways. As Morocco heads toward achieving its
phosphate production goals, its the ability to account for the resources associated with that achievement should be balanced
with the associated (and increasing) agriculture and municipal demand projections: this is key to sustainable resource allocation
(OCP, 2013).
This study adapted the WEF Nexus Tool linking industry and agriculture to to also integrate the supply chain for industrial
products. Using the Tool, we the authors evaluated the impacts of Morocco's phosphate production on the water, energy, and
food resource systems of its mining region , and then addressed the resource elements in in the supply chain management of
phosphate production. Specifically, we they assessed the impacts of phosphate mining and transportation by slurry pipeline on
potential water and energy savings in the mining area. The results suggest the need for a dynamic management of phosphate
production, one that adjusts monthly phosphate production in consideration of its potential impacts on water and energy
management in agricultural areas.
The specific objectives of the study is are to quantify the water, energy, food, energy used by theof phosphate industry in the
Khouribga region of Morocco and to assess the trade-offs of water and resource s allocations between agricultureal and
industryial use.

## 2 Materials and methods

### 2.1 Site description

We contacted the managers and engineers working in the Office Cherifien des Phosphates (OCP) group which is that country's
leading phosphate producer in Morocco, and had a lot of discussion about the site, data, policy, and goals. OCP group accounts
for 3% of the country's gross domestic product and about 20% of national exports in value over the course of the 20th century
(Croset, 2012). The OCP group ran three mining fields: in central Morocco, near the city of Khouribga, and on the Gantour
site. Khouribga, the largest mining area, includes three main sites from which raw phosphate is excavated and transported for
chemical processing and fertilizer production: Sidi Chennane (SC), Merah Lahrach (MEA), and Bani Amir (BA) (Figure 1).
The output in Khouribga is raw phosphate produced as either rock or slurry, the main component of manufactured phosphorous
fertilizers. The transport of the phosphate (rocks and slurries) from Khouribga (mining area) to Jorf Lasfar (industrial
production area) is a primary project in Morocco (OCP, 2016a). The demand for raw phosphate and the production and export
of fertilizer and its products from Jorf Lasfar drive the upstream mining activity of Khouribga. In 2015, approximately 20.1
million tons of raw phosphate were excavated, which was 58 % of total raw phosphate excavated in Morocco in 2018 (OCP,
2020), and transported to Jorf Lasfar; about 40% of this product was transported via pipeline as slurry and the balance via train
as rock.
The pipeline from Khouribga to Jorf Lasfar is 187 km and ensures the continuous transport of phosphate from the Khouribga
to Jorf Lasfar (Figure 1). As the plan was to increase phosphate production and phase out transport by train, tracks were
replaced by pipeline that ensures the continuous flow of raw phosphate from the mining to the industrial area (OCP, 2016a).
The plans impact regional water, energy, and food management: in particular, shifting from train to pipeline requires additional
water to convert dry rock into liquid slurry. Shifting from train to pipeline changes the demand for water and energy resources
at both the mining and the production locations.
In accordance with the "Green Morocco (Plan Maroc Vert)" (Stührenberg, L.2016) and the National Water Plan for Morocco,
the use of surface water as a substitute for groundwater is encouraged: and water withdrawals from the regional 's aquifers
are being phased out since 2010, eventually to be replaced entirely by surface water from , supplied to the mining sites from
the nearby Ait Messaoud dam, which has a capacity of 13.20 million m³. The plan is to allocate 4.5 million m³ yr⁻¹ of water
from the dam to the mining site. Additionally, OCP launched a plan to complete treatment plants for urban wastewater
(capacity 5 million m³ yr⁻¹) to be used for washing phosphate and industrial reuse in the mining area (OCP, 2016b). The
phosphate mining area is encircled by cropland, whose water is also supplied from the dam. In this study, the authors consider
the allocation of treated water to both the phosphate industry and agricultural irrigation (Tian et al, 2018). Both the mining and
the agricultural activities of the region represent growing enterprises that place added pressure on available water resources,
and makeing the sustainable management of the water supply a hotspot to be considered in trade-off analyses.

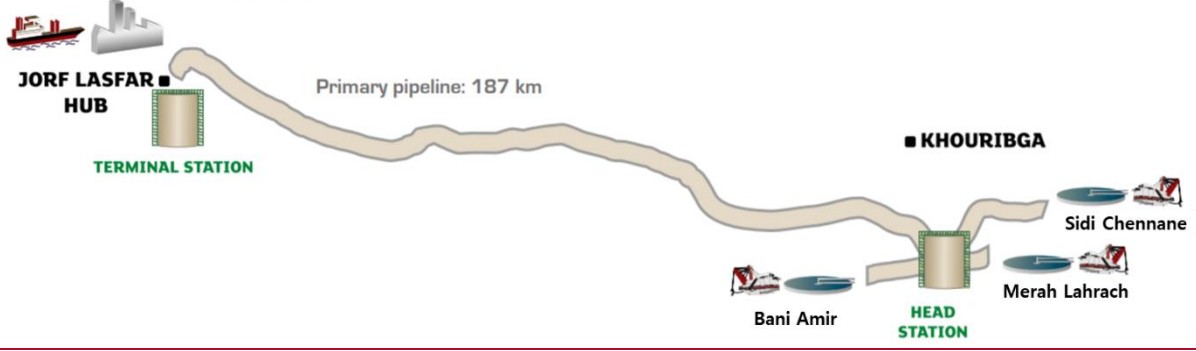

**Figure 1** Study area: including phosphate mining area (Khouribga), fertilizer manufacturing area (Jorf), and transportation
system (slurry pipeline) (http://www.ocpgroup.ma/ocpslurrypipeline/slurry-pipeline)
2.12.2 Overall Framework Development of Water-Energy-Food-Nexus - Phosphate (WEF-P) Tool
*2.2.1 Overall Framework of WEF-P Tool*

The developed WEF-P Tool, adapted from the WEF Nexus Tool 2.0 (Daher and Mohtar, 2015), considers the supply chain of final product in terms of its resource consumption, including the set of processes that pass materials forward (La Londe and Masters, 1994; Mentzer et al., 2001), and various organizations or individuals directly involved in the flow of products (Mentzer et al., 2001). It assesses the impact of various scenarios and possible responses to regional resource management needs. Table 1 shows the differences between WEF Nexus Tool 2.0 and WEF-P Tool in the context of variables, scenarios, analytical tools, and quantitative assessments.

Both the Tools offer a platform for development of the analytics necessary to understand the trade-offs and catalyse a stakeholder dialogue (Mohtar and Daher, 2016; Mohtar, R. H. and Daher, 2014). The core of the WEF Nexus is that production, consumption, and distribution of water, energy, and food are inextricably inter-linked: decisions made in one sector impact the other sectors (Hoff 2011, Mohtar and Daher, 2012). The WEF Nexus Tool 2.0 allows holistic quantification of the impact of resource allocation strategies to support informed, and inclusive stakeholder dialogue between policy makers, private sector firms, and civil society (Daher and Mohtar, 2015). Each stakeholder becomes involved at different stages and scales in the decision-making process. In the WEF-P Tool (Figure 2), water resources are shared between the phosphate industry and the agricultural interests in the region of study. Sustainable water management must holistically consider the allocation of water resources for both industrial production and the agricultural irrigation of agricultural crops. New water (grey, produced, brackish, and treated urban wastewater from urban area) ha is a resource with the potential to contribute significantly contribute to bridging water and food gaps (Mohtar et al., 2015). H; however, it new water also carries an energy footprint that must be considered when increasing local food production. Potentially, aAgriculture's demand for water potentially competes with those ofe demands of a growing industry. The Tool quantifies the use of water and energy and the amount of $CO_2$ emitted for each scenario. It also quantifies the water and energy savings resulting from choices made regarding transportation scenarios. The Tool assesses the effects of decisions of dynamic management of phosphate production as these impact water and energy securities. The WEF-P tool can assess various scenarios and help account for the interdependencies between food and industrial production, and between water and energy consumption, . Only thus allowing can the trade-offs associated with potential resource allocation pathways to be quantified.

Throughout the tool development process, the supply chain was verified with OCP and the OCP Policy Center in various ways: (i) during the data collection phase, through meetings with the OCP steering committee, financial managers, technical managers and engineers; and (ii) through follow ups with OCP Policy Center team (conference calls and email). The OCP Policy Center team shared with WEF Nexus Team their main concerns regarding the tool structure, based on input from the OCP technical team. The WEF Nexus Team used these shared concerns in their considerations of revisions to the tool structure and associated excel spreadsheets of the model. Specifically, the major aggregated processes and lines of productions were revised and identified in a functional supply chain to maximize the abilities and flexibilities of the model and ensure efficacy of the available data base for processes and production lines.

However, the WEF-P Tool has limitations in assessing economic impacts such as cost and benefit analysis. This is because cost must include the price of water, which is still under discussion, and the price of products when analysing their benefits. Raw phosphate is transported to the manufacturing area and used in the production of various fertilizers that have different prices: this makes it difficult to set the price of excavating raw phosphate in the mining area. Sustainability assessment also has qualitative aspects in terms of environmental impact. The WEF Nexus Tool 2.0 applied the sustainability index based on resource capacity and availability, however, it is still a quantitative aspect. We should consider the meaning and definition of sustainability, both quantitatively and qualitatively, and then assess the index using the stakeholders' weights for the variables related to sustainability. Additionally, spatial and temporal scales should be included in a sustainability index. For example, the pipeline transportation system requires water, which is transported with products: the pipeline causes greater water use at the origin, but also provides additional water to the destination area. Also, the water requirement differs with temporal season, such as the water intensive agricultural production season. Thus, more research is needed for a sustainability assessment based

on economic and environmental impact. However, the quantitative analysis is an essential factor for assessing sustainability,
therefore, the WEF-P Tool focuses on quantification of 1) water and requirements for phosphate production and transportation,
2) carbon emissions by energy used in product processes, 3) water supply system and transportation, and 4) dynamic production
impacts on water and energy savings.

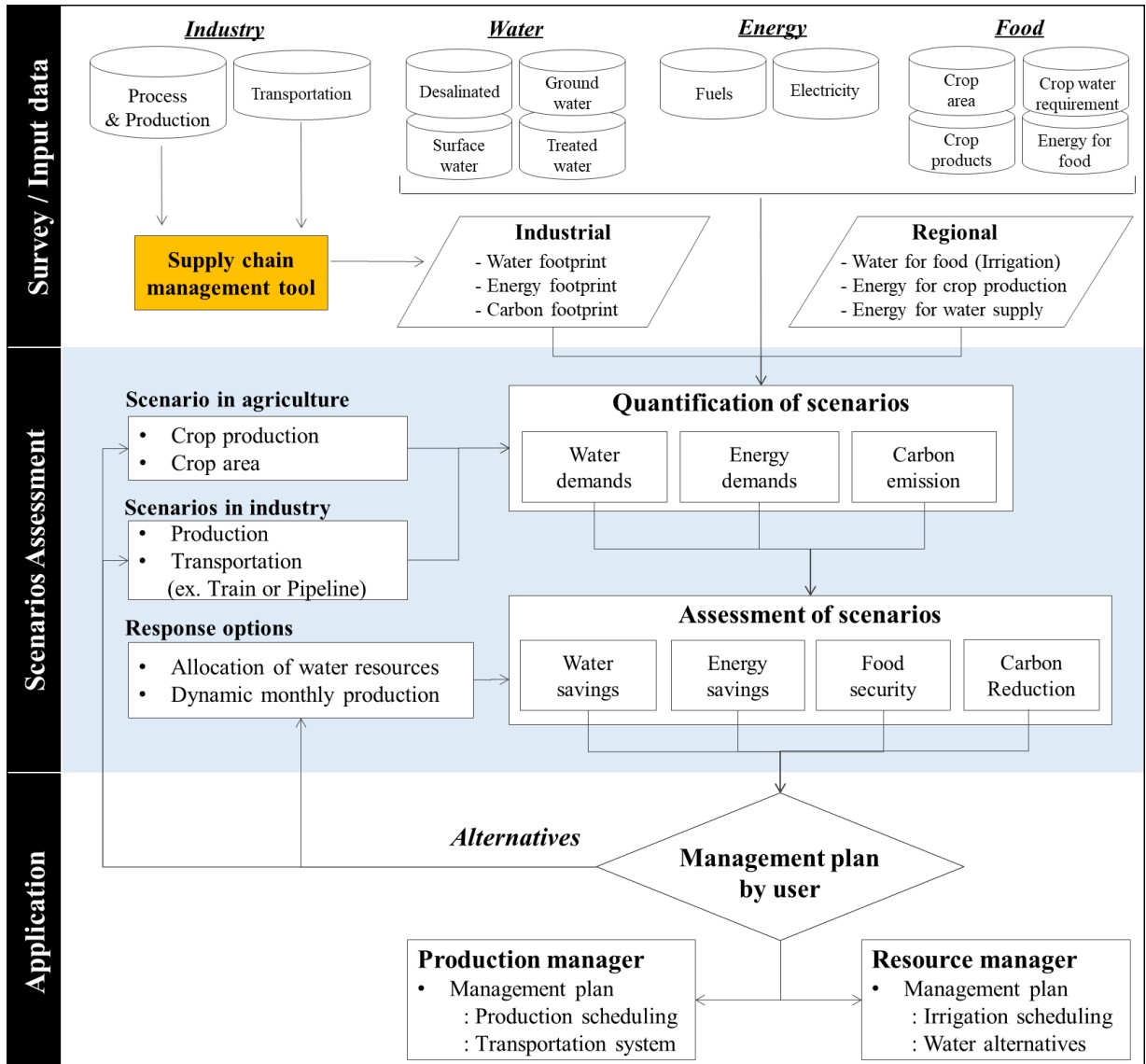

**Figure 1 2** Assessment of holistic impacts of various scenarios relating to phosphate industry, agriculture, and resource management using WEF-P Tool

~~The Tool quantifies the use of water and energy and the amount of CO₂ emitted for each scenario. It also quantifies the water~~
~~and energy savings that result from choices made regarding transportation scenarios. The tool assesses the effects of decisions~~
~~regarding dynamic management of phosphate production as these impact water and energy securities.~~
**Table 1** Comparison between WEF Nexus Tool 2.0 abd WEF-P Tool

|  | WEF Nexus Tool 2.0 | WEF-P Tool |
|---|---|---|
| Variables and scenarios | • Self-sufficiency of produced crops<br>• Type of agricultural production<br>• Sources of water (groundwater, surface water, treated water and so on)<br>• Sources of energy (natural gas, diesel, solar, wind and so on)<br>• Trade portfolio (countries of import and amounts per country) | • Static and dynamic phosphate production<br>• Transportation modes (train and pipeline)<br>• Sources of water (groundwater, surface water, treated water and so on)<br>• Water allocation between industry and agriculture |
| Analytical tool | • Food product base analysis<br>• Food-centric interlinkages among water, energy, and food<br>• Water and energy footprint based on product (ex. water footprint of crops) | • Process base analysis<br>• Phosphate-centric interlinkages among production, transportation, and resource allocation<br>• Water and energy footprint based on processes (ex. water footprint in washing process) |
| Quantitative assessment | • Water requirement for energy and agricultural production<br>• Energy requirement for agricultural and water production<br>• Land footprint for agricultural and energy production<br>• Carbon emissions from energy used for water and food production<br>• Financial cost | • Water and requirement for phosphate production and transportation<br>• Carbon emission by energy used in product processes, water supply system and transportation<br>• Dynamic production impacts on water and energy savings |


*2.2.2 Analysis of integrated supply chains linking sub-processes and transportation modes~~WEF-P Tool~~*
The WEF-P Tool used the WEF Nexus approach to assess the life cycle of the final products supply chain. The water and
energy used to produce sub- and final products ~~are~~ were calculated by adding the water and energy requirements from ~~or~~ the
sub-processes through the production supply chain ~~(Figure 2)~~. In Khouribga, raw phosphate products pass through a sequence
of functional processes: Mining and Screening (S), Washing (WW), Grinding (WG), Flotation (WF), Adaptation including
powdering (WA), and Drying (COZ - Complex Ouad Zem) for SC and (UB) for MEA (Figure 3). The mining and screening
processes include extraction from the ground, tone removal, and screening to produce pieces of phosphate rock. Here, the
supply chain is determined by the quality and size of the phosphate rock, which in turn, depends on the phosphate content at
extraction, ranges from very low to high. High quality phosphate rock is transported to a drying process from which it will
either be marketed or chemically transformed into fertilizer at the manufacturing site. Low to medium quality phosphate rock
goes is washed, dried, ground, and subjected to flotation, intended to increase the phosphate content.
The change in transportation system can affect the supply lines (Figure 3). In the mining area, the products are phosphate rocks
and slurry, both of which are transported to the manufacturing area, e. ~~E~~ach ~~product~~ with its own resource requirements. Slurry
requires flotation and adaptation, thus, is more water -intensive: p. ~~P~~hosphate rock is dried in ~~using~~ an energy intensive process
that consumes most of the energy produced in the mining area. Slurries are transported via pipeline and; rock ~~is transported~~ by
train; e. ~~E~~ach mode ~~of transportation~~ has ~~its~~ distinct~~own~~ resource needs at different stages. T~~, and~~ the two transportation
systems are also ~~have~~ distinct ~~processes~~: the pipeline supply chain includes ~~the~~ washing (water) and adaptation ~~processes to~~hat

produce slurry; the train supply chain includes ~~the~~ a fuel intensive drying process. It is possible to quantify the flow of products according to the transportation system used. When transport changes from train to pipeline, supply lines also change: the drying process is replaced by the adaptation process. ~~High quality phosphate rocks transported by train have undergone mining, screening, and drying. Low and very low-quality phosphate rocks also undergo a washing process in advance of any other processes.~~ If the phosphate is transported by pipeline, it must first be transformed to slurry, adding the adaptation process to the supply chain. Changes in the supply chain impact the water and energy consumed and, consequently, the $CO_2$ emitted. The mining and screening processes include extraction from the ground, tone removal, and screening to produce pieces of phosphate rock. ~~Here, the supply chain is determined by the quality and size of the phosphate rock, which in turn, depends on the phosphate content at the moment of extraction, which ranges from very low to high. High quality phosphate rock is transported to a drying process from which it will either be marketed or chemically transformed into fertilizer at the manufacturing site. Low to medium quality phosphate rock goes is washed, dried, ground, and subjected to flotation, intended to increase the phosphate content. Therefore, it is important to combine the supply chains for products and transportation systems.~~

**Figure 3** The functional processes and flows of products in Khouribga (mining) by transportation method.~~s~~

*2.2.3 Adaptation of process-base water, energy, and $CO_2$ footprints*

~~We considered the supply chain's integrating processes of production and transportation and adapted the water and energy footprints, which indicate the quantity of water or energy consumed in various sub-processes in integrated supply chains (Fig. 2).~~ The main function of the WEF-P Tool is identification of the relationship between resources and production, and the quantification of the resources consumed in phosphate production. The methodology is based on life cycle assessment. The water and energy footprints were analysed, indicating the quantity of water or energy consumed in various sub-processes in the supply chain's integration of production and transportation. The technical details of each process are specific and aggregated into functional processes. The main component is the footprint, which indicates the water and energy requirements for phosphate products, and the $CO_2$ emitted through energy consumption. Each process has a specific footprint based on field data and fed into the tool monthly, or when a significant change in capacity of the functional processes has occurred. For all footprint processes in Khouribga, the amount of raw phosphate is measured in commercial metric tons embedded in slurries and rock. Even if the phosphate rock changes to slurry through several processes, the amount of raw phosphate embedded in

products is not changed. Thus, the tons of phosphate in water and energy footprints indicate the raw phosphate embedded in
the products in each process and is constant through entire supply chains.
From the technical (engineering) perspective, footprints are calculated using a regression function, or average value based on
survey data; and technical experts in each process can modify this relation function as needed. The WEF-P Tool uses historical
data (from 2015) to estimate the average value of the footprint and the relationship between water/energy consumption and
phosphate production. First, we analysed the relationship between outputs of each process and water (or energy) consumption
was analyzed analysed. Second, the WEF-P Tool considered transportation of water and consumption of energy by train and
pipeline. Transportation by train was only related to fuel, i.e., diesel, consumption. However, the pipeline station consumes
electricity for operating the pipeline and freshwater is transported with slurry. The pipeline should be full, but of materials
such as slurry but it is impossible to fill the pipeline with slurry, thus it alternately carries slurry and freshwater. Therefore,
total water (or energy) consumption in the mining area includes not only water (or energy) used in processes but also that used
in transportation systems and the water consumed at the pipeline station in mining area, which basically indicates the
transported water used in the manufacturing area. WEF-Tool could quantify the water and energy consumption of the various
processes and at the pipeline station, as shown in equations (Eqs. 1-5).
$$WC_{mining\ area} = \sum_i^n (P_i \times WFP_i) + WC_{pipleline\ station} \qquad (1)$$
$$WC_{pipleline\ station} = P_{slurry} \times WC_{pipleline\ station} \qquad (2)$$
$$EC_{mining\ area} = \sum_i^n (P_i \times EFP_i) + EC_{pipleline\ station} + EC_{train} \qquad (3)$$
$$EC_{pipleline\ station} = P_{slurry} \times EFP_{pipleline\ station} \qquad (4)$$
$$EC_{train} = P_{phosphate\ rock} \times EFP_{train} \qquad (5)$$
where $WC_{mining\ area}$ (m³) is total water consumption in mining area, $EC_{mining\ area}$ (MWh or L) is total energy
consumption in mining area, $P_i$ (ton) is production from each process (i) in mining area such as mining, screening, washing,
flotation, and drying. $WFP_i$ (m³ ton$^{-1}$) and $EFP_i$ (MWh ton$^{-1}$ or L ton$^{-1}$) are water and energy footprints in each process (i).
$WC_{pipeline\ station}$ (m³) is water consumption in pipeline station, $EC_{pipeline\ station}$ (MWh or L) is energy consumption
in pipeline station, and $EC_{train}$ (MWh or L) is energy consumption by train to transport phosphate rock to manufacturing
area. $P_{slurry}$ and $P_{phosphate\ rock}$ (ton) are production of slurry and phosphate rock. $WFP_{pipeline\ station}$ (m³ ton$^{-1}$) is
water footprint at pipeline station in mining area. $EFP_{pipeline\ station}$ and $EFP_{train}$ (MWh ton$^{-1}$ or L ton$^{-1}$) are energy
footprints in pipeline station and of transportation by train. It is worth mentioning that the tool distinguishes between two types
of water: water transported from mining to manufacturing area by pipeline, and the embedded water in slurry.
$CO_2$ emissions are relevant when assessing the environmental impact of phosphate production. Although real emission in each
process in supply chain should be measured, this study is limited measuring $CO_2$ emission in mining area. In addition, $CO_2$
emission in crop area is related to soil and crops, and it is another level of research. Thus, we limited $CO_2$ emission to that
emitted by fuel energy use by machinery (direct emission) and electricity generation in power plants (indirect emission), and
the reference $CO_2$ footprints were applied (Table 2). These emissions are caused by burning the fuels used in the production
process and during generation of electricity. Fossil fuels (gasoline, diesel, coal, etc.), when burned, produce direct $CO_2$
emissions. Indirect $CO_2$ emission is also related to the source fuel used in generating electricity: indirect emission occurs in
the generation of electricity from other (non-fossil) sources, such as hydroelectric, wind power, or solar. According to USEIA
(2019) (https://www.eia.gov), one litre of gasoline used by machinery or a facility produces 2.6 kg of direct $CO_2$ emission;
and a power plant burning only coal to generate electricity, emits 1,026 tons of $CO_2$ kWh$^{-1}$. Renewable (non-fossil) electricity
emits only 15.8 tons of $CO_2$ kWh$^{-1}$. A survey of sources of electricity generation in Morocco indicates that coal is the main
fuel for power generation (43.4% of the national production). Oil and natural gas account for 25.3% and 22.7% respectively:
fossil fuels account for 90 % of the electricity produced in Morocco (IEA, 2014). Based on reference data, we calculated direct
and indirect $CO_2$ emission is calculated as shown in equations (6-7).
$$DCO_2 = \sum_i^n CFF\_F_i \times FC_i \qquad (6)$$
$$InDCO_2 = \sum_j^n CFP\_E_j \times ELC_j \qquad\qquad (7)$$
where $DCO_2$ (ton) is direct $CO_2$ emissions and $InDCO_2$ (ton) is indirect $CO_2$ emissions. $CFF\_F_i$ (ton L$^{-1}$) is $CO_2$ footprint by
burning fuel, $FC_i$ (ton L$^{-1}$) is fuel consumption by machine excluding fuel use for electricity generation, and i is the types of
fuels such as diesel or gasoline. $CFP\_E_j$ (ton MWh$^{-1}$) is $CO_2$ footprint by generating electricity, $ELC_j$ (MWh) is electricity
consumption, and j is the source of electricity generation such as coal, petroleum, natural gas, solar, wind, and hydropower.
**Table 2** $CO_2$ emission by burning fuels and generating electricity

| CO₂ emission by burning fuel | | CO₂ emission by generating electricity | | | |
|---|---|---|---|---|---|
| Sources | CO₂ emission[1] (kg of $CO_2$ L$^{-1}$) | Sources | CO₂ emission by sources[1] (ton of $CO_2$ $10^{-6}$ kWh) | Proportion of sources in Morocco[2] (%) | CO₂ emission (ton of $CO_2$ $10^{-6}$ kWh) |
| Gasoline | 2.59 | Coal | 1,026 | 43.4% | |
| Diesel | 2.96 | Petroleum | 1,026 | 25.3% | |
| | | Natural gas | 504 | 22.7% | 820.9 |
| | | Hydroelectricity | 19.7 | 6.9% | |
| | | Renewables | 15.8 | 1.7% | |

[1] U.S. Energy Information Administration (USEIA) (https://www.eia.gov)
[2] International Energy Agency (IEA)

~~**Figure 2** The functional processes and the flow of products in Khouribga (mining) and Jorf Lasfar (manufacturing)~~
**2.3 Agricultural Wwater and energy requirement for food production**
In this study, "water for food" indicates water that is withdrawn for crop production, generally for irrigation. CROPWAT 8.0
is a decision support tool developed by the Land and Water Development Division of FAO (Smith,
1992)(http://www.fao.org/land-water/databases-and-software/cropwat/en/), and was used to calculate the evapotranspiration,
crop water requirements, and irrigation requirements of four crops grown in the region. The climate data (temperature,
precipitation, humidity, wind speed, and hours of sunshine) were taken from the climatic database CLIMWAT 2.0, which
offers observed agro-climatic data from 5000 stations worldwide and provides long-term, monthly mean values of climatic
parameters. The compiled data of CLIMWAT 2.0 generally includes the period 1971-2000 (when this data was not available,
series ending after 1975 that include at least 15 years of data were used). Table 3 showed the average climate data in Khouribga
area provided from CLIMWAT 2.0.
CROPWAT 8.0 was used to calculate crop water and irrigation requirements based on soil, climate, and crop data. The
calculation procedures used in CROPWAT 8.0 are based on the FAO publication, Irrigation and Drainage Series: No. 44 and
56, Crop Evapotranspiration-Guidelines for computing crop water requirements (Allen et al, 1998; Smith, 1992). Irrigation
water requirements were calculated by estimating crop evapotranspiration (ETc), determined by multiplying the crop
coefficient (Kc) by the reference crop evapotranspiration (ETo), (see Eq. (8)). ETo is calculated using the FAO Penman–
Monteith method, as recommended by FAO and described in Eq. (9) (Allen et al., 1998).

$$ET_c = ET_0 \times K_c \tag{8}$$

$$ET_0 = \{0.408\Delta(R_n - G) + \gamma(900/T + 273)u_2(e_s - e_a)\}/\{\Delta + \gamma(1 + 0.34u_2)\} \tag{9}$$

where ~~ET~~$_0$~~ET~~ $ET_0$ is the reference crop evapotranspiration (mm/ day$^{-1}$); ET$_c$ is the crop evapotranspiration (mm d$^{-1}$); K$_c$ is the crop coefficient; $\Delta$ is the slope of the saturated vapor pressure/temperature curve (kPa °C$^{-1}$); $\gamma$ is the psychrometric constant (kPa °C$^{-1}$); u$_2$ the wind speed at 2 m height (m s$^{-1}$); R$_n$ is the total net radiation at crop surface (MJ /m$^{-2}$ day$^{-1}$); G is the soil heat flux density (MJ m$^{-2}$ day$^{-1}$ ~~MJ/m²day~~); T is the mean daily air temperature at 2 m height (°C); e$_s$ is the saturation vapor pressure (kPa); and e$_a$ is the actual vapor pressure (kPa). Crop coefficients are influenced by cultivation, local climatic conditions, and seasonal differences in crop growth patterns (Kuo et al., 2006). FAO provides crop coefficients for each stage. The values for Mediterranean countries were applied, as shown in (Table 1 4 (Allen et al., 1998)).

Irrigation water requirement was calculated by ETc and effective precipitation, as shown in Eq. (10). The effective precipitation indicated the precipitation except for runoff, and it was calculated using by the USDA Soil Conservation Service method (Eq. 11) (Smith, 1992).

$$IRReq = ET_c - P_{eff} \tag{10}$$

$$P_{eff} = P_{tot}(125 - 0.2\,P_{tot})/125 \qquad \text{for } P_{tot} < 250\ mm \tag{11}$$

$$P_{eff} = 125 + 0.1\,P_{tot} \qquad \text{for } P_{tot} > 250\ mm$$

where $IRReq$ is irrigation water requirement, ETc is the crop evapotranspiration, $P_{eff}$ is effective precipitation, and $P_{tot}$ is total precipitation.

**Table 3** Climate information in Khouribga

| Month | Precipitation (mm m$^{-1}$) | Temperature min. (°C). | Temperature max. (°C) | Relative humidity (%) | Sunshine hours (h d$^{-1}$) |
|---|---|---|---|---|---|
| Jan | 56 | 3.8 | 17.3 | 72 | 5.6 |
| Feb | 65 | 5 | 19 | 76 | 5.7 |
| Mar | 94 | 7.2 | 21.8 | 69 | 6.4 |
| Apr | 70 | 9.5 | 25.3 | 67 | 7.4 |
| May | 32 | 12.5 | 29.3 | 55 | 8.8 |
| Jun | 9 | 16.6 | 34.5 | 48 | 9.8 |
| Jul | 2 | 19.8 | 39.7 | 39 | 10.9 |
| Aug | 7 | 20 | 39.6 | 37 | 10.3 |
| Sep | 12 | 17.5 | 34.5 | 47 | 9.1 |
| Oct | 27 | 13.5 | 29 | 58 | 7.6 |
| Nov | 71 | 8.8 | 22 | 70 | 5.2 |
| Dec | 81 | 5.1 | 18.6 | 71 | 5.5 |

**Table 1 4** Crop planting and harvesting seasons, stage length and crop coefficients Information (Allen et al., 1998)

| Crop | Planting season | Harvesting season | Stage length (Days) | | | | | Crop coefficients | | |
|---|---|---|---|---|---|---|---|---|---|---|
| | | | Init. | Dev. | Mid | Late | Total | Kc init | Kc mid | Kc end |
| Olives | March | November* | 30 | 90 | 60 | 90 | 270 | 0.65 | 0.7 | 0.7 |
| Wheat | November | June* | 30 | 140 | 40 | 30 | 240 | 0.7 | 1.15 | 0.25 |
| Barley | March | July | 20 | 25 | 60 | 30 | 135 | 0.3 | 1.15 | 0.25 |
| Potato | Jan | April | 25 | 30 | 30 | 30 | 115 | 0.5 | 1.15 | 0.75 |

\* Next year

**3 Results and discussion**

**3.1 Application of scenarios**

Increasing the exportable phosphate products and changing the transportation system from train to pipeline are considered top priorities for OCP group. Therefore, we assessed the impact of increased production by applying the scenarios (Table 5). Until recently, dried phosphate was transported by train from mining to manufacturing site, but, in the near future OCP group will use only pipeline transport. The change of from train to pipeline can affect not only direct energy or water consumption by

transportation system but also that of the total supply chain in the mining site. Consequently, the production processes for slurry and for rock consume different quantities of water and energy, so that the mode of transport also becomes a scenario to allow quantification of their respective water and energy requirements.

Therefore, we applied the scenario about transportation system which indicates the only usage of pipeline. Table 4 showed the scenarios combining production and transportation. The first two scenarios are related to the 'business as usual (BAU)' scenario for production in 2015 but changing the transportation system from Khouribga to the terminal station at Jorf Lasfar. The other scenarios are related to the increase in the production.

**Table 5** Scenarios through combination of production and transportation system

| Scenario | Phosphate production | Transportation of phosphate products | |
| --- | --- | --- | --- |
| | | by pipeline | by train |
| BAU | Production in 2015 | 40 % of total phosphate | 60 % of total phosphate |
| Scenario 01 | | 100% of total phosphate | None |
| Scenario 02 | 50% increase of phosphate export | 40 % of total phosphate | 60 % of total phosphate |
| Scenario 03 | | 100% of total phosphate | None |

**3.1 2 Quantification of water and energy consumption and $CO_2$ emission by production and transport of phosphate**

To quantify the water, energy, and $CO_2$ emissions, water and energy footprints of each process in each mining site were analysed based on survey data. For example, the adaptation process is essential for pipeline transportation and large amounts water are needed in comparison to other processes, thus the relationship between the amounts of phosphate and water used in adaptation process were analysed (Figure 4 (a)). In addition, energy footprint includes electricity and fuel consumption; analysed through the linear relationship (Figure 4 (b)).

~~The amount of raw phosphate embedded in slurries and rock is measured in commercial metric tons, thus is the unit of phosphate product used in the study. In 2015, 1.68 million tons of raw phosphate was mined and transported from the mining to the manufacturing area, monthly. Target production was set at 2.45 million tons /month⁻¹ for phosphate products, and represents a 50% increase in phosphate exports (Table 2). Water and energy consumption depends on mode of transport (as slurry by pipeline or rock by train). The production processes for slurry and for rock consume different quantities of water and energy, consequently, the mode of transport also becomes a scenario to allow quantification of their respective water and energy requirements.~~

~~In the case of phosphate industry,~~ Pproduction and transport scenarios were applied and quantified for water, energy, and $CO_2$ emissions in each scenario (Table 3 6). In the mining area, 20.1 million tons of raw phosphate were produced in the 2015, ; this forms the "business as usual" (BAU) scenario. : 40% of the production was in the form of slurry and transported by pipeline; 60% was in the form of rock and transported by train. Scenario BAU indicates that 15.84 million m³ of water were used in all processing (both rock and slurry). Additional fresh water was transported through the pipeline to maintain slurry consistency in the system. For the BAU scenario, 3.85 million m³ of fresh water were transported by pipeline to the industrial area by pipeline. Scenario 1 (all raw phosphate transported by pipeline) increases the total water used to 32.14 million m³ (103% increase over BAU). Fresh water is also used to maintain the good operation of the pipeline, but with the increase in slurry transported by pipeline, the quantity of 'maintenance' fresh water decreased from 3.85 to 2.47 million m³, leading to a smaller total consumption of fresh water: a 76% increase was shown for total water consumption (for both processing and transport by pipeline).

Using only the pipeline for transport requires an additional 131,832 MWh in electricity for the flotation and adaptation processes used to produce slurry (31% increase in comparison to BAU). However, the consumption of fuel significantly

decreases as there is no need to dry phosphate rock. This results in a nearly 86% decrease in fuel consumption over the BAU scenario and a fuels savings of 176.4 million litre, which translates into a 40% decrease in $CO_2$ emission in Scenario 1. In ~~With regard to~~ Scenario 2, there was a 50% increase of raw phosphate export over the BAU scenario~~, and~~ with transport the same as in BAU. Total water consumed, including fresh water transported through the pipeline, increased by 16% over BAU, energy consumption increased by 46 %, and $CO_2$ emission increased by 39%. Scenarios 3 and 4 represent a 50% increase in phosphate exports, thus target production was set at 2.45 million tons month$^{-1}$ for raw phosphate, in total, 29.3 million tons yr$^{-1}$. Scenario 3 ~~(all raw phosphate is transformed to slurry and total production increased by 50% over BAU) also~~ indicates a total water consumption increase to 46.6 million m³ (137% over BAU), and electricity consumption increase ~~of~~d by 75%. However, transport by pipeline also led to an 80% decrease in fuel consumption (compared to BAU), and consequent 18% decrease in $CO_2$ emissions.

In summary, the comparison between BAU and scenario 1 showed the trade-off between water and energy by the change in transportation method. Pipeline transportation can save energy use and reduce $CO_2$ emission, but more water is required due to additional processes, such as adaptation and water used to operate the pipeline. However, since the water used to operate the pipeline is actually transported to Jorf Lasfar and re-used in fertilizer factories, it could be considered non-consumptive water in terms of the supply chain integrating Khouribga and Jorf Lasfar, even though it is still real water withdrawn form Khouribga.

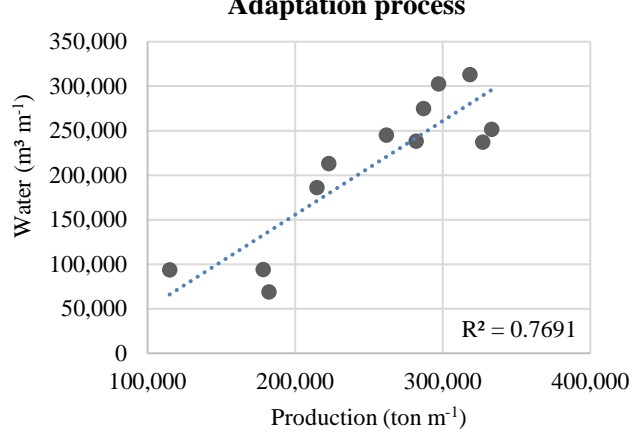
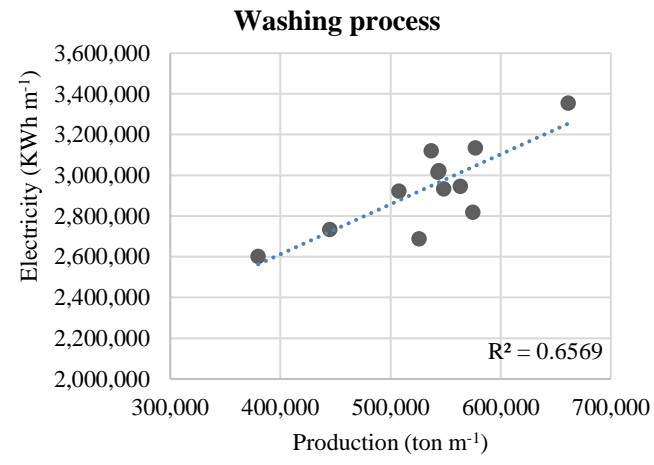

(a) Water footprint in an adaptation process in MEA    (b) Energy footprint in a washing process in MEA

**Figure 4.** Water and energy footprints in MEA based on the BAU data-base

~~**Table 2** Creation of scenarios relating to production and transportation of phosphate~~

| ~~Name of scenario~~ | ~~Contents of scenario~~ | ~~Production of phosphate (10⁶ ton yr⁻¹)~~ | ~~Phosphate in a form of slurry transported by pipeline (10⁶ ton yr⁻¹)~~ | ~~Phosphate in a form of rock transported by train (10⁶ ton yr⁻¹)~~ |
|---|---|---|---|---|
| ~~BAU~~ | ~~Production as BAU (2015)~~ | ~~20.1~~ | ~~7.8~~ | ~~12.3~~ |
| ~~Scenario 1~~ |  |  | ~~20.1~~ | ~~-~~ |
| ~~Scenario 2~~ | ~~Increase of production (50% over BAU)~~ | ~~29.3~~ | ~~11.3~~ | ~~18.0~~ |
| ~~Scenario 3~~ |  |  | ~~29.3~~ | ~~-~~ |

**Table ~~3~~ 6** Water and energy use, and $CO_2$ emission by scenario of phosphate production and transport

| Scenario (Transportation) | Water (10⁶ m³ yr⁻¹) | | Energy | | $CO_2$ emission (10⁶ ton yr⁻¹) | |
|---|---|---|---|---|---|---|
| | Water used in processes | Transported water | Electricity (MWh yr⁻¹) | Fuels (10⁶ L yr⁻¹) | Direct | Indirect |
| BAU | 15.84 | 3.85 | 424,512 | 204.0 | 0.53 | 0.35 |
| Scenario 1 | 32.14 | 2.47 | 556,344 | 27.6 | 0.07 | 0.46 |

| | | | | | |
|---|---|---|---|---|---|
| Scenario 2 | 19.35 | 3.45 | 551,495 | 297.9 | 0.77 | 0.45 |
| Scenario 3 | 45.15 | 1.45 | 743,928 | 40.5 | 0.11 | 0.61 |

**3.2 Assessment of the impacts of water allocation and treated water use in the industrial and the agricultural areas**
The main challenge of the mining area is sustainable water allocation for both the phosphate industry and ~~the~~ irrigated
agricultural areas. Thus, p~~P~~roduction targets were established for both phosphates and crops, and s~~S~~cenarios ~~were~~ evaluated
using the WEF-P Tool. Target crop production rates for Morocco's primary food crops (wheat, olive, barley, and potato), were
set as 0.1 % of national production. Table 4 7 shows that 25.07 million m³ yr⁻¹ of irrigation water is required to produce 5,722
ha of crops. In the case of wheat, irrigation requirements were calculated at 313.7 mm yr⁻¹, equivalent to 9.08 million m³ to
produce 0.1 % of national production annually.
The main water resource for the mining area is the Ait Messaoud dam. Water allocations from this source affect both phosphate
and agricultural areas. Water used for phosphate production increases when the pipeline is used to transport slurry (verses dry
rocks transported by train). The impact of water allocation under only the pipeline is calculated using various scenarios for
water allocation (Table 5 8) and the treated wastewater from urban area was considered as a water resource for both the
phosphate industry and agriculture.
In the "~~Alloc.~~ Alloc 1" scenario, supply capacity from the dam was set at 80% for the phosphate industry and 20% for the
agricultural area. The waste-water treatment plant ~~(capacity 5 million m³ yr⁻¹)~~ operates in the mining area~~,~~. F~~and f~~or scenario
~~Alloc.~~ Alloc 1, all treated water wa~~i~~s assigned to the phosphate industry. The "~~Alloc.~~ Alloc 2" scenario focuses on the
importance of water for agriculture and assigns the water equally between the phosphate and agricultural areas. Water supplied
from the dam, plus treated water from the plant may be insufficient for both industries. T~~;~~ to address this issue, treated water
and ground water were considered as supplementary water sources and ~~;~~ a treated water quantity of 2.5 million m³ yr⁻¹ (50 %
of current operation) was assigned to the two industries.
When water resources were allocated according to the Alloc. 1 scenario (80% of surface water and 100% of treated water
allocated to phosphate mining area), 9.68 million m³ additional water are required for agriculture (Table 6 9). When water is
allocated equally between the two industries (Alloc. 2 scenario), there is a shortfall of 9.61 million m³ in the phosphate industry,
but of only 70,000 m³ for agricultural irrigation. In the case of a 50% increase in phosphate production over BAU and using
the pipeline as the only mode of transport, the Alloc. 1 scenario indicates intensive water supply to phosphate mining area
rather than to agricultural area and causes an annual shortage of 5.59 and 16.07 million m³ water in the phosphate mining and
the agricultural area, respectively. To address this shortage, 2.5 million m³ of treated water could be supplied in addition to
19.16 million m³ of ground water.
Additionally, electricity is required to pump ground water and treat wastewater. Thus, the source of water may also affect
electricity consumption. Goldstein and Smith (2002) noted that 0.198 kWh is required to supply 1 cubic meter of ground water,
and the least electricity required to supplying surface water is 0.079 kWh m⁻³. Therefore, a 50% increase over BAU is
accompanied by 3,794 MWh yr⁻¹ electrical consumption for pumping ground water (Table 6 9). Increasing the use of treated
water releases the demand for ground water use, but the costs of building and operating the infrastructure and treatment facility
must be considered. In this study, the capacity of treated water was set at 2.5 million m³ yr⁻¹, and ground water requirements
were changeable only as scenarios of water allocation.

**Table 4 7** Water use for crop production under Moroccan condition

| Crops | Production* | Productivity | Area | Irrigation water requirement | |
|---|---|---|---|---|---|
| | ton | ton ha⁻¹ | ha | mm yr⁻¹ | $10^6$ m³ yr⁻¹ |
| Olive | 834 | 1.28 | 652 | 622.4 | 4.06 |
| Wheat | 4054 | 1.40 | 2895 | 313.7 | 9.08 |
| Barely | 1840 | 0.87 | 2115 | 562.7 | 11.90 |

| | | | | | |
|---|---|---|---|---|---|
| Potato | 1417 | 23.43 | 60 | 48.9 | 0.03 |
| Total | 8146 | | 5722 | 1547.7 | 25.07 |

*Crop production is the amount of 0.1% of national production in Morocco


**Table 5 8** Water allocation and treated water use scenarios

| Scenario | Sources | Capacity | Assignment of capacity | |
|---|---|---|---|---|
| | | $10^6$ m³ yr$^{-1}$ | Phosphate | Agriculture |
| Alloc 1 | Dam | 45.0 | 80% | 20% |
| | Treated water | 5.0 | 100% | 0% |
| Alloc 2 | Dam | 45.0 | 50% | 50% |
| | Treated water | 5.0 | 50% | 50% |
| Treated water supply | | 2.5 | 1st priority | 2nd priority |


**Table 6 9** Additional water and energy for solving water shortage by sceansrios of phosphate production

| Production (Only pipeline) | Water allocation | Water shortage | | Additional water supply | | Energy use for water supply | |
|---|---|---|---|---|---|---|---|
| | | Phosphate | Agriculture | Treated water | Ground water | Treated water | Ground water |
| | | $10^6$ m³ yr$^{-1}$ | $10^6$ m³ yr$^{-1}$ | $10^6$ m³ yr$^{-1}$ | $10^6$ m³ yr$^{-1}$ | MWh yr$^{-1}$ | MWh yr$^{-1}$ |
| Production as BAU | Alloc 1 | 0.00 | 9.68 | 2.50 | 7.18 | 1653 | 1421 |
| | Alloc 2 | 9.61 | 0.07 | | | | |
| 50% Increase over BAU | Alloc 1 | 5.59 | 16.07 | 2.50 | 19.16 | 1653 | 3794 |
| | Alloc 2 | 21.59 | 0.07 | | | | |

**3.3 Assessment of the impact of dynamic management of phosphate production on ground water and energy savings**
Water resource availability and water requirements for crop production are seasonal. Rainfall in June and July is less than 10
mm month$^{-1}$ and irrigation water requirements exceed 80 mm month$^{-1}$ (Figure. 35). Thus, there is water scarcity in the
agriculture area during June and July. Given that water resources are shared between the phosphate industry and the agriculture
industry, static production of phosphate could accelerate water shortage for agriculture. Dynamic production of phosphate is
considered as a scenario with greater agricultural production during non-irrigation seasons and less production during irrigation
seasons. Using the dynamic phosphate production scenario, the monthly production of phosphate decreases from 1.68 to 0.91
million tons month$^{-1}$ between May and October, representing a 50% decrease in raw phosphate export compared to BAU
scenario. Between November and April, phosphate production increases to 2.45 million tons month$^{-1}$, representing a 50%
increase in raw phosphate export in compared to BAU scenario.


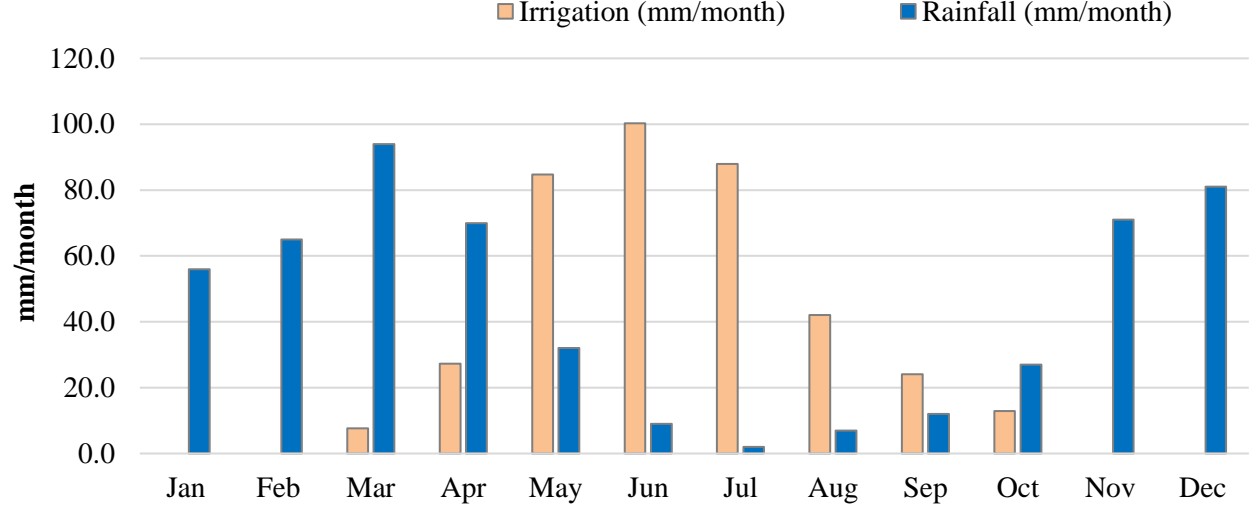

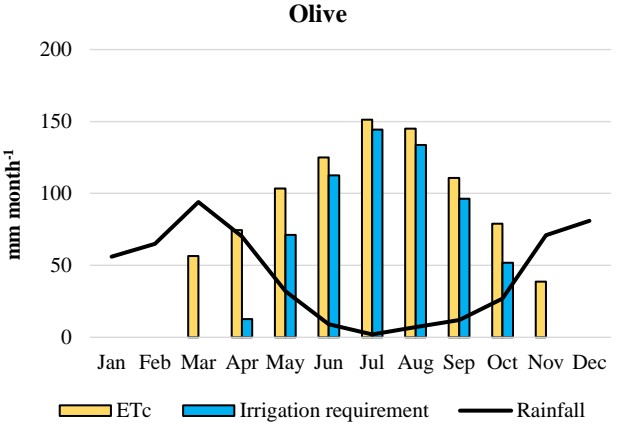

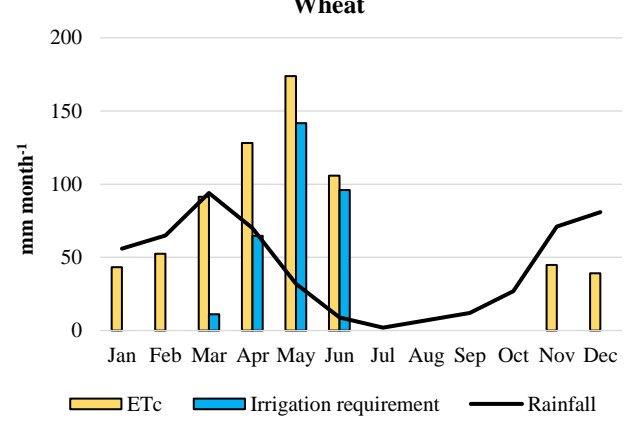

(a) olive

(b) wheat

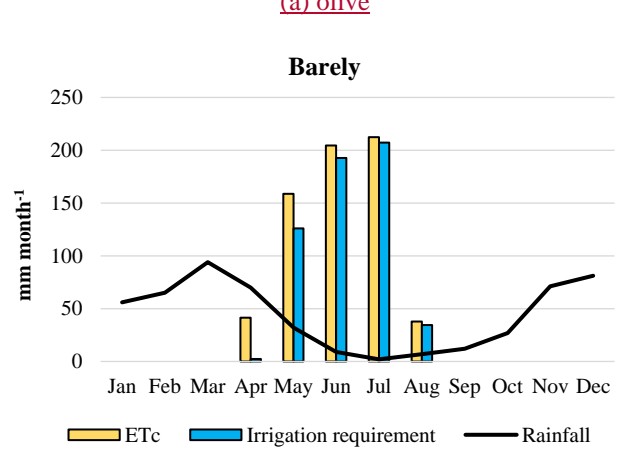

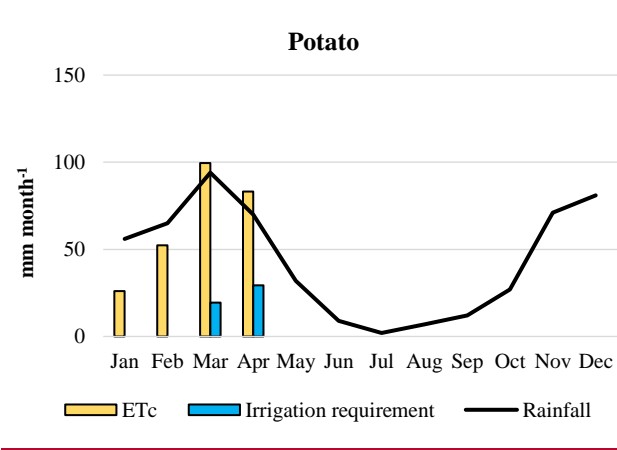

(c) barely

(d) potato

**Figure 5** Monthly irrigation water requirement and rainfall in Khouribga

Water availability and irrigation water requirements differ seasonally: dynamic monthly production of phosphate can contribute to sustainable water management. The effect of dynamic phosphate production on water supply becomes obvious when the pipeline is the only mode of transport: slurries are more water intensive than rock. Under static phosphate production, the monthly ~~water~~ demand for water from the dam in January and February was about 2.5 million m³, ~~and~~ increas~~ing~~ed to 7.0 million m³ month⁻¹ in June (Fig~~ure.~~ure 46). Nevertheless, dynamic phosphate production decreases the water demanded during the water scarce season. Moreover, the lack of water supply is covered by ground water: dynamic production uses less ground

water than static production (Figure. 7). During the water scarce season (May to July), total ground water used is 5.77 million
m³ in static phosphate production. This decreases by 10% in dynamic production, potentially saving 0.58 million m³ of ground
water during the water scarce season. Groundwater resources constitute an important aspect of the national hydraulic heritage
and represent the only water resource in this hyper arid climate (Tale, 2006). Thus, dynamic phosphate production carries
positive impact on sustainable water management and water conservation.
Dynamic phosphate production also contributes to electricity savings: supplying water from the dam, ground
, or treatment require electricity for pumping, transporting, and treating (Figure. 8). Total electricity consumed in
supplying water to the phosphate and agriculture industries was 9,971 MWh yr$^{-1}$ under the static production scenario
(phosphate slurries, no rocks). This number decreased to 9,828 MWh yr$^{-1}$ when phosphate slurries were produced dynamically.
About 143 MWh electricity can be saved annually,  accompanied by a reduction of 117 tons of $CO_2$ emission.

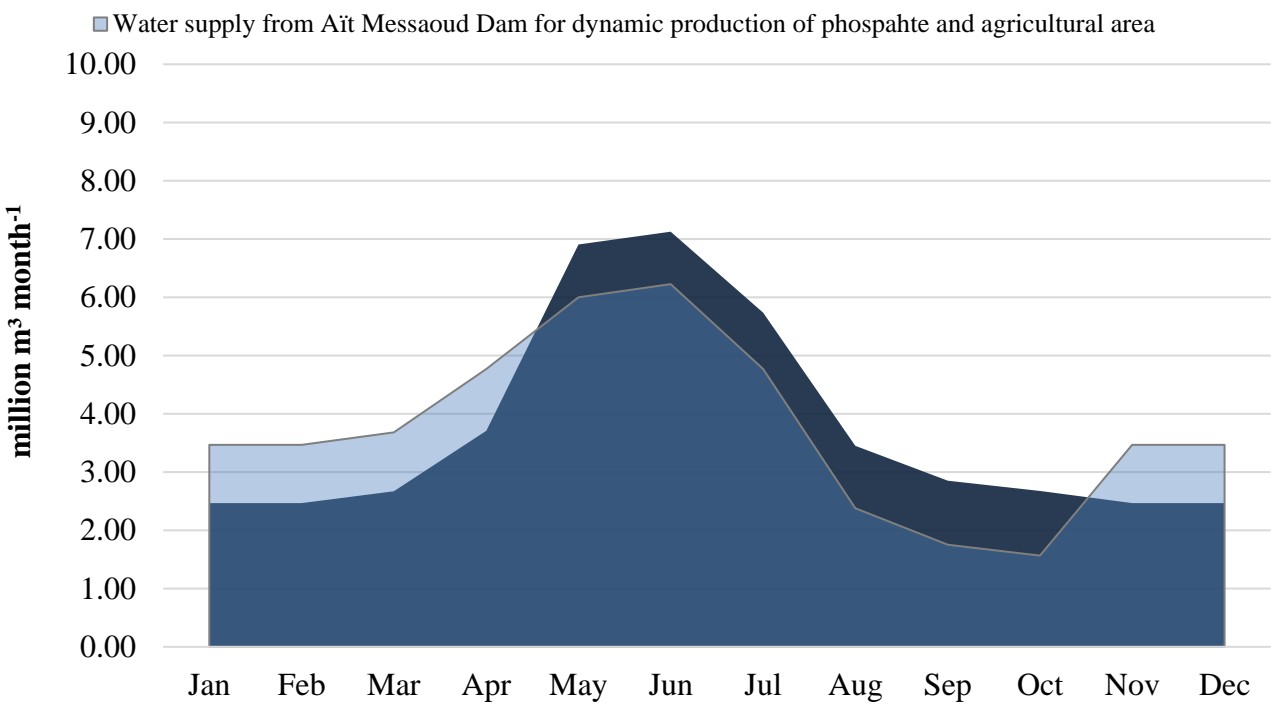

**Figure 4** ~~Monthly water supply from Aït Messaoud Dam~~

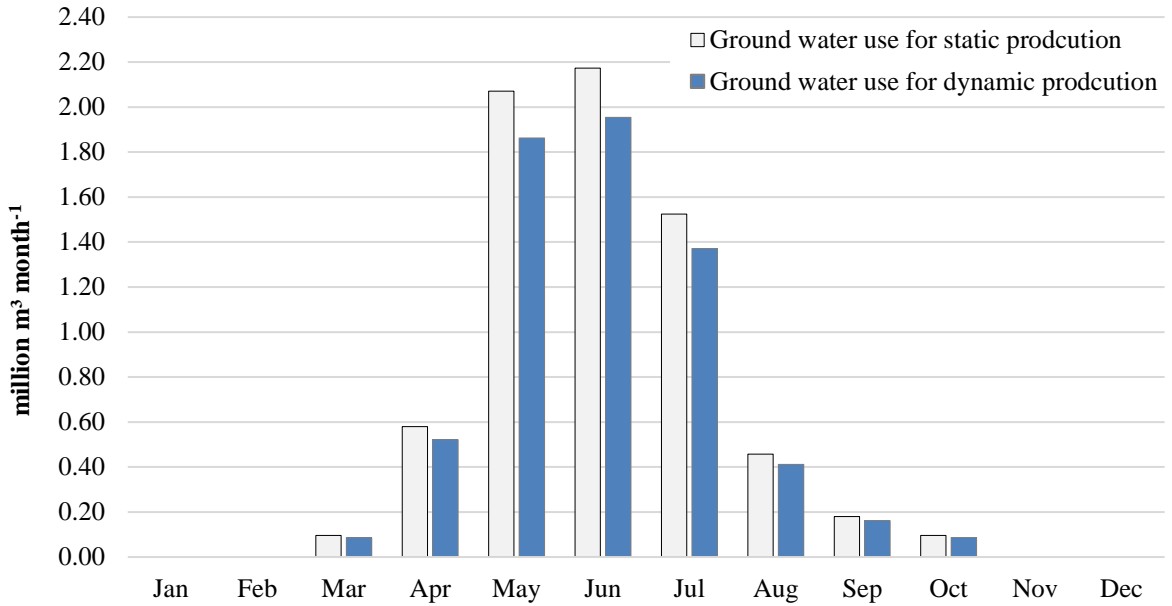

**Figure 5** Monthly ground water use by static and dynamic production of phosphate slurries transported by pipeline


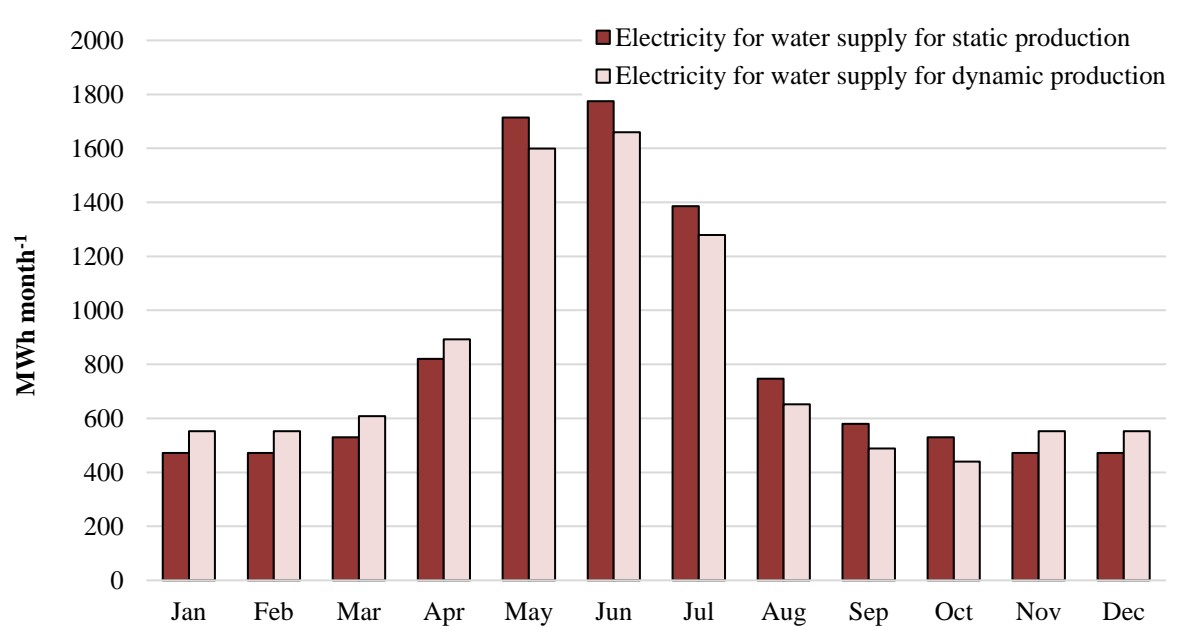

**Figure 6** Monthly electricity consumption for supplying water by static and dynamic production of phosphate slurries
transported by pipeline

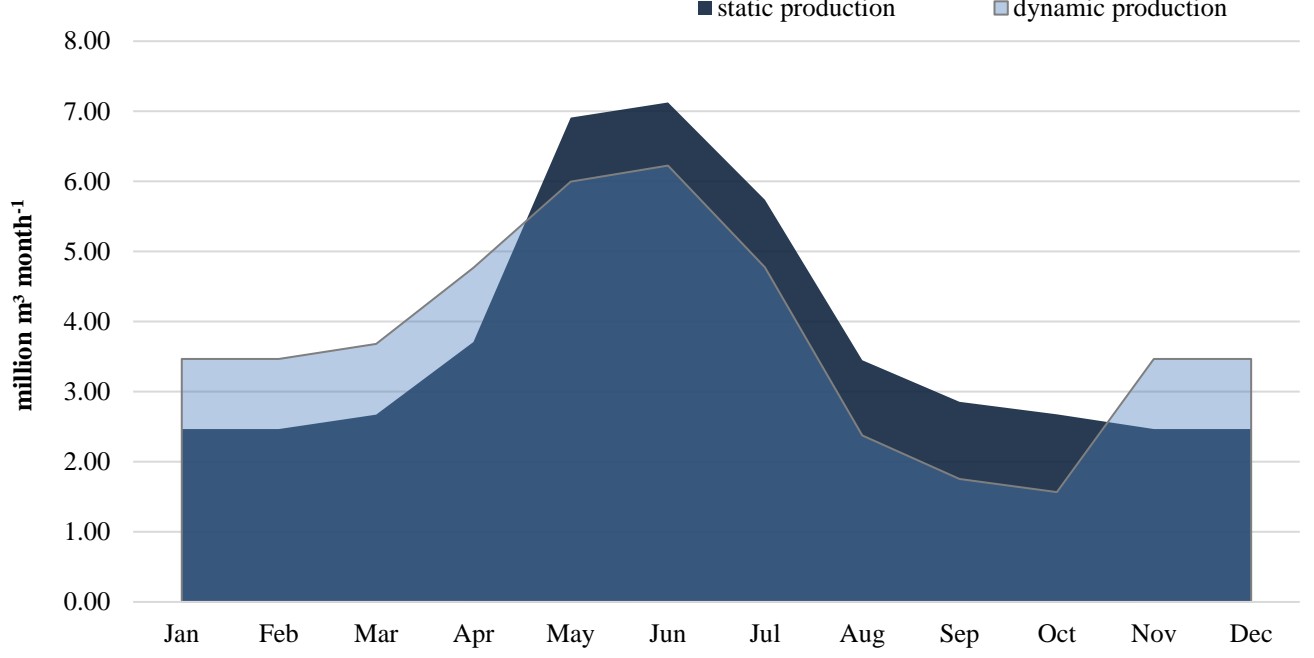

**Figure 6** Monthly water supply from Aït Messaoud Dam

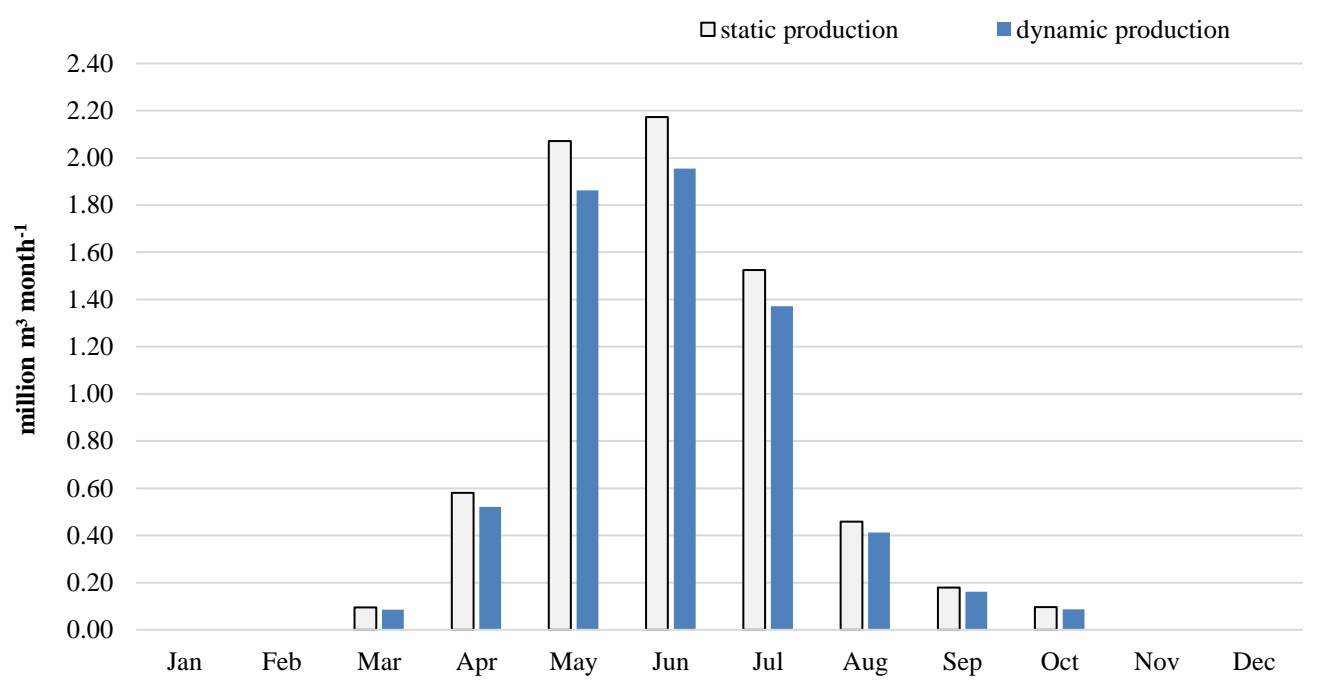

**Figure 7** Monthly ground water use by static and dynamic production of phosphate slurries transported by pipeline


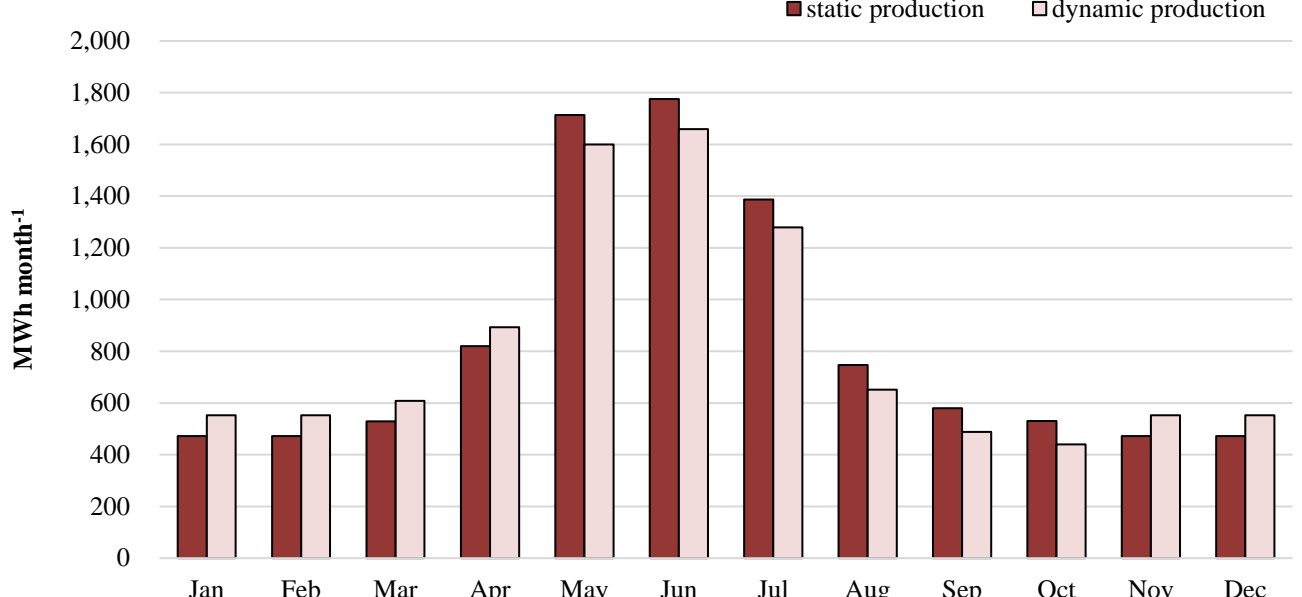

**Figure 8** Monthly electricity consumption for supplying water by static and dynamic production of phosphate slurries
transported by pipeline

## 4 Conclusions

As Morocco continues to work toward meeting its projected phosphate production goals, it is important to assess and quantify the potential resource competition between the growing municipal and agricultural sectors. ~~The WEF-P Tool integrates water-energy-food management and supply chain management for phosphate production. The strength of the WEF-P Tool is that it links the resource needs of the phosphate and agricultural areas. Previously, the WEF Nexus approach focused on water and energy used in food production, thus it was mainly related to agriculture. However, Khouribga is also a representative phosphate mining area and a main consumer of water and energy. Therefore, Ss~~Sustainable resource management strives for ~~a~~ symbiosis between the phosphate industry and other sectors, and endeavours ~~ring~~ to create synergy through multiple strategies. The WEF-P Tool integrates water-energy-food management and supply chain management for phosphate production, . ~~This tool~~ considering ~~s not only~~ the trade-offs between water, energy, and food, as well as ~~but also~~ a systematic analysis based on the total supply chain management of phosphate production. In other words, the WEF-P Tool offers a decision support system to provide quantifiable trade-off analyses for management decisions such as increasing production, transportation systems, and water allocation. The developed WEF-P Tool enables users to:

- understand and identify the associated footprints of the primary functional production processes and existing flows in production lines;
- identify the main sources of data to be gathered and fed into the model on a specific temporal basis;
- identify the techniques employed to conserve or produce water and energy and minimize the impacts of phosphate production;
- form a translational platform between sectors and stakeholders to evaluate proposed scenarios and their associated resource requirements

As phosphate mining increases, options that contribute to reducing water and energy stress include increased reliance on transport by pipeline and dynamic management of phosphate production. This tool assesses the impacts of various production pathways, including ~~for~~ specific process decisions throughout the phosphate supply chain, such as the choices for transport by

pipeline or train and the impacts on regional water and energy use. For example, transport by pipeline instead of train can contribute ~~an~~ to energy savings due to the elimination of the phosphate drying process (a main consumer of fuels). At the same time, the slurry adaptation processes are main consumers of water, ~~however~~though, because the pipeline also transfers fresh water to Jorf Lasfar where the fertilizers are produced, the water embedded in slurry is a main water resource for Jorf Lasfar. Previous~~ly, to receiving this source,~~ the main water resource in Jorf Lasfar was desalinated water, which ~~uses~~ consumes~~some~~ energy in desalination. Transport by pipeline contributes to a savings of desalinated water and energy for desalinating. The dynamic management scenario is assessed for its impacts on regional water and energy savings: dynamic management of phosphate production indicates different production quantities during irrigated and non-irrigated seasons. Less phosphate production during irrigation season can contribute more surface water for agricultural use and is accompanied by a savings of ground water and the energy required to pump ground water.

~~Nevertheless, there are limitations in the WEF-P Tool. However,~~ Further consideration of the economics of the phosphate operation is needed: static production may bring stability to operations (meeting local and export demand), but there are benefits from dynamic production that can be attributed to reduced competition with other water consuming sectors. Additional variables, such as ~~relating to~~ facility operation, ~~labor~~labour, economic cost/benefit of static and dynamic production, etc., should be quantified and included for additional trade-off assessments. ~~Supply chain management in the tool is based only on the 2015 dataset, thus it was difficult to validate the tool.~~ Quantification of water and energy for phosphate production is strongly dependent on the relationship between production and resource consumption: this can change in future scenarios. Proper water availability for the right place and time in a changing climate requires analysis of complex scientific, technical, socio-economical, regulatory, and political issues.

Beyond the limitations, the deliverables from this study include a conceptual and analytical model of the phosphate supply chain in Morocco, the WEF-P Tool. The Tool can assess the various scenarios to offer an effective means of ensuring the sustainable management of limited resources to both agricultural area and phosphate industry. It quantifies the products (phosphate) and resource footprints (water, energy) across the supply chain; identifies the interlinkages between water and energy in phosphate production and transport, and establishes reference values for comparison of outcomes and performance. The WEF-P Tool enables the user to evaluate trade-offs between water resource allocations and the impact of the Moroccan phosphate industry with agricultural water use.

**Author contribution**

Sang-Hyun Lee, Amjad T. Assi, and Rabi H. Mohtar conceived and designed the research; Sang-Hyun Lee and Amjad T. Assi analysed the data; Sang-Hyun Lee, Bassel T. Daher, and Fatima E. Mengoub contributed analysis tools; Sang-Hyun Lee and Amjad T. Assi wrote the paper.

**Competing Interests**

The authors declare that there are no conflicts of interest regarding this publication.

**Acknowledgments**

This research was funded by the OCP Policy Center and the OCP foundation and also supported by the Japan Science and Technology Agency as part of the Belmont Forum. The authors thank the OCP Policy Center for arranging meetings with their engineers and managers in the mining and production areas for data collection. The authors also express their appreciation to Mary Schweitzer of the WEF Nexus Research Group at Texas A&M University for her editorial assistance and contributions.

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
