# Peer review of "A Water-Energy-Food Nexus Approach for Conducting Trade-off Analysis: Morocco's Phosphate Industry in the Khouribga Region"

_Hydrology and Earth System Sciences, 2020_

## Referee Comment (RC1) · Anonymous Referee #1 · 16 Jun 2020

This paper brings a timely and relevant topic about going beyond classical mono criteria approaches to a more complete comprehensive approach that analyses not just one aspect of the system but rather involves more indicators which help to conduct trade-offs analysis on shared primary resources between competing sectors. The authors applied a Water-Energy-Food Nexus-tool to assess the interaction between the agricultural sector and the phosphate industry on regional water, energy, and food systems in Khouribga, a representative phosphate mining area of Morocco. The authors suggested that their developed tool can assist in making a more sustainable resource management plan. Their results recommended the dynamic monthly management of phosphate production in terms of water and energy savings, especially during the

water-scarce season. However, I have some methodological and structural concerns that need to be addressed before publication.

General comments:

1. The authors developed and used WEF-P tool which takes into consideration a number of footprints indicators such as water and energy. The authors decided to exclude the economic perspective from their analysis without giving a concrete reason. Since the authors claimed that their tool can "becomes a management-decision aid for effectively ensuring more sustainable management of limited resources and increased reliability of water resources for both agricultural and industrial use". The economic aspect of phosphate production needs to be addressed and included in this paper. Otherwise, the authors need to give a very good reason if they decide to don't do so (not the way they mention it now briefly in their study limitation) and maybe avoid overestimating the effectiveness of their tool as a decision aid.

2. The water footprint (WF) is defined loosely. The authors referred to water footprint calculation while in the end, they seem to use the evapotranspiration as irrigation water requirement which needs clarification since the water required for irrigation is not the same as evapotranspiration and not the same as water footprint of crop production.

3. The authors developed target production scenarios that they mention in their result and discussion section. How are the scenarios developed and why? The authors could include their scenarios in the materials and method section: "The production of raw phosphate of the year 2015 forms the "business as usual scenario" (BAU). Three additional scenarios will be considered based on a combination of the production process and the mode of transportation. Scenarios 1 is when all raw phosphate is transported by pipelines . . ." something like this.

4. When calculating the WF, what was the considered period? Is it the calendar year or the growing period of crops? This needs clarification and the study boundaries should be well defined. If the authors considered the calendar year, they also need to assume

that there was nothing planted from the previous year. Otherwise, in some months of the year, there will be an overlap between current crops and last year crops that should be considered. For wheat, for example, it is panted in November so it will be harvested in the next year. Is the water used in the next year will not be accounted for or is wheat considered a rainfed crop?

5. The authors need to improve the structure of their paper. The data, methodology, units and scenarios need to be defined before the results and discussion section. It is sometimes hard to differentiate the study assumptions and the defined scenarios from the results and discussion.

Specific comments:

1. The introduction and the article, in general, have a relatively limited literature review. This lack of references is, in my opinion, caused by two facts. First, the authors developed and used a tool while they didn't introduce anything concerning the tool creation, why it is needed? What is the difference between the WEF Nexus Tool 2.0 and the one created here and other frameworks and Nexus tools? What are the limitations of other tools in assessing what the current tool could assess? Second, the authors should make more references to other studies that used the WEF concept and compare their framework and findings. This will also be useful for the authors to place their findings in the context of other studies that applied the WEF Nexus in the paper's discussion.

2. Lines 47 - 62: WEF concept has also been recognized as a strong support to achieve the Sustainable Development Goals (SDGs) see Terrapon-Pfaff et al. (2018) for instance: Terrapon-Pfaff, J., Ortiz, W., Dienst, C., and Gröne, M.-C.: Energising the WEF nexus to enhance sustainable development at local level, Journal of environmental management, 223, 409-416, https://doi.org/10.1016/j.jenvman.2018.06.037, 2018.

3. Line 66: You better refer to the world bank (2019) and not to the link in the text and in reference list refer to "World development indicators".

4. Line 68: Same as the previous comment, refer to FAO (2015) in the text and the following in the reference list: FAO: FAOSTAT Online Database, Statistics Division, Food and Agriculture Organization of the United Nations (FAO), Rome, Italy, 2015.

5. Line 70: also, here refer to world bank 2019 and maybe use suffixes a and b following the publication year to make the difference between the references to world bank data you made.

6. Line 71: maybe add a reference here?

7. There are many sentences in the introduction that seem to be quotes from literature that are not cited. Check lines 74 - 84 for example. For instance, you should give a reference to the following: "especially in a country that imports nearly 90% of the energy it consumes".

8. Lines 97 - 98: "New water (grey, produced, brackish, and waste) is a resource with the potential to significantly contribute to bridging water and food gaps (Mohtar et al., 2015)" you make a difference between grey and wastewater and between produced and brackish water. What is the difference between grey and wastewater? What is the source of produced water then, are you referring to desalinisation? What is the share of this new water resources use in total water used in agriculture? How safe is the use of treated wastewater in agriculture in Morocco to consider its contribution as significant?

9. Lines 113 - 135: In your site description, it is not clear what is your system boundaries. Are you considering the whole region Khouribga? Or just the mining areas and it's surrounding? How is the surrounding defined for agriculture activity for instance?

10. Line 117: "fields indicate 1.68 million tons of raw phosphate were excavated". What represents this number for the total country's production of raw phosphate?

11. Line 128: "Plan Maroc Vert". It is maybe better to refer to the English name: the "Green Morocco" plan. Maybe add a reference to the Green plan or the national water

plan that refers to moving from groundwater to surface water use.

12. Lines 160 - 161 "footprints are calculated using a regression function, or average value based on survey data, and technical experts in each process can modify this relation function as needed". This is not very clear. What regression function you used to calculate water and energy footprint? Please elaborate in this and explain what data comes from surveys and which technical experts you mean?

13. Line 206: You better refer to FAO report 46.

14. What is the methodology you used for accounting the water footprint of crop production?

15. Lines 231 - 232: Is it possible to include the unit of phosphate production somewhere in the materials and methods section?

16. The following sentences give the same information: "In 2015, 1.68 million tons of raw phosphate was mined and transported from the mining to the manufacturing area, monthly" and "In the mining area, 20.1 million tons of raw phosphate were produced in the 2015". The only difference is monthly or yearly production. Can you combine the information about target production and the BAU scenario to avoid repetitions?

17. Lines 266 – 267: Are the considered crops the only produced crops in that area? Why setting the target production to exactly 0.1%? Is that the potential production of the considered area? Is that the target production for each crop?

18. Food production is not only crop production but also livestock production. Since you are not including this aspect of food production in your study you need to maybe spend one sentence in the limitation of the paper or somewhere to make this clear.

19. Line 274: The waste-water treatment plant capacity seems to be taken from somewhere, maybe add a reference?

20. In figure 3, the amount of rainfall in March seems to exceed rainfall in November

and December but irrigation is still needed in March. You better include the harvested date next to the planting date in Table 1 to give an idea of how many crops have their growing period in March.

21. The first sentences in Line 303 and line 316 seem to be the same.

22. Line 327 – 331: Dynamic phosphate production contributes to electricity savings in only 6 months of the year (from May till October). However, in the rest of the year, the consumption of electricity in the dynamic phosphate production was higher than the static production.

Minor comments: 1. Line 64 and Line 74: You miss a comma (Taleb, 2006) and (OCP, 2013). These are just examples; you need to check all your references and format them according to HESS guidelines.

2. Line 91, energy is repeated twice.

3. Table 1. You better remove the reference from the table's title and insert it in Line 225: "FAO provides crop coefficients for each stage". Alternatively, you can add it as a note under the table.

4. In Table 1's title "Information" doesn't need to be capitalized.

5. In Table 1: Plant data: do you mean Planting date?

6. Line 233: remove the additional backslash –> tons months-1

7. Figure 4: Include more description in the figure's title and reduce the text in the figure's legend. Try to include the legend in the figure as it seems now to be outside.

8. Same for figure 5: Groundwater use is already in the figure's title. The legend could be just: Dynamic production and Static production.

9. Same for figure 6.

10. There are some small typographical errors throughout the paper and these should

be corrected.
* * *

---

## Referee Comment (RC2) · Anonymous Referee #2 · 17 Jun 2020

Summary

In this paper, the author emphasized a unique type of nexus that can help us to have a clear idea about the importance of Phosphate production in Morocco and its impact on water and energy use and agricultural production. The author imported the WEF tool and applied it to the WEF-P case study. The results seem to be promising if the nexus Phosphate doesn't exist. Otherwise, the WEF tool is more efficient in the calculation of water and energy demand and carbon emission, but in the case of the phosphate, it is quite critical since there is a need to calculate the environmental risks caused by the phosphate production on water resources and crop production. Moreover, phosphate

production didn't impact only the water and energy footprint and CO2 emission but it can destroy the agricultural land (phosphogypsum, radioactive wastewater), affect air quality (dust) and human health (phosphate rock has been reported to possess toxicity to varying degrees, and Be, As, Cd, Hg, Tl, and Ra are generally designated as extremely toxic). All this has led me to have some interrogations:

1- The author didn't give a clear idea about the water footprint and energy footprint and how they contribute in the assessment scenarios, did they matter when it comes to producing a clean food? And concerning the Carbone footprint, is the Carbone show a real danger in front of all the radioactive components of phosphate?

2- The author applied the tool considering the phosphate as a simple nexus component like water or energy. Otherwise, there is a need to emphasize all environmental and economic aspects related to this product and the interaction between agricultural and phosphate production in the study area, to consider that this tool is efficient and can be considered as the best decision-making tool when it comes to this type of nexus.

3- The paper is not well structured, there is a gap between different sections in the paper. The author didn't explain the choice of the scenarios and the methodology of data collection.

4- In the discussion part, no scientific comparison, even if the tool developed by the author, it is important to compare the findings, especially in the WEF nexus impact on water and energy footprint and CO2 emission.

Specific comments

1- The introduction is too general and missing a good literature review. The author didn't talk about the novelty of the use of the WEF-P tool and the difference between it and the WEF tool (http://www.wefnexustool.org). There is also a need to emphasize the pros and cons of the used tool, especially that phosphate production has a very interesting economic value but a very bad environmental impact, and here comes the

importance of the concept of the sustainability index existing in the WEF tool 2.0.

2- Line 65: it is better to use the actual information as 2019 for the population growth and avoid using hyperlinks in the text as reference the same in the lines 68 and 70

3- Line 76 to 80: the author needs to add a reference for the quote

4- Part 2.1: from where came the differentiation of the different types of water and from which background the new water concept is coming? and is the wastewater used is coming from phosphate washing? If yes, are you sure that it is safe?

5- Figure 1: the description of the figure is missing. Otherwise, there are two missing key concepts need to be considered: sustainability index and environmental index (related mainly with the Phosphate toxicity)

6- In the site description: it will be good to have a general idea of the study area (location, climatic conditions: rainfall, temperature, wind, radiation, water resources, soil resources, agricultural activity, energy source) since the author involved the evapotranspiration calculation and water and energy footprints.

7- Part 2.2.2 the author underlined mainly the different steps of phosphate production and missed a good explanation of the footprint calculation and the data gathering methodology and date frame of the collected data. For the $CO_2$ emission, the author linked it only with energy use, but he forgot to mention the importance of having a healthy soil can play a crucial role in carbon sequestration.

8- Part 2.3: the author used the ETP requirement to calculate the irrigation water requirement which needs to be revised and does the used data in this calculation are reflecting the exact situation of the study area?

9- Table 1: do you mean by plan data the plantation season or the date of data collection?

10- For the results and discussion, the choice of the scenarios should be clarified

before

11- Figure 3: it will be better if you consider the ETP to extract good information

12- Line 64: (Taleb 2006) a comma is missing

13- Line 74: (OCP 2013) a comma is missing

14- Figures from 1 to 6: add short descriptions

---

## Author Comment (AC1) · 10 Aug 2020

Dear Editor and Reviewers

Thank you for considering the manuscript for publication in the Hydrology and Earth System Sciences (HESS) and in-depth review of the manuscript. In this study, we developed the WEF-P tool which is a decision support system for linking phosphate industry to agriculture in terms of water-energy nexus perspective. In particular, we adapted the supply chain analysis to quantify the water and energy footprints and assessed the impacts of water allocation between industry and agriculture through dynamic production of phosphate using the WEF-P Tool.

[Figure]

The main comments from reviewers were related to 1) lack of literature reviews, 2) strength of this tool in comparison to others, and 3) economic and environmental impacts assessment. Therefore, in the revised manuscript, we revised the introduction with more literature reviews and reorganized the structure of our manuscript in order to improve its readability and highlight the novelty of the present work. In particular, we have detailed explanation about methodology of the tool, data survey, scenarios, and footprints modeling. In addition, we compared this WEF-P Tool with WEF Nexus 2.0, and added the limitation of economic and environmental impacts assessment through the WEF-P Tool.

Main revisions - Revising introduction with more literature reviews - Reconstructing and revising the materials and methods section - Adding limitations of economic and environmental impacts assessment

In the revision notes, you will find a point-by-point reply to specific comments.

We appreciate again your thoughtful comments and look forward to hearing your reply.

Kind regards, on behalf of all co-authors,

Please also note the supplement to this comment:
https://hess.copernicus.org/preprints/hess-2020-197/hess-2020-197-AC1-supplement.pdf

**Supplement:**

**Reply to the reviewers for hess-2020-197**

Title: A Water-Energy-Food Nexus Approach for Conducting Trade-off Analysis: Morocco's Phosphate Industry in the Khouribga Region

Authors: Sang-Hyun Lee, Amjad T. Assi, Bassel T. Daher, Fatima E. Mengoub, Rabi H. Mohtar

Dear Editor and Reviewers

Thank you for considering the manuscript for publication in the **Hydrology and Earth System Sciences (HESS)** and in-depth review of the manuscript. In this study, we developed the WEF-P tool which is a decision support system for linking phosphate industry to agriculture in terms of water-energy nexus perspective. In particular, we adapted the supply chain analysis to quantify the water and energy footprints and assessed the impacts of water allocation between industry and agriculture through dynamic production of phosphate using the WEF-P Tool.

The main comments from reviewers were related to 1) lack of literature reviews, 2) strength of this tool in comparison to others, and 3) economic and environmental impacts assessment. Therefore, in the revised manuscript, we revised the introduction with more literature reviews and reorganized the structure of our manuscript in order to improve its readability and highlight the novelty of the present work. In particular, we have detailed explanation about methodology of the tool, data survey, scenarios, and footprints modeling. In addition, we compared this WEF-P Tool with WEF Nexus 2.0, and added the limitation of economic and environmental impacts assessment through the WEF-P Tool.

Main revisions
- Revising introduction with more literature reviews
- Reconstructing and revising the materials and methods section
- Adding limitations of economic and environmental impacts assessment

In the revision notes, you will find a point-by-point reply to specific comments.

We appreciate again your thoughtful comments and look forward to hearing your reply.

Kind regards, on behalf of all co-authors,
* * *
Sang-Hyun Lee

**Revision Notes**

**Major Comments (Reviewer #1)**

**Comments 1**

| | |
|---|---|
| Reviewer's comments | 1. The authors developed and used WEF-P tool which takes into consideration a number of footprints indicators such as water and energy. The authors decided to exclude the economic perspective from their analysis without giving a concrete reason. Since the authors claimed that their tool can "becomes a management-decision aid for effectively ensuring more sustainable management of limited resources and increased reliability of water resources for both agricultural and industrial use".

The economic aspect of phosphate production needs to be addressed and included in this paper. Otherwise, the authors need to give a very good reason if they decide to don't do so (not the way they mention it now briefly in their study limitation) and maybe avoid overestimating the effectiveness of their tool as a decision aid. |
| Response | Some of phosphates are exported but a lot of them are transported to Jorf Lasfar and used as raw materials for phosphorous fertilizers. Thus, the economic value of phosphate could be changed by the types of fertilizers, and it is actually difficult to apply the static economic value to the model. In addition, still there are a lot of discussion about water value are ongoing. Thus, we added more explanation why we did not mention the economic perspective in this study. |
| Revision (Line 180-194) | However, the WEF-P Tool has limitations in assessing economic impacts such as cost and benefit analysis. This is because cost must include the price of water, which is still under discussion, and the price of products when analysing their benefits. Raw phosphate is transported to the manufacturing area and used in the production of various fertilizers that have different prices: this makes it difficult to set the price of excavating raw phosphate in the mining area. Sustainability assessment also has qualitative aspects in terms of environmental impact. The WEF Nexus Tool 2.0 applied the sustainability index based on resource capacity and availability, however, it is still a quantitative aspect. We should consider the meaning and definition of sustainability, both quantitatively and qualitatively, and then assess the index using the stakeholders' weights for the variables related to sustainability. Additionally, spatial and temporal scales should be included in a sustainability index. For example, the pipeline transportation system requires water, which is transported with products: the pipeline causes greater water use at the origin, but also provides additional water to the destination area. Also, the water requirement differs with temporal season, such as the water intensive agricultural production season. Thus, more research is needed for a sustainability assessment based on economic and environmental impact. However, the quantitative analysis is an essential factor for assessing sustainability, therefore, the WEF-P Tool focuses on quantification of 1) water and requirements for phosphate production and transportation, 2) carbon emissions by energy used in product processes, 3) water supply system and transportation, and 4) dynamic production impacts on water and energy savings. |

**Comments 2**

| | |
|---|---|
| Reviewer's comments | 2. The water footprint (WF) is defined loosely. The authors referred to water footprint calculation while in the end, they seem to use the evapotranspiration as irrigation water requirement which needs clarification since the water required for irrigation is not the same as evapotranspiration and not the same as water footprint of crop production. |
| Response | Irrigation water requirement was calculated using CropWat model, and not only evapotranspiration but runoff of rainfall was applied as well. We used the reference methodology (USDA SCS method) from CropWat explained in FAO No. 46 report.

Thus, we added more explanation about irrigation requirement modeling based on ETc and runoff that is provided in CropWat model. Please find the addition explanation as below. |

**Revision Notes**

| | |
|---|---|
| Revision (Line 307-314) | Irrigation water requirement was calculated by ETc and effective precipitation, as shown in Eq. (10). The effective precipitation indicated the precipitation except for runoff, and was calculated using the USDA Soil Conservation Service method (Eq. 11) (Smith, 1992).

$$IRReq = ET_c - P_{eff} \qquad\qquad (10)$$
$$P_{eff} = P_{tot}\,(125 - 0.2\,P_{tot})/\,125 \qquad \text{for } P_{tot} < 250\ mm \qquad (11)$$
$$P_{eff} = 125 + 0.1\,P_{tot} \qquad\qquad \text{for } P_{tot} > 250\ mm$$

where $IRReq$ is irrigation water requirement, ETc is the crop evapotranspiration, $P_{eff}$ is effective precipitation, and $P_{tot}$ is total precipitation. |

**Comments 3**

| | |
|---|---|
| Reviewer's comments | 3. The authors developed target production scenarios that they mention in their result and discussion section. How are the scenarios developed and why? The authors could include their scenarios in the materials and method section: "The production of raw phosphate of the year 2015 forms the "business as usual scenario" (BAU). Three additional scenarios will be considered based on a combination of the production process and the mode of transportation. Scenarios 1 is when all raw phosphate is transported by pipelines . . ." something like this. |
| Response | When we develop this tool, we contacted the managers and engineers working in the OCP group and OCP policy center, and had a lot of discussion about the data, policy, and goals.
Based on the meetings, we set the scenario variables such as increasing products and changing transportation method from train to pipeline.
To apply the reviewer's comment, we added more explanation of target scenarios and the section 3.1 Application of scenarios. |
| Revision (Line 121-133) | The output in Khouribga is raw phosphate produced as either rock or slurry, the main component of manufactured phosphorous fertilizers. The transport of the phosphate (rocks and slurries) from Khouribga (mining area) to Jorf Lasfar (industrial production area) is a primary project in Morocco (OCP, 2016a). The demand for raw phosphate and the production and export of fertilizer and its products from Jorf Lasfar drive the upstream mining activity of Khouribga. In 2015, approximately 20.1 million tons of raw phosphate were excavated and transported to Jorf Lasfar; about 40% of this product was transported via pipeline as slurry and the balance via train as rock.
The pipeline from Khouribga to Jorf Lasfar is 187 km and ensures the continuous transport of phosphate from the Khouribga to Jorf Lasfar (Figure 1). As the plan was to increase phosphate production and phase out transport by train, tracks were replaced by pipeline that ensures the continuous flow of raw phosphate from the mining to the industrial area (OCP, 2016a). The plans impact regional water, energy, and food management: in particular, shifting from train to pipeline requires additional water to convert dry rock into liquid slurry. Shifting from train to pipeline changes the demand for water and energy resources at both the mining and the production locations. |
| Revision (Line 319-330) | 3.1 Application of scenarios
Increasing the exportable phosphate products and changing the transportation system from train to pipeline are considered top priorities for OCP group. Therefore, we assessed the impact of increased production by applying the scenarios (Table 5). Until recently, dried phosphate was transported by train from mining to manufacturing site, but, in the near future OCP group will use only pipeline transport. The change of from train to pipeline can affect not only direct energy or water consumption by transportation system but also that of the total supply chain in the mining site. Consequently, the production processes for slurry and for rock consume different quantities of water and energy, so that the mode of transport also becomes a scenario to allow quantification of their respective water and energy requirements.
Therefore, we applied the scenario about transportation system which indicates the only usage of pipeline. Table 4 showed the scenarios combining production and transportation. The first two scenarios are related to the 'business as usual (BAU)' scenario for production in 2015 but changing the transportation system from Khouribga to the terminal station at Jorf Lasfar. The other scenarios are related to the increase in the production. |

**Revision Notes**

Table 5 Scenarios through combination of production and transportation system

| Scenario | Phosphate production | Transportation of phosphate products | |
|---|---|---|---|
| | | by pipeline | by train |
| BAU | Production in 2015 | 40 % of total phosphate | 60 % of total phosphate |
| Scenario 01 | | 100% of total phosphate | None |
| Scenario 02 | 50% increase of phosphate export | 40 % of total phosphate | 60 % of total phosphate |
| Scenario 03 | | 100% of total phosphate | None |

**Comments 4**

| Reviewer's comments | 4. When calculating the WF, what was the considered period? Is it the calendar year or the growing period of crops? This needs clarification and the study boundaries should be well defined. If the authors considered the calendar year, they also need to assume that there was nothing planted from the previous year. Otherwise, in some months of the year, there will be an overlap between current crops and last year crops that should be considered. For wheat, for example, it is panted in November so it will be harvested in the next year. Is the water used in the next year will not be accounted for or is wheat considered a rainfed crop? |
|---|---|
| Response | We considered growing period of crops. For example, the irrigation water of wheat in this year means the sum of irrigation from Nov in last year to June in this year, as shown in the revised Table 4. Also, we added the more information of monthly irrigation requirement by crops through the new graphs (Figure 5).

Table 4 Crop planting and harvesting seasons, stage length and crop coefficients |

| Crop | Planting season | Harvesting season | Stage length (Days) | | | | | Crop coefficients | | |
|---|---|---|---|---|---|---|---|---|---|---|
| | | | Init. | Dev. | Mid | Late | Total | Kc init | Kc mid | Kc end |
| Olives | March | November* | 30 | 90 | 60 | 90 | 270 | 0.65 | 0.7 | 0.7 |
| Wheat | November | June* | 30 | 140 | 40 | 30 | 240 | 0.7 | 1.15 | 0.25 |
| Barley | March | July | 20 | 25 | 60 | 30 | 135 | 0.3 | 1.15 | 0.25 |
| Potato | Jan | April | 25 | 30 | 30 | 30 | 115 | 0.5 | 1.15 | 0.75 |

* Next year

**Revision Notes**

| | |
|---|---|
| Revision |
[Figure]

Figure 5 Monthly irrigation water requirement and rainfall in Khouribga |

**Comments 5**

| | |
|---|---|
| Reviewer's comments | 5. The authors need to improve the structure of their paper. The data, methodology, units and scenarios need to be defined before the results and discussion section. It is sometimes hard to differentiate the study assumptions and the defined scenarios from the results and discussion. |
| Response | We revised the methodology part with more details of site description and framework of WEF-P tool. First, we added more explanation of site, units, and footprints analysis in 2 Materials and methods. In addition, we added the comparison between WEF Nexus Tool 2.0 and WEF-P Tool in order to explain the details of framework of the tool.
In 3 Results and discussion, we made "3.1 Application of scenarios" to define the scenarios before representing the simulation results. |

**Revision Notes**

**Comments 1**

| | |
|---|---|
| Reviewer's comments | 1. The introduction and the article, in general, have a relatively limited literature review. This lack of references is, in my opinion, caused by two facts. First, the authors developed and used a tool while they didn't introduce anything concerning the tool creation, why it is needed? What is the difference between the WEF Nexus Tool 2.0 and the one created here and other frameworks and Nexus tools? What are the limitations of other tools in assessing what the current tool could assess? Second, the authors should make more references to other studies that used the WEF concept and compare their framework and findings. This will also be useful for the authors to place their findings in the context of other studies that applied the WEF Nexus in the paper's discussion. |
| Response | We appreciated your comments. In revision, we tried to represent why this study is important and what is the difference from previous research through more literature reviews in Introduction section. In addition, we emphasized contribution of this study in Conclusions section. Please find the additional paragraph as bellow.

WEF-P Tool referenced the concept of WEF Nexus Tool 2.0. However, the details of methodology are quite different. For example, the key methodology in WEF-P is supply chain analysis including materials, transportation, and resources. Thus, we add more explanation of the framework of WEF-P tool and novelty of this tool. |

[revised manuscript text omitted]

**Comments 2**

| Reviewer's comments | 2. Lines 47 - 62: WEF concept has also been recognized as a strong support to achieve the Sustainable Development Goals (SDGs) see Terrapon-Pfaff et al. (2018) for instance: Terrapon-Pfaff, J., Ortiz, W., Dienst, C., and Gröne, M.-C.: Energising the WEF nexus to enhance sustainable development at local level, Journal of environmental management, 223, 409-416, https://doi.org/10.1016/j.jenvman.2018.06.037, 2018. |
|---|---|
| Response | We added the linkages of the Nexus to SDGs in Introduction. |
| Revision (Line 49-58) | The demand for water, energy, and food, is expected to increase due to drivers such as population growth, economic development, urbanisation, and changing consumer habits (Terrapon-Pfaff et al., 2018). The interlinkages across key natural resource sectors and improving their production efficiency offer a win-win strategy for environmental sustainability, whether for current or future generations (Ringler et al., 2013). Accordingly, application of the Water-Energy-Food (WEF) nexus concept or approach is expected to make implementation of the Sustainable Development Goals (SDGs) more efficient and robust (Brandi et al., 2014; Yumkella and Yillia, 2015). The SDGs are classic examples of the necessity to acknowledge multidimensional, nexus interlinkages and trade-offs, particularly as governments are challenged to maximize benefits and invest limited resources. Infrastructure and capital are needed to achieve national SDG targets and the nexus concept is now used to highlight interdependencies between resources and the need for integrated, sustainable governance and management of those resources (Pahl-Wostl, 2019). |

**Comments 3**

| Reviewer's comments | 3. Line 66: You better refer to the world bank (2019) and not to the link in the text and in reference list refer to "World development indicators". |
|---|---|
| Response | We applied your comment and revised it. |

**Comments 4**

| Reviewer's comments | 4. Line 68: Same as the previous comment, refer to FAO (2015) in the text and the following in the reference list: FAO: FAOSTAT Online Database, Statistics Division, Food and Agriculture Organization of the United Nations (FAO), Rome, Italy, 2015. |
|---|---|
| Response | We applied your comment and revised it. |

**Revision Notes**

**Comments 5**

| Reviewer's comments | 5. Line 70: also, here refer to world bank 2019 and maybe use suffixes a and b following the publication year to make the difference between the references to world bank data you made. |
|---|---|
| Response | We applied your comment and revised it. |

**Comments 6**

| Reviewer's comments | 6. Line 71: maybe add a reference here? |
|---|---|
| Response | We applied your comment and revised it. |

**Comments 7**

| Reviewer's comments | 7. There are many sentences in the introduction that seem to be quotes from literature that are not cited. Check lines 74 - 84 for example. For instance, you should give a reference to the following: "especially in a country that imports nearly 90% of the energy it consumes". |
|---|---|
| Response | We checked the entire introduction and added some references. |
| Revision (Line 95-98) | Morocco uses recycling and reverse osmosis desalination to relieve some of the pressure on its fresh water resources and help secure the water necessary for phosphate production processes (OCP, 2016b). Each water source carries a distinct energy tag that must be accounted for, especially in a country that imports nearly 90% of its consumed energy (World Bank, 2019c). |
| Revision (Line 102-104) | As Morocco heads toward achieving its phosphate production goals, the ability to account for the resources associated with that achievement should be balanced with the associated (and increasing) agriculture and municipal demand projections: this is key to sustainable resource allocation (OCP, 2013). |

**Comments 8**

| Reviewer's comments | 8. Lines 97 - 98: "New water (grey, produced, brackish, and waste) is a resource with the potential to significantly contribute to bridging water and food gaps (Mohtar et al., 2015)" you make a difference between grey and wastewater and between produced and brackish water. What is the difference between grey and wastewater? What is the source of produced water then, are you referring to desalinisation? What is the share of this new water resources use in total water used in agriculture? How safe is the use of treated wastewater in agriculture in Morocco to consider its contribution as significant? |
|---|---|
| Response | The treated water in this study comes from the urban wastewater treatment plant, and it is used for washing phosphate in mining area. We were not able to check the water quality of treated water but several studies showed the application of urban wastewater treatment in agricultural area. |
| Revision (Line 137-143) | Additionally, OCP launched a plan to complete treatment plants for urban wastewater (capacity 5 million m³ yr-1) to be used for washing phosphate and industrial reuse in the mining area (OCP, 2016b). The phosphate mining area is encircled by cropland, whose water is also supplied from the dam. In this study, the authors consider the allocation of treated water to both the phosphate industry and agricultural irrigation (Tian et al, 2018). Both the mining and the agricultural activities of the region represent growing enterprises that place added pressure on available water resources, making the sustainable management of the water supply a hotspot to be considered in trade-off analyses.

Tian, Y., Ding, J., Zhu, D., and Morris, N.: The effect of the urban wastewater treatment ratio on agricultural water productivity: based on provincial data of China in 2004–2010. Applied Water Science, 8(5), 144, 2018 |

**Revision Notes**

**Comments 9**

| | |
|---|---|
| Reviewer's comments | 9. Lines 113 - 135: In your site description, it is not clear what is your system boundaries. Are you considering the whole region Khouribga? Or just the mining areas and it's surrounding? How is the surrounding defined for agriculture activity for instance? |
| Response | The tool first set the boundary in three mining site in Khouribga, and we added more explanation of sites. |
| Revision (Line 115-120) | The phosphate industry is controlled by the Office Cherifien des Phosphates (OCP) group in Morocco. OCP is that country's leading phosphate producer and accounts for 3% of the country's gross domestic product and about 20% of national exports in value over the course of the 20th century (Croset, 2012). The OCP group ran three mining fields: in central Morocco, near the city of Khouribga, and on the Gantour site. Khouribga, the largest mining area, includes three main sites from which raw phosphate is excavated and transported for chemical processing and fertilizer production: Sidi Chennane (SC), Merah Lahrach (MEA), and Bani Amir (BA) (Figure 1). |

**Comments 10**

| | |
|---|---|
| Reviewer's comments | 10. Line 117: "fields indicate 1.68 million tons of raw phosphate were excavated". What represents this number for the total country's production of raw phosphate? |
| Response | First, we changed the monthly production to yearly production, and added the paragraph indicating Khouribga is biggest mining area as below. |
| Revision (Line 124-127) | In 2015, approximately 20.1 million tons of raw phosphate were excavated, which was 58 % of total raw phosphate excavated in Morocco in 2018 (OCP, 2020), and transported to Jorf Lasfar; about 40% of this product was transported via pipeline as slurry and the balance via train as rock. |

**Comments 11**

| | |
|---|---|
| Reviewer's comments | 11. Line 128: "Plan Maroc Vert". It is maybe better to refer to the English name: the "Green Morocco" plan. Maybe add a reference to the Green plan or the national water plan that refers to moving from groundwater to surface water use. |
| Response | We added the English name and reference.
Stührenberg, L.: Plan Maroc Vert: les grands principes et avancées de la stratégie agricole marocaine. Bulletins de synthèse souveraineté alimentaire, 20, 2016. |

**Comments 12**

| | |
|---|---|
| Reviewer's comments | 12. Lines 160 - 161 "footprints are calculated using a regression function, or average value based on survey data, and technical experts in each process can modify this relation function as needed". This is not very clear. What regression function you used to calculate water and energy footprint? Please elaborate in this and explain what data comes from surveys and which technical experts you mean? |
| Response | Footprint indicates the amount of water or energy consumed per final products, which have various sub-processes in supply chain. Each process has a distinct footprint, identified as a regression function or average value from the technical (engineering) perspective. Based on the survey data, average electricity footprint (kWh/ton) can be estimated. The WEF-P Tool estimated the average value of the footprint and the function of the relationship between water-energy consumption and phosphate production using the historical data (in this study, year 2015). Technical experts in each process can modify the relation function once needed. |
| Revision (Line 227-237) | The main function of the WEF-P Tool is identification of the relationship between resources and production, and the quantification of the resources consumed in phosphate production. The methodology is based on life cycle assessment. The water and energy footprints were |

**Revision Notes**

| | |
|---|---|
| | analysed, indicating the quantity of water or energy consumed in various sub-processes in the supply chain's integration of production and transportation. The technical details of each process are specific and aggregated into functional processes. The main component is the footprint, which indicates the water and energy requirements for phosphate products, and the $CO_2$ emitted through energy consumption. Each process has a specific footprint based on field data and fed into the tool monthly, or when a significant change in capacity of the functional processes has occurred. For all footprint processes in Khouribga, the amount of raw phosphate is measured in commercial metric tons embedded in slurries and rock. Even if the phosphate rock changes to slurry through several processes, the amount of raw phosphate embedded in products is not changed. Thus, the tons of phosphate in water and energy footprints indicate the raw phosphate embedded in the products in each process and is constant through entire supply chains. |
| Revision (Line 335-339) | To quantify the water, energy, and $CO_2$ emissions, water and energy footprints of each process in each mining site were analysed based on survey data. For example, the adaptation process is essential for pipeline transportation and large amounts water are needed in comparison to other processes, thus the relationship between the amounts of phosphate and water used in adaptation process were analysed (Figure 4 (a)). In addition, energy footprint includes electricity and fuel consumption; analysed through the linear relationship (Figure 4 (b)).

[Figure]

[Figure]

 (a) Water footprint in an adaptation process in MEA      (b) Energy footprint in a washing process in MEA
 **Figure 4.** Water and energy footprints in MEA based on the BAU data-base |

**Comments 13**

| | |
|---|---|
| Reviewer's comments | 13. Line 206: You better refer to FAO report 46. |
| Response | We added the reference of FAO report 46.

 Smith, M.: CROPWAT-A computer program for irrigation planning and management. FAO Irrigation and Drainage Paper No. 46. FAO, Rome, 1992. |

**Comments 14**

| | |
|---|---|
| Reviewer's comments | 14. What is the methodology you used for accounting the water footprint of crop production? |
| Response | Irrigation water requirement was calculated using CropWat model, and not only evapotranspiration but runoff of rainfall was applied as well. We used the reference methodology (USDA SCS method) from CropWat explained in FAO No. 46 report.
 Thus, we added more explanation about irrigation requirement modeling based on ETc and runoff that is provided in CropWat model. Please find the addition explanation as below. |
| Revision (Line 307-314) | Irrigation water requirement was calculated by ETc and effective precipitation, as shown in Eq. (10). The effective precipitation indicated the precipitation except for runoff, and was calculated using the USDA Soil Conservation Service method (Eq. 11) (Smith, 1992). |

**Revision Notes**

$$IRReq = ET_c - P_{eff} \qquad\qquad (10)$$
$$P_{eff} = P_{tot}\,(125 - 0.2\,P_{tot})/\,125 \qquad \text{for } P_{tot} < 250\ mm \qquad (11)$$
$$P_{eff} = 125 + 0.1\,P_{tot} \qquad\qquad \text{for } P_{tot} > 250\ mm$$

where $IRReq$ is irrigation water requirement, ETc is the crop evapotranspiration, $P_{eff}$ is effective precipitation, and $P_{tot}$ is total precipitation.

**Comments 15**

| Reviewer's comments | 15. Lines 231 - 232: Is it possible to include the unit of phosphate production somewhere in the materials and methods section? |
|---|---|
| Response | The technical details of each process are specific and aggregated into more functional processes. For all processes and transportation systems in Khouribga site, we applied the commercial metric tons as the unit, and we added the explanation of it. |
| Revision (Line 233-237) | For all footprint processes in Khouribga, the amount of raw phosphate is measured in commercial metric tons embedded in slurries and rock. Even if the phosphate rock changes to slurry through several processes, the amount of raw phosphate embedded in products is not changed. Thus, the tons of phosphate in water and energy footprints indicate the raw phosphate embedded in the products in each process and is constant through entire supply chains. |

**Comments 16**

| Reviewer's comments | 16. The following sentences give the same information: "In 2015, 1.68 million tons of raw phosphate was mined and transported from the mining to the manufacturing area, monthly" and "In the mining area, 20.1 million tons of raw phosphate were produced in the 2015". The only difference is monthly or yearly production. Can you combine the information about target production and the BAU scenario to avoid repetitions? |
|---|---|
| Response | We changed monthly to yearly production. |

**Comments 17-18**

| Reviewer's comments | 17. Lines 266 – 267: Are the considered crops the only produced crops in that area? Why setting the target production to exactly 0.1%? Is that the potential production of the considered area? Is that the target production for each crop?
 18. Food production is not only crop production but also livestock production. Since you are not including this aspect of food production in your study you need to maybe spend one sentence in the limitation of the paper or somewhere to make this clear. |
|---|---|
| Response | We admit that is limitation of this study. We do not have exact data of agricultural area near mining area in Khouribga. First, we checked the agricultural area using MODIS-based global land cover data, found a lot of crop area near by Khouribga. However, we were not able to collect more data about exact area, crops, and irrigation system.
 Thus, we constructed the tool to be able to adapt the agricultural area as user scenarios. In other words, we assumed the agricultural area near by mining area and set the target production as user scenario instead of setting the agricultural boundary. It could be limitation in terms of feasibility but this tool is decision support system, thus it can provide results with various situation of agricultural production plans. |

**Comments 19**

| Reviewer's comments | 19. Line 274: The waste-water treatment plant capacity seems to be taken from somewhere, maybe add a reference? |
|---|---|
| Response | We added the reference. |
| Revision (Line 137-139) | Additionally, OCP launched a plan to complete treatment plants for urban wastewater (capacity 5 million m³ yr-1) to be used for washing phosphate and industrial reuse in the mining area (OCP, 2016b). |

**Revision Notes**

**Comments 20**

| Reviewer's comments | 20. In figure 3, the amount of rainfall in March seems to exceed rainfall in November and December but irrigation is still needed in March. You better include the harvested date next to the planting date in Table 1 to give an idea of how many crops have their growing period in March. |
|---|---|
| Response | We considered growing period of crops. For example, the irrigation water of wheat in this year means the sum of irrigation from Nov in last year to June in this year, as shown in the revised Table 4. Also, we added the more information of monthly irrigation requirement by crops through the new graphs. |
| Revision | Table 4 Crop planting and harvesting seasons, stage length and crop coefficients |

Table 4 Crop planting and harvesting seasons, stage length and crop coefficients

| Crop | Planting season | Harvesting season | Init. | Dev. | Mid | Late | Total | Kc init | Kc mid | Kc end |
|---|---|---|---|---|---|---|---|---|---|---|
| Olives | March | November* | 30 | 90 | 60 | 90 | 270 | 0.65 | 0.7 | 0.7 |
| Wheat | November | June* | 30 | 140 | 40 | 30 | 240 | 0.7 | 1.15 | 0.25 |
| Barley | March | July | 20 | 25 | 60 | 30 | 135 | 0.3 | 1.15 | 0.25 |
| Potato | Jan | April | 25 | 30 | 30 | 30 | 115 | 0.5 | 1.15 | 0.75 |

* Next year

**Comments 21**

| Reviewer's comments | 21. The first sentences in Line 303 and line 316 seem to be the same. |
|---|---|
| Response | We revised it. |

**Comments 22**

| Reviewer's comments | 22. Line 327 – 331: Dynamic phosphate production contributes to electricity savings in only 6 months of the year (from May till October). However, in the rest of the year, the consumption of electricity in the dynamic phosphate production was higher than the static production. |
|---|---|
| Response | We considered more phosphate production during drought season from May to Oct., and less phosphate production during rainy season from Nov. to April. However, we found that total energy use in a year in dynamic production scenario is less than static production scenario because of groundwater use saving. |

**Revision Notes**

**Minor Comments (Reviewer #1)**

**Comments 1**

| Reviewer's comments | 1. Line 64 and Line 74: You miss a comma (Taleb, 2006) and (OCP, 2013). These are just examples; you need to check all your references and format them according to HESS guidelines. |
|---|---|
| Response | We revised them and checked all references. |

**Comments 2**

| Reviewer's comments | 2. Line 91, energy is repeated twice. |
|---|---|
| Response | We revised it. |

**Comments 3**

| Reviewer's comments | 3. Table 1. You better remove the reference from the table's title and insert it in Line 225: "FAO provides crop coefficients for each stage". Alternatively, you can add it as a note under the table. |
|---|---|
| Response | We revised the table and inserted the reference in the manuscript. |

**Comments 4**

| Reviewer's comments | 4. In Table 1's title "Information" doesn't need to be capitalized. |
|---|---|
| Response | We revised it to "Table 4 Crop planting and harvesting seasons, stage length and crop coefficients". |

**Comments 5**

| Reviewer's comments | 5. In Table 1: Plant data: do you mean Planting date? |
|---|---|
| Response | Yes, we changed it to planting data |

**Comments 6**

| Reviewer's comments | 6. Line 233: remove the additional backslash –> tons months-1 |
|---|---|
| Response | We removed it. |

**Revision Notes**

| | |
|---|---|
| Reviewer's comments | 7. Figure 4: Include more description in the figure's title and reduce the text in the figure's legend. Try to include the legend in the figure as it seems now to be outside.

8. Same for figure 5: Groundwater use is already in the figure's title. The legend could be just: Dynamic production and Static production.

9. Same for figure 6. |
| Response | We changed the legends in Figures 4-6 |
| Revision |
[Figure]

Figure 6 Monthly water supply from Aït Messaoud Dam

Figure 7 Monthly ground water use by static and dynamic production of phosphate slurries transported by pipeline |

**Revision Notes**

[Figure]

Figure 8 Monthly electricity consumption for supplying water by static and dynamic production of phosphate slurries transported by pipeline

**Comments 10**

| Reviewer's comments | 10. There are some small typographical errors throughout the paper and these should be corrected. |
|---|---|
| Response | We checked the entire manuscript, and corrected typo. |

---

## Author Comment (AC2) · 10 Aug 2020

Dear Editor and Reviewers

Thank you for considering the manuscript for publication in the Hydrology and Earth System Sciences (HESS) and in-depth review of the manuscript. In this study, we developed the WEF-P tool which is a decision support system for linking phosphate industry to agriculture in terms of water-energy nexus perspective. In particular, we adapted the supply chain analysis to quantify the water and energy footprints and assessed the impacts of water allocation between industry and agriculture through dynamic production of phosphate using the WEF-P Tool.

[Figure]

The main comments from reviewers were related to 1) lack of literature reviews, 2) strength of this tool in comparison to others, and 3) economic and environmental impacts assessment. Therefore, in the revised manuscript, we revised the introduction with more literature reviews and reorganized the structure of our manuscript in order to improve its readability and highlight the novelty of the present work. In particular, we have detailed explanation about methodology of the tool, data survey, scenarios, and footprints modeling. In addition, we compared this WEF-P Tool with WEF Nexus 2.0, and added the limitation of economic and environmental impacts assessment through the WEF-P Tool.

Main revisions - Revising introduction with more literature reviews - Reconstructing and revising the materials and methods section - Adding limitations of economic and environmental impacts assessment

In the revision notes, you will find a point-by-point reply to specific comments.

We appreciate again your thoughtful comments and look forward to hearing your reply.

Kind regards, on behalf of all co-authors,

Please also note the supplement to this comment:
https://hess.copernicus.org/preprints/hess-2020-197/hess-2020-197-AC2-supplement.pdf

**Supplement:**

**Revision Notes**

**Reply to the reviewers for hess-2020-197**

Title: A Water-Energy-Food Nexus Approach for Conducting Trade-off Analysis: Morocco's Phosphate Industry in the Khouribga Region

Authors: Sang-Hyun Lee, Amjad T. Assi, Bassel T. Daher, Fatima E. Mengoub, Rabi H. Mohtar

Dear Editor and Reviewers

Thank you for considering the manuscript for publication in the **Hydrology and Earth System Sciences (HESS)** and in-depth review of the manuscript. In this study, we developed the WEF-P tool which is a decision support system for linking phosphate industry to agriculture in terms of water-energy nexus perspective. In particular, we adapted the supply chain analysis to quantify the water and energy footprints and assessed the impacts of water allocation between industry and agriculture through dynamic production of phosphate using the WEF-P Tool.

The main comments from reviewers were related to 1) lack of literature reviews, 2) strength of this tool in comparison to others, and 3) economic and environmental impacts assessment. Therefore, in the revised manuscript, we revised the introduction with more literature reviews and reorganized the structure of our manuscript in order to improve its readability and highlight the novelty of the present work. In particular, we have detailed explanation about methodology of the tool, data survey, scenarios, and footprints modeling. In addition, we compared this WEF-P Tool with WEF Nexus 2.0, and added the limitation of economic and environmental impacts assessment through the WEF-P Tool.

Main revisions
- Revising introduction with more literature reviews
- Reconstructing and revising the materials and methods section
- Adding limitations of economic and environmental impacts assessment

In the revision notes, you will find a point-by-point reply to specific comments.

We appreciate again your thoughtful comments and look forward to hearing your reply.

Kind regards, on behalf of all co-authors,
* * *
Sang-Hyun Lee

**Revision Notes**

**Major Comments (Reviewer #2)**

**Comments 1**

| | |
|---|---|
| Reviewer's comments | 1- The author didn't give a clear idea about the water footprint and energy footprint and how they contribute in the assessment scenarios, did they matter when it comes to producing a clean food? And concerning the Carbone footprint, is the Carbone show a real danger in front of all the radioactive components of phosphate? |
| Response | Footprint indicates the amount of water or energy consumed per final products, which have various sub-processes in supply chain. Each process has a distinct footprint, identified as a regression function or average value from the technical (engineering) perspective. Based on the survey data, average electricity footprint (kWh/ton) can be estimated. The WEF-P Tool estimated the average value of the footprint and the function of the relationship between water-energy consumption and phosphate production using the historical data (in this study, year 2015). Technical experts in each process can modify the relation function once needed.

To calculate $CO_2$ footprint, we need to consider various factors and complex relationship. In particular, $CO_2$ emission has not been measured in mining area, and $CO_2$ emission from crop area is another level of research. Therefore, in this study, we limited $CO_2$ emitted by fuel energy use (direct emission) and electricity generation (indirect emission) and applied the reference $CO_2$ footprint. |
| Revision (Line 227-237) | The main function of the WEF-P Tool is identification of the relationship between resources and production, and the quantification of the resources consumed in phosphate production. The methodology is based on life cycle assessment. The water and energy footprints were analysed, indicating the quantity of water or energy consumed in various sub-processes in the supply chain's integration of production and transportation. The technical details of each process are specific and aggregated into functional processes. The main component is the footprint, which indicates the water and energy requirements for phosphate products, and the $CO_2$ emitted through energy consumption. Each process has a specific footprint based on field data and fed into the tool monthly, or when a significant change in capacity of the functional processes has occurred. For all footprint processes in Khouribga, the amount of raw phosphate is measured in commercial metric tons embedded in slurries and rock. Even if the phosphate rock changes to slurry through several processes, the amount of raw phosphate embedded in products is not changed. Thus, the tons of phosphate in water and energy footprints indicate the raw phosphate embedded in the products in each process and is constant through entire supply chains. |
| Revision (Line 263-267) | Although real emission in each process in supply chain should be measured, this study is limited measuring $CO_2$ emission in mining area. In addition, $CO_2$ emission in crop area is related to soil and crops, and it is another level of research. Thus, we limited $CO_2$ emission to that emitted by fuel energy use by machinery (direct emission) and electricity generation in power plants (indirect emission), and the reference $CO_2$ footprints were applied (Table 2).

**Table 2** $CO_2$ emission by burning fuels and generating electricity

_See table below_ |

**Table 2** $CO_2$ emission by burning fuels and generating electricity

| $CO_2$ emission by burning fuel | | $CO_2$ emission by generating electricity | | | |
|---|---|---|---|---|---|
| Sources | $CO_2$ emission[1] (kg of $CO_2$ $L^{-1}$) | Sources | $CO_2$ emission by sources[1] (ton of $CO_2$ $10^{-6}$ kWh) | Proportion of sources in Morocco[2] (%) | $CO_2$ emission (ton of $CO_2$ $10^{-6}$ kWh) |
| Gasoline | 2.59 | Coal | 1,026 | 43.4% | |
| Diesel | 2.96 | Petroleum | 1,026 | 25.3% | |
| | | Natural gas | 504 | 22.7% | 820.9 |
| | | Hydroelectricity | 19.7 | 6.9% | |
| | | Renewables | 15.8 | 1.7% | |

[1] U.S. Energy Information Administration (https://www.eia.gov)
[2] International Energy Agency, 2014.

**Revision Notes**

**Comments 2**

| Reviewer's comments | 2- The author applied the tool considering the phosphate as a simple nexus component like water or energy. Otherwise, there is a need to emphasize all environmental and economic aspects related to this product and the interaction between agricultural and phosphate production in the study area, to consider that this tool is efficient and can be considered as the best decision-making tool when it comes to this type of nexus. |
|---|---|
| Response | Some of phosphates are exported but a lot of them are transported to Jorf Lasfar and used as raw materials for phosphorous fertilizers. Thus, the economic value of phosphate could be changed by the types of fertilizers, and it is actually difficult to apply the static economic value to the model. In addition, still there are a lot of discussion about water value are ongoing. Thus, we added more explanation why we did not mention the economic perspective in this study. |
| Revision (Line 180-194) | However, the WEF-P Tool has limitations in assessing economic impacts such as cost and benefit analysis. This is because cost must include the price of water, which is still under discussion, and the price of products when analysing their benefits. Raw phosphate is transported to the manufacturing area and used in the production of various fertilizers that have different prices: this makes it difficult to set the price of excavating raw phosphate in the mining area. Sustainability assessment also has qualitative aspects in terms of environmental impact. The WEF Nexus Tool 2.0 applied the sustainability index based on resource capacity and availability, however, it is still a quantitative aspect. We should consider the meaning and definition of sustainability, both quantitatively and qualitatively, and then assess the index using the stakeholders' weights for the variables related to sustainability. Additionally, spatial and temporal scales should be included in a sustainability index. For example, the pipeline transportation system requires water, which is transported with products: the pipeline causes greater water use at the origin, but also provides additional water to the destination area. Also, the water requirement differs with temporal season, such as the water intensive agricultural production season. Thus, more research is needed for a sustainability assessment based on economic and environmental impact. However, the quantitative analysis is an essential factor for assessing sustainability, therefore, the WEF-P Tool focuses on quantification of 1) water and requirements for phosphate production and transportation, 2) carbon emissions by energy used in product processes, 3) water supply system and transportation, and 4) dynamic production impacts on water and energy savings. |

**Comments 3**

| Reviewer's comments | 3- The paper is not well structured, there is a gap between different sections in the paper. The author didn't explain the choice of the scenarios and the methodology of data collection. |
|---|---|
| Response | We revised the methodology part with more details of site description and framework of WEF-P tool. First, we added more explanation of site, units, and footprints analysis in 2 Materials and methods. In addition, we added the comparison between WEF Nexus Tool 2.0 and WEF-P Tool in order to explain the details of framework of the tool. In addition, we re-constructed the methodology section with more details of site description and overall framework of WEF-P Tool. In 3 Results and discussion, we made "3.1 Application of scenarios" to define the scenarios before representing the simulation results. |

**Revision Notes**

**Comments 4**

| | |
|---|---|
| Reviewer's comments | 4- In the discussion part, no scientific comparison, even if the tool developed by the author, it is important to compare the findings, especially in the WEF nexus impact on water and energy footprint and CO2 emission. |
| Response | We agreed with your comment. The main purpose of this study is to develop the tool and apply it to link the industry and agriculture in the context of regional water-energy boundary. However, we assumed some scenarios about agricultural production and also focused on quantitative assessment such as water and energy requirement.

Thus, it is limited to assess the economic and environmental impacts through the current tool. However, this tool is able to be improved with more case data and field survey, thus this tool is useful as the platform adapting the scientific research and policy of industry and agriculture.

We added some founding and contribution of this study briefly in Conclusions section. Please understand these limitations. |
| Revision (Line 459-468) | In other words, the WEF-P Tool offers a decision support system to provide quantifiable trade-off analyses for management decisions such as increasing production, transportation systems, and water allocation. The developed WEF-P Tool enables users to:
• understand and identify the associated footprints of the primary functional production processes and existing flows in production lines;
• identify the main sources of data to be gathered and fed into the model on a specific temporal basis;
• identify the techniques employed to conserve or produce water and energy and minimize the impacts of phosphate production;
• form a translational platform between sectors and stakeholders to evaluate proposed scenarios and their associated resource requirements |
| Revision (Line 489-495) | Beyond the limitations, the deliverables from this study include a conceptual and analytical model of the phosphate supply chain in Morocco, the WEF-P Tool. The Tool can assess the various scenarios to offer an effective means of ensuring the sustainable management of limited resources to both agricultural area and phosphate industry. It quantifies the products (phosphate) and resource footprints (water, energy) across the supply chain; identifies the interlinkages between water and energy in phosphate production and transport, and establishes reference values for comparison of outcomes and performance. The WEF-P Tool enables the user to evaluate trade-offs between water resource allocations and the impact of the Moroccan phosphate industry with agricultural water use. |

**Revision Notes**

**Specific Comments**

**Comments 1**

| | |
|---|---|
| Reviewer's comments | 1- The introduction is too general and missing a good literature review. The author didn't talk about the novelty of the use of the WEF-P tool and the difference between it and the WEF tool (http://www.wefnexustool.org). There is also a need to emphasize the pros and cons of the used tool, especially that phosphate production has a very interesting economic value but a very bad environmental impact, and here comes the importance of the concept of the sustainability index existing in the WEF tool 2.0. |
| Response | We appreciated your comments.
 In revision, we tried to represent why this study is important and what is the difference from previous research through more literature reviews in Introduction section.
 However, as we mentioned in Major comment, there are limitations of economic and environmental impacts assessment through this tool. We explained these limitations in revision. |
| Revision (Line 71-83) | The nexus framework is dependent on the stakeholders, scales of boundary, and analytical tools. In considering the application of the nexus as a platform, an integrated modelling approach is essential. These issues manifest in very different ways across each sector, but their impacts are often closely related in terms of trade-offs. In particular, the sub-nexus needs to be effectively conceptualized and a theoretical sub-nexus developed. Private-sector water, energy, and food supply chain players are the key stakeholders to address current contradictions that arising as a consequence of attempts to develop a grand nexus approach (Allan et al., 2015). Accordingly, we must consider the "specialized" nexus of multi-stakeholders, such as agriculture, industry and urban areas, for which water, energy and food are treated as subsystems. Current nexus frameworks often focus on macro-level drivers of resource consumption patterns (Biggs et al., 2015), but major nexus challenges are faced at local levels (Terrapon-Pfaff et al., 2018). Thus, 'larger scale' extraction and consumption of natural resources may lead to depletion of natural capital stocks and increased climate risk with no equitable share of the benefits (Hoff, 2011; Rockström et al., 2009). Al-Saidi and Elagib (2017) showed the importance of exploring driving forces and interactions at different scales in the conceptual development of the nexus, emphasizing more case-study based recommendations in the reality of institutions, bureaucracies, and environmental stakeholders. |
| Response | However, when we develop this tool, we contacted the managers and engineers working in the OCP group and OCP policy center, and had a lot of discussion about the data, policy, and goals. Based on the meetings, we set the scenario variables such as increasing products and changing transportation method from train to pipeline. |
| Revision (Line 172-179) | Throughout the tool development process, the supply chain was verified with OCP and the OCP Policy Center in various ways: (i) during the data collection phase, through meetings with the OCP steering committee, financial managers, technical managers and engineers; and (ii) through follow ups with OCP Policy Center team (conference calls and email). The OCP Policy Center team shared with WEF Nexus Team their main concerns regarding the tool structure, based on input from the OCP technical team. The WEF Nexus Team used these shared concerns in their considerations of revisions to the tool structure and associated excel spreadsheets of the model. Specifically, the major aggregated processes and lines of productions were revised and identified in a functional supply chain to maximize the abilities and flexibilities of the model and ensure efficacy of the available data base for processes and production lines. |
| Response | The WEF-P Tool referenced the concept of WEF Nexus Tool 2.0. However, the details of methodology are quite different. For example, the key methodology in WEF-P is supply chain analysis including materials, transportation, and resources. Thus, we add more explanation of the framework of WEF-P tool and novelty of this tool. |

**Revision Notes**

The developed WEF-P Tool, adapted from the WEF Nexus Tool 2.0 (Daher and Mohtar, 2015), considers the supply chain of final product in terms of its resource consumption, including the set of processes that pass materials forward (La Londe and Masters, 1994; Mentzer et al., 2001), and various organizations or individuals directly involved in the flow of products (Mentzer et al., 2001). It assesses the impact of various scenarios and possible responses to regional resource management needs. Table 1 shows the differences between WEF Nexus Tool 2.0 and WEF-P Tool in the context of variables, scenarios, analytical tools, and quantitative assessments.

Table 1 Comparison between WEF Nexus Tool 2.0 abd WEF-P Tool

|  | WEF Nexus Tool 2.0 | WEF-P Tool |
|---|---|---|
| Variables and scenarios | • Self-sufficiency of produced crops
• Type of agricultural production
• Sources of water (groundwater, surface water, treated water and so on)
• Sources of energy (natural gas, diesel, solar, wind and so on)
• Trade portfolio (countries of import and amounts per country) | • Static and dynamic phosphate production
• Transportation modes (train and pipeline)
• Sources of water (groundwater, surface water, treated water and so on)
• Water allocation between industry and agriculture |
| Analytical tool | • Food product base analysis
• Food-centric interlinkages among water, energy, and food
• Water and energy footprint based on product (ex. water footprint of crops) | • Process base analysis
• Phosphate-centric interlinkages among production, transportation, and resource allocation
• Water and energy footprint based on processes (ex. water footprint in washing process) |
| Quantitative assessment | • Water requirement for energy and agricultural production
• Energy requirement for agricultural and water production
• Land footprint for agricultural and energy production
• Carbon emissions from energy used for water and food production
• Financial cost | • Water and requirement for phosphate production and transportation
• Carbon emission by energy used in product processes, water supply system and transportation
• Dynamic production impacts on water and energy savings |

**Revision Notes**

**Comments 2**

| Reviewer's comments | 2- Line 65: it is better to use the actual information as 2019 for the population growth and avoid using hyperlinks in the text as reference the same in the lines 68 and 70 |
|---|---|
| Response | We applied your comment and revised it. |

**Comments 3**

| Reviewer's comments | 3- Line 76 to 80: the author needs to add a reference for the quote |
|---|---|
| Response | We applied your comment and revised it. |

**Comments 4**

| Reviewer's comments | 4- Part 2.1: from where came the differentiation of the different types of water and from which background the new water concept is coming? and is the wastewater used is coming from phosphate washing? If yes, are you sure that it is safe? |
|---|---|
| Response | In addition, OCP launched a plan to complete treatment plants for urban wastewater, primarily for industrial reuse in the mining area (capacity 5 million m³ yr-1), allowing using for washing phosphate (OCP, 2016b). The phosphate mining area is encircled by cropland, whose water is also supplied from the dam. In this study, we considered the allocation of treated water to phosphate industry and agriculture irrigation and Tian et al. (2018) showed the usage of treated wastewater from urban area for agriculture irrigation.. |

**Comments 5**

| Reviewer's comments | 5- Figure 1: the description of the figure is missing. Otherwise, there are two missing key concepts need to be considered: sustainability index and environmental index (related mainly with the Phosphate toxicity) |
|---|---|
| Response | WEF-P in this study focused on the estimation of water and energy based on supply chain analysis but the economic and environmental impacts assessment was not included. |
| | In addition, sustainability has a lot for meaning itself, and it is related to qualitative assessment, thus it is difficult to define what is sustainability. In previous version, we considered the availability as sustainability index. The availability index is calculated using maximum capacity and current consumption. For example, a large quantity of water available indicates 'available water', while a negative value of water availability indicates that water use has exceeded maximum capacity. |
| | However, at this version of tool, we focused on quantification of resource saving and put the availability and sustainability index to next version. It is limitation of this tool, and we mention this limitation in Conclusion. |
| | We revised the entire section "2.2.1 Overall Framework of WEF-P Tool" |
| Revision (Line 148-194) | 2.2.1 Overall Framework of WEF-P Tool |
| | The developed WEF-P Tool, adapted from the WEF Nexus Tool 2.0 (Daher and Mohtar, 2015), considers the supply chain of final product in terms of its resource consumption, including the set of processes that pass materials forward (La Londe and Masters, 1994; Mentzer et al., 2001), and various organizations or individuals directly involved in the flow of products (Mentzer et al., 2001). It assesses the impact of various scenarios and possible responses to regional resource management needs. Table 1 shows the differences between WEF Nexus Tool 2.0 and WEF-P Tool in the context of variables, scenarios, analytical tools, |

and quantitative assessments.

[revised manuscript text omitted]

**Revision Notes**

**Comments 6**

| | |
|---|---|
| Reviewer's comments | 6- In the site description: it will be good to have a general idea of the study area (location, climatic conditions: rainfall, temperature, wind, radiation, water resources, soil resources, agricultural activity, energy source) since the author involved the evapotranspiration calculation and water and energy footprints. |
| Response | We added the table of climate information. |
| Revision | **Table 2** Climate information in Khouribga |

**Table 2** Climate information in Khouribga

| Month | Precipitation (mm m$^{-1}$) | Temperature min. (°C). | Temperature max. (°C) | Relative humidity (%) | Sunshine hours (h d$^{-1}$) |
|---|---|---|---|---|---|
| Jan | 56 | 3.8 | 17.3 | 72 | 5.6 |
| Feb | 65 | 5 | 19 | 76 | 5.7 |
| Mar | 94 | 7.2 | 21.8 | 69 | 6.4 |
| Apr | 70 | 9.5 | 25.3 | 67 | 7.4 |
| May | 32 | 12.5 | 29.3 | 55 | 8.8 |
| Jun | 9 | 16.6 | 34.5 | 48 | 9.8 |
| Jul | 2 | 19.8 | 39.7 | 39 | 10.9 |
| Aug | 7 | 20 | 39.6 | 37 | 10.3 |
| Sep | 12 | 17.5 | 34.5 | 47 | 9.1 |
| Oct | 27 | 13.5 | 29 | 58 | 7.6 |
| Nov | 71 | 8.8 | 22 | 70 | 5.2 |
| Dec | 81 | 5.1 | 18.6 | 71 | 5.5 |

**Comments 7**

| | |
|---|---|
| Reviewer's comments | 7- Part 2.2.2 the author underlined mainly the different steps of phosphate production and missed a good explanation of the footprint calculation and the data gathering methodology and date frame of the collected data. For the CO2 emission, the author linked it only with energy use, but he forgot to mention the importance of having a healthy soil can play a crucial role in carbon sequestration. |
| Response | We agreed there are limitations in calculation of CO2 emission. To calculate $CO_2$ emission, we need to measure real emission in each process in supply chain but it was limited to measure $CO_2$ emission in mining area. In addition, $CO_2$ emission in crop area is related to soil and crops, and it is another level of research. Thus, we limited $CO_2$ emission which is emitted by fuel energy use by machinery (direct emission) and electricity generation in power plants (indirect emission), and the reference $CO_2$ footprints were applied, as shown in Table 2. |
| Revision | **Table 2** $CO_2$ emission by burning fuels and generating electricity |

**Table 2** $CO_2$ emission by burning fuels and generating electricity

| CO$_2$ emission by burning fuel | | CO$_2$ emission by generating electricity | | | |
|---|---|---|---|---|---|
| Sources | CO$_2$ emission[1] (kg of CO$_2$ L$^{-1}$) | Sources | CO$_2$ emission by sources[1] (ton of CO$_2$ 10$^{-6}$ kWh) | Proportion of sources in Morocco[2] (%) | CO$_2$ emission (ton of CO$_2$ 10$^{-6}$ kWh) |
| Gasoline | 2.59 | Coal | 1,026 | 43.4% | |
| Diesel | 2.96 | Petroleum | 1,026 | 25.3% | |
| | | Natural gas | 504 | 22.7% | 820.9 |
| | | Hydroelectricity | 19.7 | 6.9% | |
| | | Renewables | 15.8 | 1.7% | |

[1] U.S. Energy Information Administration (https://www.eia.gov)
[2] International Energy Agency, 2014.

**Revision Notes**

**Comments 8**

| | |
|---|---|
| Reviewer's comments | 8- Part 2.3: the author used the ETP requirement to calculate the irrigation water requirement which needs to be revised and does the used data in this calculation are reflecting the exact situation of the study area? |
| Response | Irrigation water requirement was calculated using CropWat model, and not only evapotranspiration but runoff of rainfall was applied as well. We used the reference methodology (USDA SCS method) from CropWat explained in FAO No. 46 report. Thus, we added more explanation about irrigation requirement modeling based on ETc and runoff that is provided in CropWat model. Please find the addition explanation as below.

We do not have exact data of agricultural area near mining area in Khouribga. First, we checked the agricultural area using MODIS-based global land cover data, found a lot of crop area near by Khouribga. However, we were not able to collect more data about exact area, crops, and irrigation system.
Thus, we constructed the tool to be able to adapt the agricultural area as user scenarios. In other words, we assumed the agricultural area near by mining area and set the target production as user scenario instead of setting the agricultural boundary. It could be limitation in terms of feasibility but this tool is decision support system, thus it can provide results with various situation of agricultural production plans. |
| Revision (Line 307-314) | Irrigation water requirement was calculated by ETc and effective precipitation, as shown in Eq. (10). The effective precipitation indicated the precipitation except for runoff, and was calculated using the USDA Soil Conservation Service method (Eq. 11) (Smith, 1992).

$$IRReq = ET_c - P_{eff} \qquad\qquad\qquad (10)$$
$$P_{eff} = P_{tot}\,(125 - 0.2\,P_{tot})/\,125 \qquad \text{for } P_{tot} < 250\ mm \qquad (11)$$
$$P_{eff} = 125 + 0.1\,P_{tot} \qquad\qquad \text{for } P_{tot} > 250\ mm$$

where $IRReq$ is irrigation water requirement, ETc is the crop evapotranspiration, $P_{eff}$ is effective precipitation, and $P_{tot}$ is total precipitation. |

**Revision Notes**

**Comments 9**

| Reviewer's comments | 9- Table 1: do you mean by plan data the plantation season or the date of data collection? |
|---|---|
| Response | It indicates the plantation season |

**Comments 10**

| Reviewer's comments | 10- For the results and discussion, the choice of the scenarios should be clarified before |
|---|---|
| Response | To apply the reviewer's comment, we added more explanation of target scenarios and the section 3.1 Application of scenarios. |
| Revision (Line 319-330) | 3.1 Application of scenarios
Increasing the exportable phosphate products and changing the transportation system from train to pipeline are considered top priorities for OCP group. Therefore, we assessed the impact of increased production by applying the scenarios (Table 5). Until recently, dried phosphate was transported by train from mining to manufacturing site, but, in the near future OCP group will use only pipeline transport. The change of from train to pipeline can affect not only direct energy or water consumption by transportation system but also that of the total supply chain in the mining site. Consequently, the production processes for slurry and for rock consume different quantities of water and energy, so that the mode of transport also becomes a scenario to allow quantification of their respective water and energy requirements.
Therefore, we applied the scenario about transportation system which indicates the only usage of pipeline. Table 4 showed the scenarios combining production and transportation. The first two scenarios are related to the 'business as usual (BAU)' scenario for production in 2015 but changing the transportation system from Khouribga to the terminal station at Jorf Lasfar. The other scenarios are related to the increase in the production.

Table 5 Scenarios through combination of production and transportation system

[TABLE BELOW] |

Table 5 Scenarios through combination of production and transportation system

| Scenario | Phosphate production | Transportation of phosphate products | |
|---|---|---|---|
| | | by pipeline | by train |
| BAU | Production in 2015 | 40 % of total phosphate | 60 % of total phosphate |
| Scenario 01 | | 100% of total phosphate | None |
| Scenario 02 | 50% increase of phosphate export | 40 % of total phosphate | 60 % of total phosphate |
| Scenario 03 | | 100% of total phosphate | None |

**Revision Notes**

**Comments 11**

| Reviewer's comments | 11- Figure 3: it will be better if you consider the ETP to extract good information |
|---|---|
| Response | We applied your comment, and made additional graphs. |
| Revision |
[Figure]

Figure 5 Monthly irrigation water requirement and rainfall in Khouribga |

**Comments 12**

| Reviewer's comments | 12- Line 64: (Taleb 2006) a comma is missing |
|---|---|
| Response | We revised it. |

**Comments 13**

| Reviewer's comments | 13- Line 74: (OCP 2013) a comma is missing |
|---|---|
| Response | We revised it. |

**Comments 14**

| Reviewer's comments | 14- Figures from 1 to 6: add short descriptions |
|---|---|
| Response | We added the short descriptions. |

---

## Author Response (AR2)

**Revision Notes**

**Reply to the reviewers for hess-2020-197**

Title: A Water-Energy-Food Nexus Approach for Conducting Trade-off Analysis: Morocco's Phosphate Industry in the Khouribga Region

Authors: Sang-Hyun Lee, Amjad T. Assi, Bassel T. Daher, Fatima E. Mengoub, Rabi H. Mohtar

Dear Editor and Reviewer

Thank you for considering the manuscript for publication in the **Hydrology and Earth System Sciences (HESS)** and in-depth review of the manuscript.

We appreciate your thoughtful comments and corrected minor remarks.

Kind regards, on behalf of all co-authors,
* * *
Sang-Hyun Lee

**Revision Notes**

**Minor comments**

Line 200: abd --> and.
➔ We changed it.

Line 207: UB seems to be not defined.
➔ UB and COZ are the name of facilities for drying products, Thus, we removed the UB and COZ, and used "Drying" as process in supply chain.

Line 323: remove "of".
➔ We removed "of".

Line 365 form --> from.
➔ We changed it.

Line 418: in compared --> compared.
➔ We changed it.